# Dendritic calcium signals in rhesus macaque motor cortex drive an optical brain-computer interface

Eric M. Trautmann[1,2,15✉], Daniel J. O'Shea [1,2,15✉], Xulu Sun [3,15✉], James H. Marshel[4], Ailey Crow[4], Brian Hsueh[1], Sam Vesuna[4], Lucas Cofer[2], Gergő Bohner[5], Will Allen[1], Isaac Kauvar[1], Sean Quirin[4], Matthew MacDougall[6], Yuzhi Chen[7,8,9], Matthew P. Whitmire[7,8,9], Charu Ramakrishnan [4], Maneesh Sahani [5], Eyal Seidemann [7,8,9], Stephen I. Ryu[2,10], Karl Deisseroth [1,4,11,12,13,16✉] & Krishna V. Shenoy [1,2,4,12,13,14,16✉]

Calcium imaging is a powerful tool for recording from large populations of neurons in vivo. Imaging in rhesus macaque motor cortex can enable the discovery of fundamental principles of motor cortical function and can inform the design of next generation brain-computer interfaces (BCIs). Surface two-photon imaging, however, cannot presently access somatic calcium signals of neurons from all layers of macaque motor cortex due to photon scattering. Here, we demonstrate an implant and imaging system capable of chronic, motion-stabilized two-photon imaging of neuronal calcium signals from macaques engaged in a motor task. By imaging apical dendrites, we achieved optical access to large populations of deep and superficial cortical neurons across dorsal premotor (PMd) and gyral primary motor (M1) cortices. Dendritic signals from individual neurons displayed tuning for different directions of arm movement. Combining several technical advances, we developed an optical BCI (oBCI) driven by these dendritic signalswhich successfully decoded movement direction online. By fusing two-photon functional imaging with CLARITY volumetric imaging, we verified that many imaged dendrites which contributed to oBCI decoding originated from layer 5 output neurons, including a putative Betz cell. This approach establishes new opportunities for studying motor control and designing BCIs via two photon imaging.

[1] Neurosciences Graduate Program, Stanford University, Stanford, CA, USA. [2] Department of Electrical Engineering, Stanford University, Stanford, CA, USA. [3] Department of Biology, Stanford University, Stanford, CA, USA. [4] Department of Bioengineering, Stanford University, Stanford, CA, USA. [5] Gatsby Computational Neuroscience Unit, University College London, London, UK. [6] Department of Neurosurgery, Stanford University, Stanford, CA, USA. [7] Center for Perceptual Systems, University of Texas, Austin, TX, USA. [8] Department of Psychology, University of Texas, Austin, TX, USA. [9] Department of Neuroscience, University of Texas, Austin, TX, USA. [10] Department of Neurosurgery, Palo Alto Medical Foundation, Palo Alto, CA, USA. [11] Department of Psychiatry and Behavioral Science, Stanford University, Stanford, CA, USA. [12] Howard Hughes Medical Institute, Stanford University, Stanford, CA, USA. [13] Wu Tsai Neuroscience Institute, Stanford University, Stanford, CA, USA. [14] Department of Neurobiology, Stanford University, Stanford, CA, USA. [15] These authors contributed equally: Eric M. Trautmann, Daniel J. O'Shea, Xulu Sun. [16] These authors jointly supervised: Karl Deisseroth, Krishna V. Shenoy. ✉email: etrautmann@gmail.com; djoshea@gmail.com; xlsun79@gmail.com; deissero@stanford.edu; shenoy@stanford.edu

Understanding mechanisms by which populations of neurons give rise to behavior is a primary goal of systems neuroscience, and a foundational step for developing brain–computer interfaces (BCIs) to treat people with neurological injury and disease. BCIs seek to restore lost function by decoding neural activity from the brain in real time to control a medical device such as a prosthetic arm or computer interface[1,2]. Investigations using rhesus macaques (*Macaca mulatta*) have served a vital role in developing clinically-viable BCIs, by exploring decoding algorithms and system designs and by advancing our basic scientific understanding of the motor system[3–13].

Typically, such experiments in monkeys use intracortical multielectrode arrays, which are nearly identical to those approved for use in humans (e.g., Utah arrays), to facilitate translation of scientific and technical achievements from pre-clinical research in monkeys into improvements in clinical BCIs. For some aspects of BCI implementation and translation, such as studying biocompatibility, stability, and longevity, using the same implantable sensor is of central importance. For other aspects of BCI experimentation, the central goal is the scientific study of neural population dynamics underlying the control of arm movements and locomotion. Such studies aim to obtain fundamental understanding that can inform the design of future high-performance BCI systems[14,15]. The tremendous recent expansion of tools for measuring activity from large populations of neurons in animal models provides a fertile experimental landscape for exploring the design of next-generation BCIs[16,17]. Here, we describe such an approach by demonstrating two-photon (2P) imaging in macaque motor cortex and leveraging this to implement an optical BCI (oBCI).

In recent years, techniques for imaging activity of large populations of neurons have improved rapidly. Imaging calcium dynamics using GCaMP is a popular and widely used modality, and has proved successful for measuring primary visual cortex and somatosensory cortex in macaques[18–24] and in marmosets[25,26], as well as primary motor cortex in marmosets[27]. While calcium imaging methods do not yet recover the finely time-resolved spiking activity capable with electrophysiology, they capture a complementary view of neural population activity by contextualizing activity within a densely sampled, spatially localized, and genetically annotated map of the neural tissue.

Optical methods can also readily access large neural populations[16,28], and can access additional neurons or brain areas simply by translating the objective lens or adjusting the scan pattern. An oBCI is therefore well suited for enabling researchers to explore optimal approaches for measuring neural activity (e.g., which area(s) to record from, how many neurons are needed, electrode density and distribution, etc.). The knowledge gained can help set the design specifications for future electrode-array based BCIs. Using optical techniques, it is possible to dissociate the limitations of present-day electrode arrays, which are surgically implanted in a fixed location, from the fundamental study of neural population activity critical to the design of next-generation BCIs.

In this study, we developed an oBCI that operated by decoding neural population signals in real time and at single-cell resolution in macaque motor cortex. In electrical recordings, neurons throughout the motor cortical lamina display tuning for different movements, making it desirable to access signals from both superficial and deep layers[29]. However, due to the light-scattering properties of brain tissue, it is not currently possible in primates to image somatic calcium signals of deep layer five neurons, which serve as the primary motor cortical output of movement signals to subcortical motor circuits and the spinal cord[30]. Reliably imaging neuronal somas at this depth would require

fundamental advances in physics and imaging technology or surgical implantation of a large (>1 mm diameter) penetrating lens[31,32].

Implanting such a lens, however, has unique drawbacks and benefits relative to surface 2P imaging: it requires lesioning and occasionally removing tissue immediately adjacent to the imaged tissue region. It is unclear the extent to which this tissue disruption impacts the neural population dynamics of the tissue under study, though recent studies using this approach demonstrated that neurons imaged in the motor cortex display tuning to reach direction [32]. With the penetrating lens approach, imaging a new population of neurons in a different field of view requires a new surgical implantation procedure. Despite these limitations, however, the microendoscopic imaging approach is compatible with head-fixation free imaging and optical access is easier to maintain, while still providing single-cell resolution and the ability to use genetic targeting strategies.

Here we developed a fundamentally different approach to imaging calcium signals, including signals from neurons in both deep and superficial layers. This is of high value because neurons across all layers potentially generate signals relevant to understanding motor control and for driving BCI decoders[29,33]. Cortical neurons, particularly layer five output neurons, extend apical dendrites to the most superficial layer of cortex. These apical dendrites have been previously shown to generate dendritic spikes as shown in rat and human neuron recording[34], and represent behaviorally relevant information using calcium imaging in mice[30,35–37], presenting an opportunity to record signals originating from deep neurons by imaging their superficial compartments, shown schematically in Fig. 1. This possibility is particularly intriguing in light of recent work demonstrating that dendritic calcium transients are highly correlated with somatic activity in layer five neurons[38]. Nevertheless, to our knowledge, these superficial signals have not been specifically recorded using electrical or optical methods for either

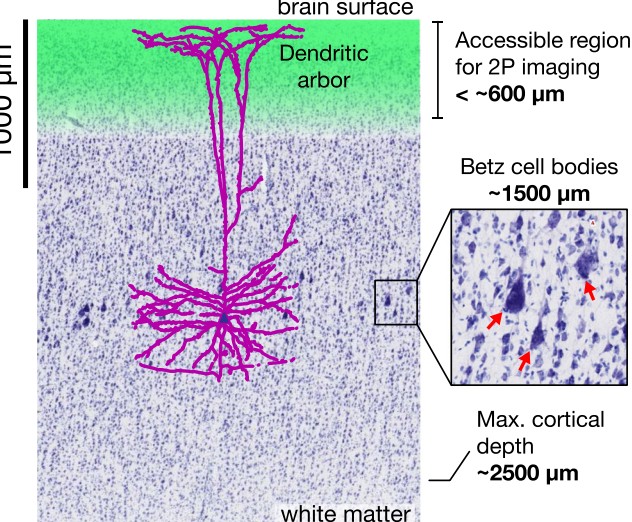

## Macaque motor cortex

**Fig. 1 Dendritic calcium signals are readily accessible from layer five neurons.** Two photon (2P) calcium imaging is currently capable of recording neural activity from the surface down to approximately 600 µm (green region), but photon scattering poses a challenge for imaging deeper. In this work, we demonstrate that it is possible to record neural activity of Layer 5 pyramidal neurons with cell bodies approximately 1500 µm below the surface (red arrows, inset) by imaging apical dendrites in superficial layers (purple), in addition to somatic signals from neurons located in layers 2/3 (blue cell bodies). Nissl stain image source: brainmaps.org.

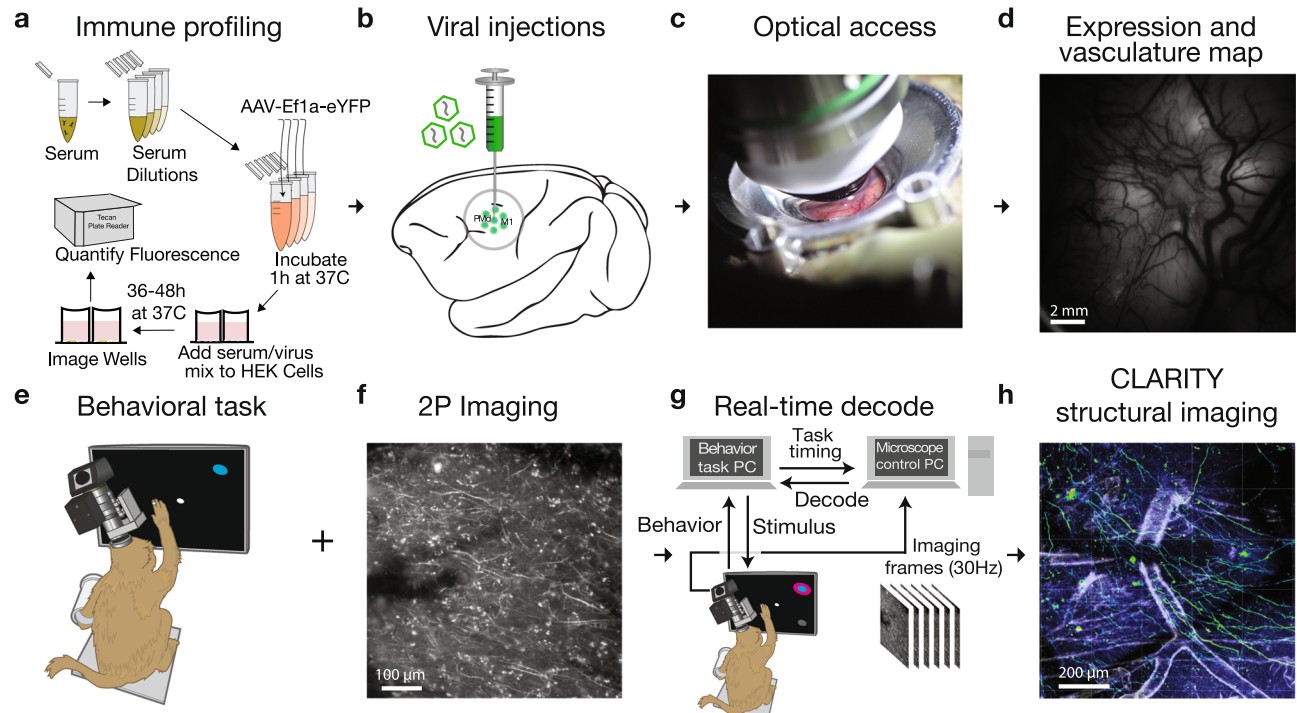

**Fig. 2 Experimental pipeline for combining functional imaging during motor behaviors with structural imaging in macaque monkeys. a** Prior to imaging, we performed a neutralizing antibody assay in order to select an appropriate viral serotype tailored to the immune response of each monkey. **b** Viral constructs were injected into cortex to deliver the calcium reporter gene. **c** A chamber designed for chronic 2P imaging in premotor cortex and motor cortex was implanted. **d** Widefield (1P) imaging was used to assess GCaMP expression and establish vascular fiducial markers for navigating to specific sites on the cortex (representative example, single image collected prior to most recording sessions). **e** A macaque was trained to perform a reaching task to radially arranged targets. **f** 2P imaging was used to obtain functional signals at single-cell resolution from motor cortex (contrast-enhanced mean-intensity projection from one representative dataset of 36 sessions). **g** During training trials, a decoder was trained on the imaging data obtained during reaching movements. Subsequently, during test trials, this decoder was run in real time to decode (predict) the reach target from 2P imaging data. **h** Ex vivo CLARITY was performed to identify cell morphology, projection patterns and cell type (anti-GCaMP antibody labeling green, vasculature white, and results from single imaging session).

basic science or BCI applications in macaques or other nonhuman primates (NHPs); thus, it remains unknown whether these superficial non-somatic signals could drive a BCI and provide insight into neural population dynamics.

In this report, we sought to determine whether 2P calcium imaging of superficial neural processes in macaque motor cortex provides stable reaching-related signals capable of driving a real-time BCI. Achieving this experimental capability required combining multiple technical advances spanning implant design, optics, genetics, and low-latency computation (Fig. 2). We performed immune profiling for individual macaques to measure pre-existing antibodies before and after viral injection. We designed an imaging implant to stabilize brain movement encountered when imaging in the motor cortex during reaching behavior. We found that the forces produced during arm reaching were sufficient to induce significant tissue movement with conventional head restraint systems, presenting a greater stabilization challenge than faced in studies of macaque visual cortex[19–21,39] or in marmosets[26,27]. Coupled with the additional challenge of imaging fine neural processes in addition to somatic signals, our stabilization-optimized implant and head restraint system proved essential to imaging during motor behaviors.

These technical advances collectively enabled imaging of neural activity in macaque motor cortex with micron resolution across a large field of view, providing access to somatic and dendritic neural sources. We leveraged widefield (1P) imaging of vascular landmarks and GCaMP expression with 2P imaging to repeatedly localize the same neurons and dendrites across many experimental sessions, and observed movement tuning in single

dendrites. We decoded these superficial neural signals in real time using an optimized, low-latency image processing pipeline to drive a real-time oBCI. We also localized functionally-imaged neural sources in a post-mortem 3D CLARITY volume[40,41], and demonstrated that layer 5 cells with GCaMP-labeled apical dendrites were densely represented within this tissue volume. In one case, dendritic signal sources originated from a Betz cell, a class of very large, corticospinal projection neuron.

This study serves as a proof of concept for leveraging a non-somatic neural compartment from which to record BCI-relevant neural signals via optical neural recording technologies. This approach opens up opportunities for studying BCI technology designs[42], e.g., optimizing electrode placement in a 3D volume to maximize information extraction, balancing tradeoffs between dense sampling of a small neural volume against broad sampling over a larger volume. Our demonstration also illustrates new possibilities for investigating how motor cortex controls arm movements by combining the benefits of optical functional imaging in awake, behaving NHPs with postmortem anatomical imaging. Importantly, these capabilities include the ability to make closed-loop experiment adjustments based on real-time neural read-outs[43,44].

## Results

**Obtaining optical access to motor cortex.** Obtaining optical access to the brain in rhesus monkeys is especially challenging for 2P calcium imaging due to the large sizes and short working distances of current multiphoton objective lenses. 2P imaging

requirements place constraints on the geometry of an implantable imaging chamber, while the implant must also provide tissue stabilization to minimize motion from pulse, respiration, and forces placed on the implant during natural behaviors in a motor task. Lastly, this chamber must remain sealed except during experiments but also provide easy access to the edge of the dura for cleaning and maintenance.

To address these challenges, we developed an imaging implant that provided imaging access to a 12 mm diameter region of cortical tissue using a commercially available objective lens (Fig. 3a, b). This chamber used an implantable titanium cylinder and silicone artificial dura and was designed to balance the competing goals of maximizing the volume of imageable tissue, while minimizing the craniotomy diameter[45,46]. The chamber also provided access to the edge of the dura to remove new tissue growth and to allow for flushing fluid to reduce the risk of infection, and to enable thorough cleaning if infections did arise[47].

Functional 2P imaging requires stabilizing neural tissue at the scale of microns during an imaging experiment. Changes in intracortical pressure due to pulse and respiration can cause periodic tissue motion inside the skull at the scale of many hundreds of microns, which presents a considerably greater challenge in macaques than in rodents or other smaller animals. In principle, fixed-depth rigid windows may be used to restrict tissue motion, as is common in rodent experiments. In our experience, for chronic implants on the dorsal aspect of the skull, the brain may often recede from the widow over time rendering the stabilizing pressure of the window ineffective, in contrast with recent demonstrations of 2P imaging in V1[20,39,48]. To address this, we developed a removable tissue stabilizer, which mounts inside the recording chamber and uses a glass coverslip to apply gentle downward pressure on the surface of the artificial dura, restricting residual motion of the cortex (Fig. 3b). Importantly, this pressure was only applied during imaging sessions, which reduced the potential for damage due to chronic focal compression. Design files for the imaging chamber and artificial dura components are provided in the supplementary materials.

Micron-scale head restraint is crucial for successfully imaging at cellular resolution. Commercial primate head-fixation systems, designed for electrophysiology, are insufficiently stiff to prevent mechanical flexure at the scale of tens or hundreds of microns from the forces produced by natural arm movements (e.g., ~1 kg arm, order of 1 m/s$^2$ acceleration). We developed a highly rigid three-point head restraint system (Fig. 3c, d), which is conceptually similar to existing halo-style head restraint systems[49–51], but in conjunction with the tissue stabilizer, restricted the motion of tissue within the imaging plane to 1–2 pixels for the majority of imaging sessions (Supplementary Figs. 1–2). This level of mechanical rigidity in the implant and head stabilization (and associated complexity of the preparation) was not required for stable imaging while the monkey was sitting and passively viewing images on a screen, as is done when studying visual-processing in V1 for example[20,48], but was essential when introducing arm movement behaviors.

We integrated this stabilization system into an experimental rig capable of both widefield fluorescence imaging (Fig. 3e) and 2P imaging (Fig. 3f).

## Validating functional GCaMP expression and 2P imaging in motor and visual cortex.
Although it remains unclear whether pre-injection immunological status affects expression of virally-delivered constructs in the CNS[52], pre-existing antibodies may neutralize the virus before transfection and result in low expression. We performed immune profiling for individual macaques to measure preexisting antibodies before viral injection, with the goal of avoiding viral serotypes that might trigger an immune response (see "Methods" section). While our results did not directly show a causal relationship between preexisting antibodies and viral transfection efficiency, we found that monkey subjects with significant pre-injection anti-AAV antibodies could develop immunoreactivity to small volumes of AAV viruses injected into cortex (Supplementary Fig. 3).

We injected AAV1-CaMKIIα-GCaMP6f at two sites in Monkey X and injected two additional sites each with AAV1-hSyn-GCaMP5G and primate codon-optimized (AAV1-CaMKIIα-NES-mGCaMP6f (indicated by the "m" preceding GCaMP, Supplementary Fig. 4) with the nuclear export signal (NES) target peptide. We monitored the expression of GCaMP after virus injection using widefield imaging, and observed first GCaMP expression at 4.5 weeks of post-injection (Fig. 4a, b). In addition, we tested four virus constructs encoding GCaMP6f in the visual cortex of monkey L. These included versions with and without the NES element and with and without the primate codon-optimized. The viruses tested were: (1) AAV1-CaMKIIa-GCaMP6f; (2) AAV1-CaMKIIa-mGCaMP6f; (3) AAV1-CaMKIIa-NES-GCaMP6f; (4) AAV1-CaMKIIa-NES-mGCaMP6f. Robust functional signals were observed via widefield imaging in response to visual stimuli for all four viruses (Supplementary Fig. 5, recorded using a separate imaging setup, see "Methods" section and ref. [21] for more details). We note that because we have not performed a systematic comparison, we cannot be certain whether the primate codon-optimized transgene and/or the NES target peptide influence GCaMP expression in the primate cortex.

Using vascular features identified via widefield imaging as fiducial landmarks, we located the sites of GCaMP expression with 2P imaging and observed neurons expressing GCaMP (Fig. 4b–f). These vascular features provided reliable landmarks, allowing for relative localization of the imaging field of view within the imaging chamber, and subsequent identification of individual neurons across imaging sessions. This approach allowed us to return to a specific field of view repeatedly to image neuronal processes over a timespan of several weeks (Supplementary Fig. 6). In principle, this approach should also work for considerably longer timescales. These landmarks also facilitated identification of neurons in ex vivo imaging using CLARITY (Fig. 4f and shown later in Figs. 8 and 9).

2P imaging fields of view included all fluorescent signals present in superficial cortical layers, typically 150–350 μm deep, including somatic signals in superficial cortical layers (Fig. 4e, f) as well as calcium signals from GCaMP-expressing dendrites (Fig. 4g). In several injection sites across two subjects (monkeys W and X), we also observed bright fluorescence from some cell bodies which appeared to have filled nuclei and non-modulated signals from somas located up to 550 μm from the surface of cortex, and appeared within 5 weeks of injection (Supplementary Fig. 7). The GCaMP expression we observed in monkey X is relatively sparser than typical rodent work or that observed by other groups imaging in macaque V1 (see e.g., [20,48]), though this may result from differences in injection protocol, virus batch, immune status of the individual subject, or other factors. The decode results presented here were imaged at injection sites for the AAV1-CaMKIIα-GCaMP6f construct, located in M1, where we observed the densest expression and most functional tuning.

We identified ROIs within each of the 36 imaging datasets using Suite2P, an open-source tool which does not require assumptions about the shape of ROIs and can readily identify elongated structures belonging to neural processes[53]. Suite2P identified 596 ± 126 ROIs (mean ± s.d.) across the 36

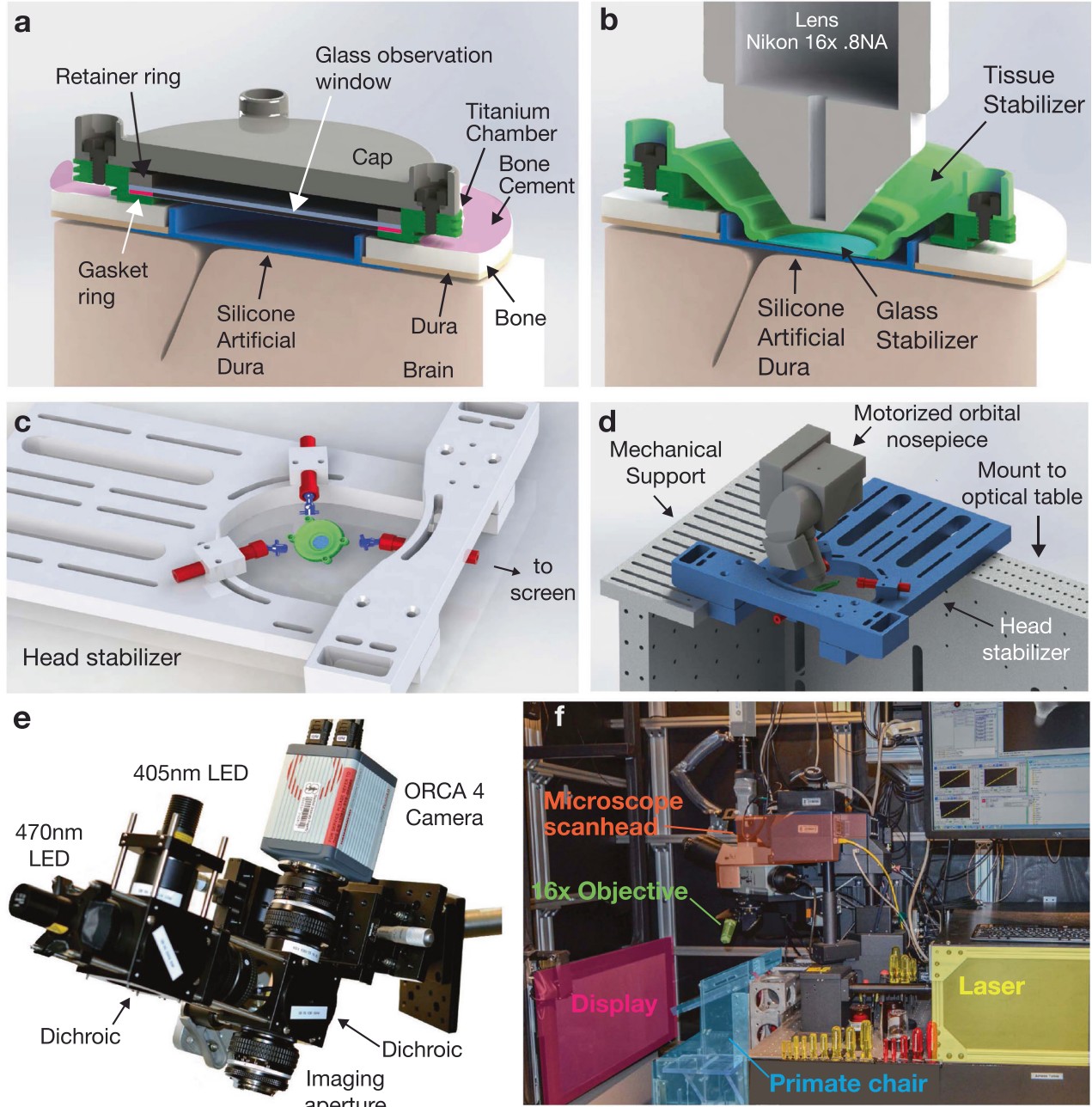

**Fig. 3 Implantable chamber and imaging apparatus. a** Implantable titanium chamber in non-imaging configuration enabled observation through a glass window if the cap is removed. The glass window enabled long-term application of antibiotics and drugs to help maintain the health of the tissue margin. **b** During imaging, the cap and glass window were removed, and a temporary stabilizer is placed inside the chamber to restrict tissue motion via gentle downwards pressure on the surface of cortex. **c**, **d** While imaging, the implant was stabilized using three-point fixation to reduce motion of the tissue to micron levels. **e** Widefield imaging was performed using a custom microscope (see "Methods" section). **f** During 2P imaging, a macaque sat in a standard primate chair in front of a stimulus display screen. The 2P imaging system is placed in a cantilevered position off the edge of the optical table to access the primate's motor cortex.

imaging datasets (total 21,451 ROIs). Functional modulation of GCaMP-expressing neuronal processes was observed in gyral M1 (Fig. 5a) during motor behaviors and assessed using an instructed-delay reach task, which elicited rapid, straight reaches to each of four radially positioned targets (Fig. 5b). The majority of identified ROIs exhibited a response just prior to or during arm movement (65.9%, rank-sum test, $p < 0.01$). This modulation time-locked to movement was readily visible in single trial raster plots (Supplementary Fig. 8). In contrast, relatively few ROIs exhibited responses time-locked to the target cue (1.3%, rank-sum

test, $p < 0.01$) or to the go cue (1.7%, rank-sum test, $p < 0.01$). During movement, 31.0% of ROIs exhibited direction-tuned responses (ANOVA, $p < 0.01$).

We identified a subset of 4365 ROIs (20.3%, or 121 ± 69 mean ± s.d. per imaging session) corresponding to putative dendritic or axonal processes by using the shape of the ROIs (see "Methods" section, Supplementary Fig. 9). This subset of ROIs similarly responded significantly to movement (81.3%, rank-sum test, $p < 0.01$) with few responses to the target cue (2.8%, rank-sum test, $p < 0.01$) or the go cue (1.5%, rank-sum test, $p < 0.01$). Direction-

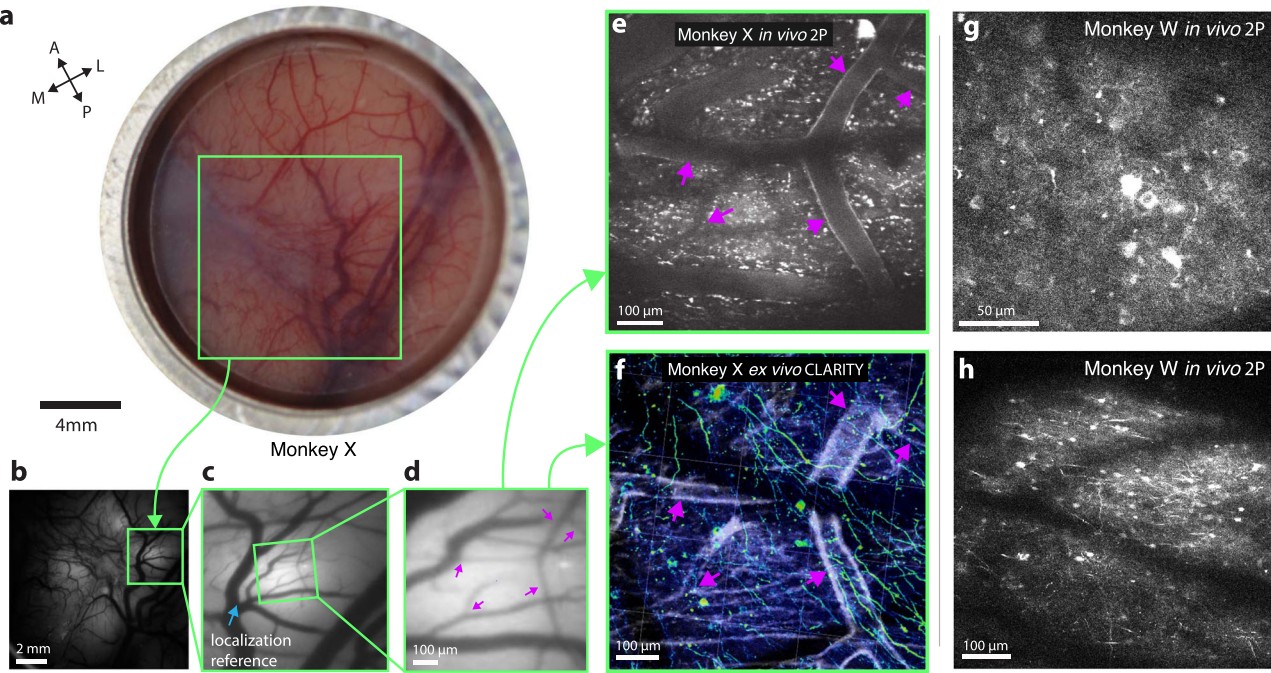

**Fig. 4 Multiscale, multi-modal imaging. a** Imaging chamber with stabilizer in place under ambient illumination, approximately two weeks after implant. **b** Cortical surface imaged using widefield (1P) imaging (representative example expression photo taken from the start of each imaging session). **c** Zoomed in region highlighted in green box in **b**. Vascular landmark used to calibrate microscope stage positions indicated with blue arrow. **d** Further zoomed widefield image showing microvascular features used for localizing 2P FOVs and aligning 2P imaging with CLARITY, marked with magenta arrows. **e** 2P image acquired from the same FOV as **d**. vascular landmarks marked with purple arrows. **f** CLARITY volume from the same FOV in **d** and **e** (anti-GCaMP antibody labeling green, vasculature white, results from a single ex vivo imaging session). **g**, **h** Two example fields of view including neural processes and L2/3 cell bodies ~250 μm below the surface vasculature (monkey W, example images from one of ~20 imaging sessions).

tuned responses were observed in 50.3% of putative dendritic/axonal ROIs (ANOVA, $p < 0.01$). ROIs that did not meet this selection criterion comprised mainly processes that cut through the imaging plane at an angle (appearing as small puncta), and background neuropil without clearly distinguishable processes. Relatively few somatic signals were identified at the superficial planes at which we imaged.

We constructed a tuning map for all ROIs within a FOV by regressing calcium responses against a 50 ms time-lagged hand velocity signal (Fig. 5c and Supplementary Fig. 10), which empirically maximized the correlation with hand kinematics (Supplementary Fig. 11a). Trial-averaged responses of dendritic or axonal ROIs generally exhibited consistent increases or decreases in activity preceding hand movement (Fig. 5d and Supplementary Figs. 10d–i and 11b). Using demixing-PCA[54] and targeted dimensionality reduction (TDR)[55], we identified three dimensions of neural activity (weighted linear combinations of direction-tuned ROIs) exhibiting a condition-independent signal[48] and reach-direction dependent signals (see "Methods" section). Single trial neural trajectories exhibited peri-movement modulation consistent with what we and others have previously observed using firing rates measured from M1 using electrode arrays[49,50] (Fig. 5e and Supplementary Fig. 10e, j). Within the TDR subspace, single trial trajectories also separated according to reach direction, consistent with the direction tuning observed in individual ROIs. Using PCA on the raw imaging data, we found that 6–8 dimensions captured the majority of variance during this task (Supplementary Fig. 12).

Within each experimental session, estimated tuning remained stable for the majority of ROIs; only 11.3% or 752/6653 direction-tuned ROIs exhibited a significant drift in preferred direction (shuffle test, $p < 0.01$, see "Methods" section and Supplementary Fig. 13). Pixel-wise direction tuning within individual ROIs was

generally homogenous; only 3.1% of ROIs and 2.8% of putative dendritic/axonal ROIs exhibited significant tuning heterogeneity (assessed using k-means clustering, see "Methods" section). We returned to certain fields of view across multiple experimental sessions and observed similar neural structures, although we were not able to quantitatively assess tuning stability over time (Supplementary Fig. 6).

**Real-time decoding of reaching from 2P imaging of dendrites.** To demonstrate real-time decoding of arm movement from functional neural imaging responses, we implemented a real-time, discrete target decoder as has been previously developed using electrophysiological recordings[7,8,56]. Here the discrete-target oBCI was driven by a low-latency image processing pipeline, trained to decode reach direction from dendritic calcium signals (Fig. 6a). The majority of GCaMP-expressing neurons with functional signals in this experiment were located in gyral M1, caudal to premotor cortex. We did not observe sufficient preparatory activity from signals in this recording area to facilitate decoding the upcoming reach target from delay period signals[7,8].

At the start of each decoding session, training data were obtained by storing 2P imaging frames during the peri-movement period of natural arm reaching for 8–15 trials per target direction (Fig. 6b). We used the trial-averaged peri-movement imaging frames from the training data to train the decoder (Fig. 6c). After this training period, target classification was performed by analyzing frames during the same peri-movement integration window ($T_{int}$), which began at a fixed delay after the Go Cue ($T_{skip}$) instructing the monkey to reach (Supplementary Fig. 14). We compared these frames with the per-reach-direction template frames learned during training using a minimum mean squared error (MMSE) decoder (see "Methods" section). The decoded target was subsequently sent to the task controller and a visual

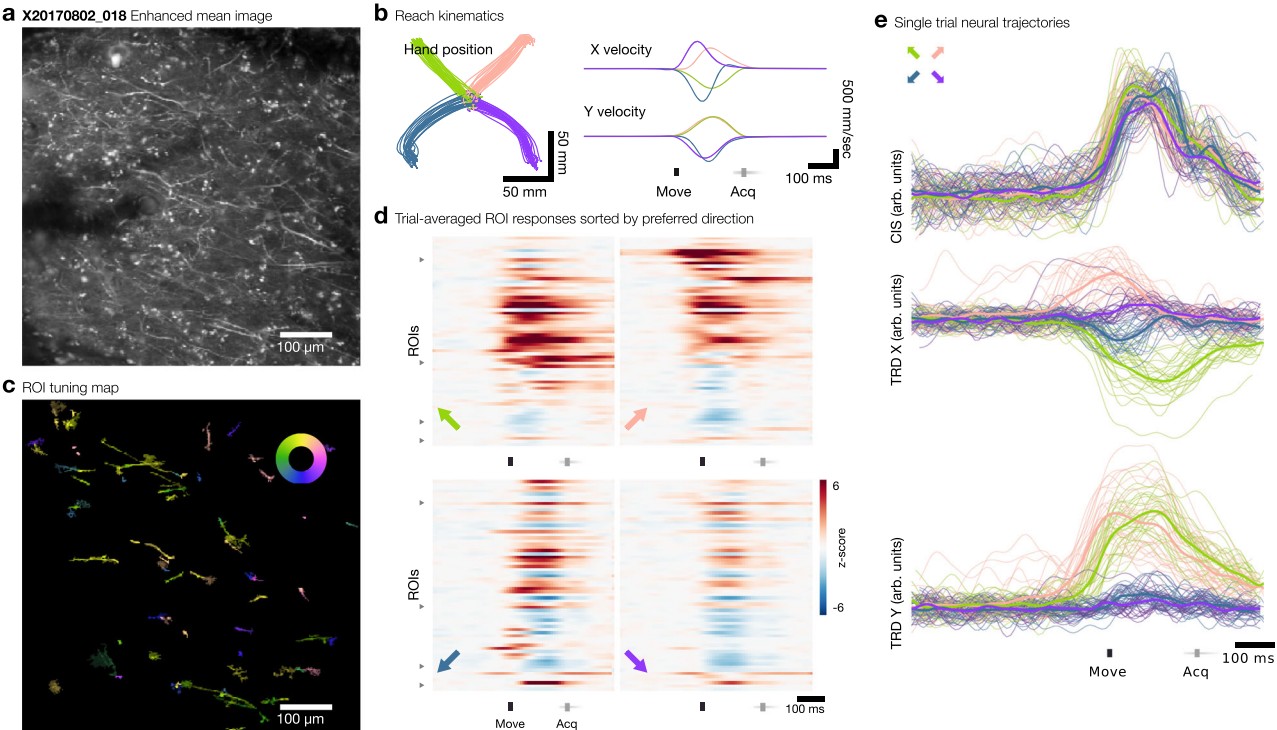

**Fig. 5 Functional responses during reaching behavior. a** Contrast enhanced mean image of example FOV from Monkey X, site 2, one of 36 analyzed imaging sessions. **b** Reaching kinematics observed during behavioral task. Move indicates movement onset; Acq indicates target acquisition. **c** ROI tuning map showing preferred direction of ROIs with significant direction tuning. Inset color wheel indicates reach direction. **d** Trial-averaged responses of dendritic ROIs, normalized to baseline fluorescence, for the four reaching directions indicated by the colored arrow in the bottom left of each raster. ROIs are sorted by preferred direction beginning with rightwards and proceeding counterclockwise; triangular ticks at left edge indicate locations of preferred directions of up-right, up-left, down-left, down-right. **e** Single-trial (thin lines) and trial-averaged (thick lines) population trajectories aligned to behaviorally-defined movement onset, projected along condition-independent signal (CIS) dimension and condition-dependent TDR X and Y dimensions (see "Methods" section), color coded by reach direction condition.

cue was presented to the monkey to indicate decode success or failure. An additional liquid reward was provided after successful decode trials. The total processing time after frame acquisition was less than 15 ms (<1/2 of the imaging-frame period, measured from end of final galvanometer scan line), and the decoded target was visually rendered on the screen prior to the monkey touching the target with his hand. In this experiment, the decoded target presentation did not (at least directly) affect the monkey's motor output, but instead served to ask whether signal-to-noise ratio of the optically recorded neural population was sufficiently high, even with a brief $T_{int}$, to drive a functioning, real-time optical behavioral decoder.

We note that this online decoder uses raw pixel values and not detected ROIs. As such, it is likely that many pixels can contain modulated signal (even if weakly modulated) despite not obviously being associated with a neural process or soma, particularly in the cases where (1) a neuron may be only very weakly expressing GCaMP, or (2) a neuron with particularly shallow modulation depth, or (3) a pixel contains signals from multiple dendrites or other sources (e.g., due to the anisotropic focal volume of the 2P laser due to the extended z-axis of the point spread function).

We assessed the online decode performance for selecting the correct target, from either two or four possible targets, and were able to achieve up to 86.6% successful decode for two targets ($p = 2.17e-39$, two-sided binomial test, chance level 50%, 290 trials) and 69.9% successful decode for the four-target case ($p = 2.6e-22$, two-sided binomial test, chance level 25%, 211 trials) (Fig. 6d). Decoder accuracy for single sessions remained above

chance for hundreds of trials (Fig. 6e), and aggregated decoder performance for all sessions was significantly above chance ($p = 5.60e-6$ for the two target task, 27 sessions spanning 16 days, signed-rank test; $p = 0.0039$ for the four-target task, nine sessions spanning 8 days, signed-rank test; Fig. 6f). We observed a slight but significant decrease in decode performance over the duration of decode sessions (two condition: $-0.05\%$ per trial, $p = 1.40e-28$; four condition: $-0.09\%$ per trial, $p = 2.13e-09$).

Decoder confidence could also be assessed as a function of time within individual trials by assessing the difference in decoder score for sequential frames within a trial. The within-trial decode scores were calculated by averaging the six frames prior to each assessed timepoint, and calculating the decode score for this frame relative to the training data for each condition. These decode score values diverged around the time of movement onset ($t = 0$) and remained separated through the duration of movement (Fig. 6g, larger decoder value indicates that frames are more dissimilar from training data).

We also compared the online MMSE decoder with an offline decoder to understand the timing of signal divergence from these signals, and to benchmark our online performance against an optimal linear classifier. The offline decoder was implemented using a multiclass support vector machine classifier with 5-fold cross-validation. We observed that offline decode performance diverge from chance levels ~250 ms after the go cue presentation (Fig. 6h) and ~100 ms prior to movement onset (Fig. 6i), reaching a mean decode accuracy of 61.3% 200 ms after movement onset. As anticipated, this is somewhat better performance than the mean online decode performance for

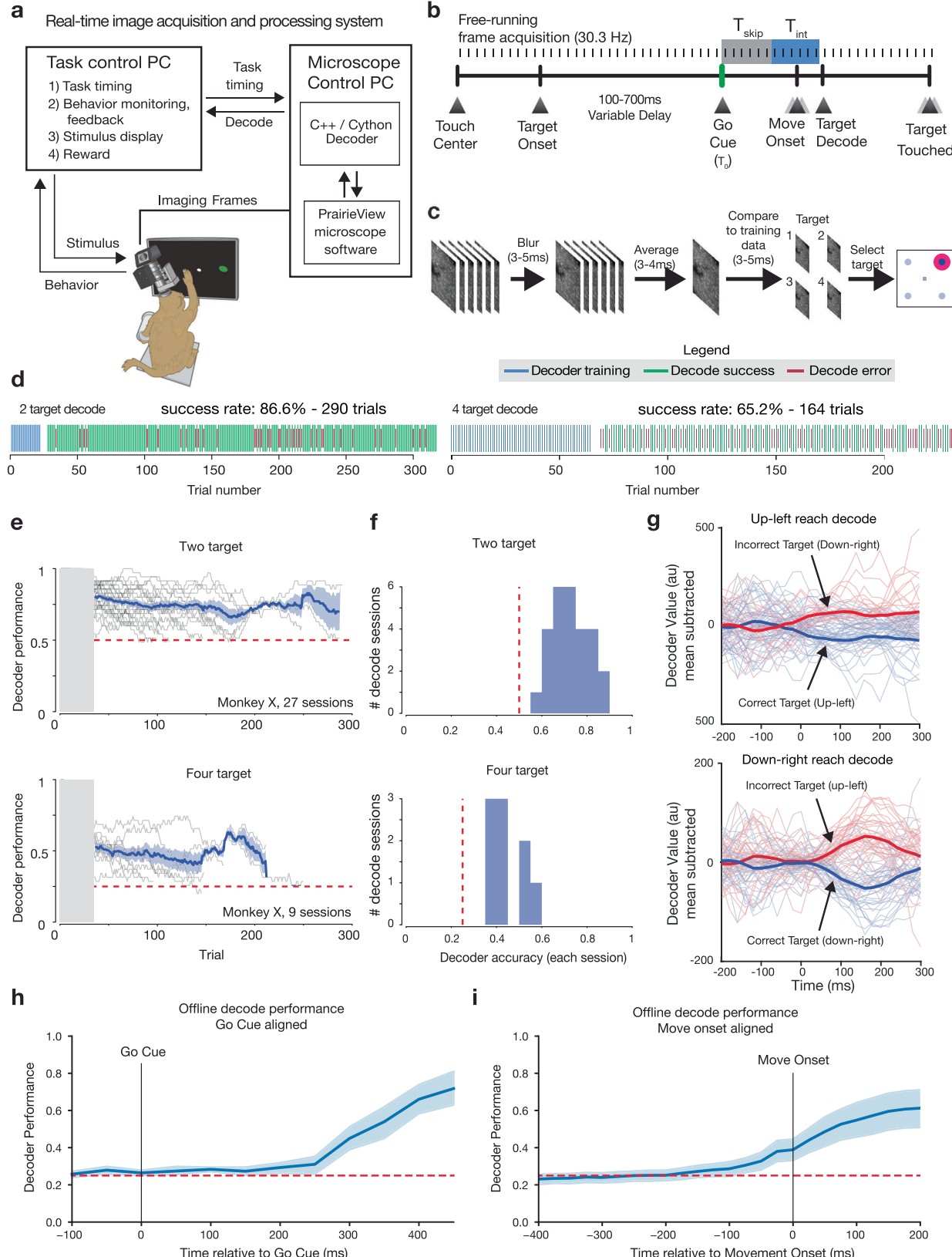

4-condition decode (45.4%, nine sessions). Lastly, we assessed the impact of artificially dropping (blanking) pixels on overall decode performance and found that decode performance degraded gradually as the fraction of masked pixels increased, eventually reaching chance levels when 100% of pixels were masked (Supplementary Fig. 15).

To rule out the possibility that artifacts could be contributing to improve online decode success, we performed a control experiment by running online decode sessions in fields of view without GCaMP expression, but which included auto-fluorescent puncta with similar geometries to neurons and dendrites. Possible artifacts include movement induced motion of the imaging plane

**Fig. 6 Real-time decode of neural activity from functional imaging in motor cortex. a** Real-time stimulus control was implemented by decoding frames acquired from the microscope directly from memory buffers on the acquisition hardware. This provided rapid low-level access to imaging data to train and implement the decoder. Decode results were sent via ethernet UDP to the task and stimulus control computer. **b** Frames were acquired by the imaging system and are integrated by the decoder during a fixed time window ($T_{int}$) beginning at a fixed latency ($T_{skip}$) after the go cue. **c** The frames acquired during the integration time were lightly blurred, averaged, and decoded using pixel-wise minimal mean squared error (MMSE) relative to training data (see "Methods" section). **d** Timelines of two example oBCI decode sessions (monkey X). The decoder was trained using the first 20–40 trials in a given block (blue ticks). Subsequent trials were classified using MMSE decoder using raw pixel values as features. **e** Decode performance over the course of many individual sessions for two target (top) and four target (bottom) tasks, monkey X (mean ± s.e.m.). Decoder performance was stable for up to hundreds of trials. In many cases, sessions were manually halted to record from a different field of view, not due to decreased decoder performance. Chance decoder performance is indicated by the dashed magenta line. **f** Histogram of mean success rate. Chance decoder performance is indicated by the magenta line. **g** Offline decoder score (cross-condition mean subtracted mean-squared error) using rolling 6-frame average for single trials, lower values represent images closer to the training set. **h** Offline decoder performance across sessions (mean ± s.e.m.) using multi-class SVM on go-cue aligned data **i** Same as **h** for movement onset aligned data.

or minor differences in background illumination due to stimulus position on screen. We performed three decode sessions while imaging in areas with bright endogenous autofluorescence, but far from an injection site and without GCaMP expression. As anticipated, imaging in these areas provided fluorescent signals that did not modulate. When running the online decoder in these sessions, all three performed decode at chance levels (session 1: 2 conditions, 38 success, 32 fail, 54.3%, binomial test $p = 0.55$; session 2: 2 conditions, 65 success, 46 fail, 58.5%, binomial test $p = 0.09$; session 3: 14 success, 14 fail, 50%, binomial test $p = 1.0$). These control experiments argue that oBCI decoding performance was not driven by artifactual signals created by tissue movement.

Though the online decoder operated on raw pixel values, we performed additional offline analysis to establish the relative contribution of dendritic signals to the online decode. As before, we used Suite2P to identify morphologically-defined, putative dendritic ROIs (Fig. 7a, b). We observed that the range of pixel values contained in the training data for the online decoder is greater for dendritic pixels (pixels located within dendritic ROIs) than for non-dendritic pixels (Fig. 7c, d). Similarly, the variance of pixel values across all time points and across the four reach conditions is higher for dendritic pixels than non-dendritic pixels (Fig. 7e).

Next, we asked whether it is possible to decode using dendritic signals exclusively. Dendritic pixels constituted between 1–9% of total pixels across all datasets (Fig. 7f), but offline decode performance remains comparably high when using only these pixels (Fig. 7g). Offline decode performance is significantly higher when using dendritic pixels than when using a randomized selection of non-dendritic pixels, when the number of random pixels is selected to match the number of dendritic pixels (Fig. 7h). Lastly, decode performance was significantly higher for sessions in which the data contained a higher percentage of dendritic pixels (Fig. 7i). We note that this result does not indicate that dendrites carry more information than somatic signals in general, and instead likely reflects that the expression profile that we observed in Monkey X, which contained many strongly-modulated dendrites.

**Identifying functionally imaged cells in CLARITY anatomical imaging**. Having demonstrated that dendritic calcium signals are modulated during movement and are capable of driving an oBCI, we sought to identify the somatic sources of these signals. This was not achievable using in vivo 2P imaging due to the depth from the cortical surface. Instead, we cleared a large tissue volume containing the entirety of motor cortex using CLARITY (cleared volume approximately 2.5 cm × 2.5 cm × 1 cm). We then used fine-scale geometric features of the cortical vasculature and of GCaMP-expressing neuronal processes to locate in vivo 2P

imaging regions within the ex vivo imaged tissue volume. This process is illustrated for one example oBCI decoding session (Fig. 8a–i).

The expression patterns observed in the ex vivo tissue volume demonstrate that many neurons in layer 2/3 and 5 cells expressed GCaMP6 both somatically and in their apical dendrites, which extended towards the cortical surface (Supplementary Movie 1) and arborized within the superficial layer imaged in vivo. We quantified the number of neuron cell bodies that expressed GCaMP6 indicators in cortical layers 2/3 and 5 in one CLARITY imaging volume (approximately 1.3 mm × 1.3 mm × 2.2 mm pictured in Fig. 8j, k). We identified 22 pyramidal cells in layers 2/3 and 31 in layer 5 (Fig. 8j, k, segmented somas and their dendritic arbors are highlighted). We were able to trace seven layer 2/3 and 12 layer 5 somas and their apical dendrites extending to the surface (Fig. 8j). In some instances, we could identify a specific neuron in the CLARITY volume that was also imaged in vivo during oBCI decoding. This was enabled by the presence of clearly identifiable features observed across 2P in vivo functional imaging, 2P in vivo volumetric z-stacks, and ex vivo 2P CLARITY imaging. One example was a directionally tuned apical dendrite of a cell originating approximately 1500 μm below the cortical surface (white dashed rectangle in Fig. 8d, e and reconstructed in Fig. 8h, i). The cell body of this neuron was approximately 60 μm in diameter, suggesting that this cell is likely to be a Betz cell, a class of large upper motor neuron which projects to the spinal cord[57]. A second example is illustrated in Fig. 9, facilitated by the idiosyncratic shape of the GCaMP-expressing dendrite.

## Discussion

We developed an all-optical motor BCI driven by calcium signals, enabled by a suite of engineered optimizations for stable 2P imaging in rhesus macaques engaged in motor tasks. The optical window, as part of the custom-designed implant, affords imaging access to many hundreds of thousands of neurons across PMd and M1 and is compatible with other brain regions. Using this implant, we achieved stable, chronic 2P imaging of motor cortical neurons in awake, behaving macaques. We demonstrated that signals imaged from dendrites located in superficial layers were modulated during movement and exhibited directionally tuned responses. We employed a low-latency image processing to use these dendritic signals to drive a real-time oBCI, which decoded the direction of a monkey's reaching movement from neural activity to provide low-latency visual feedback and reward. We leveraged wide-field imaging of vascular fiducial markers to return to the same neurons in one region over seven sessions spanning 13 days (Supplementary Fig. 6). Lastly, we identified the source of these neuronal processes imaged in 2P with ex vivo CLARITY. This demonstrated that imaged dendrites originated

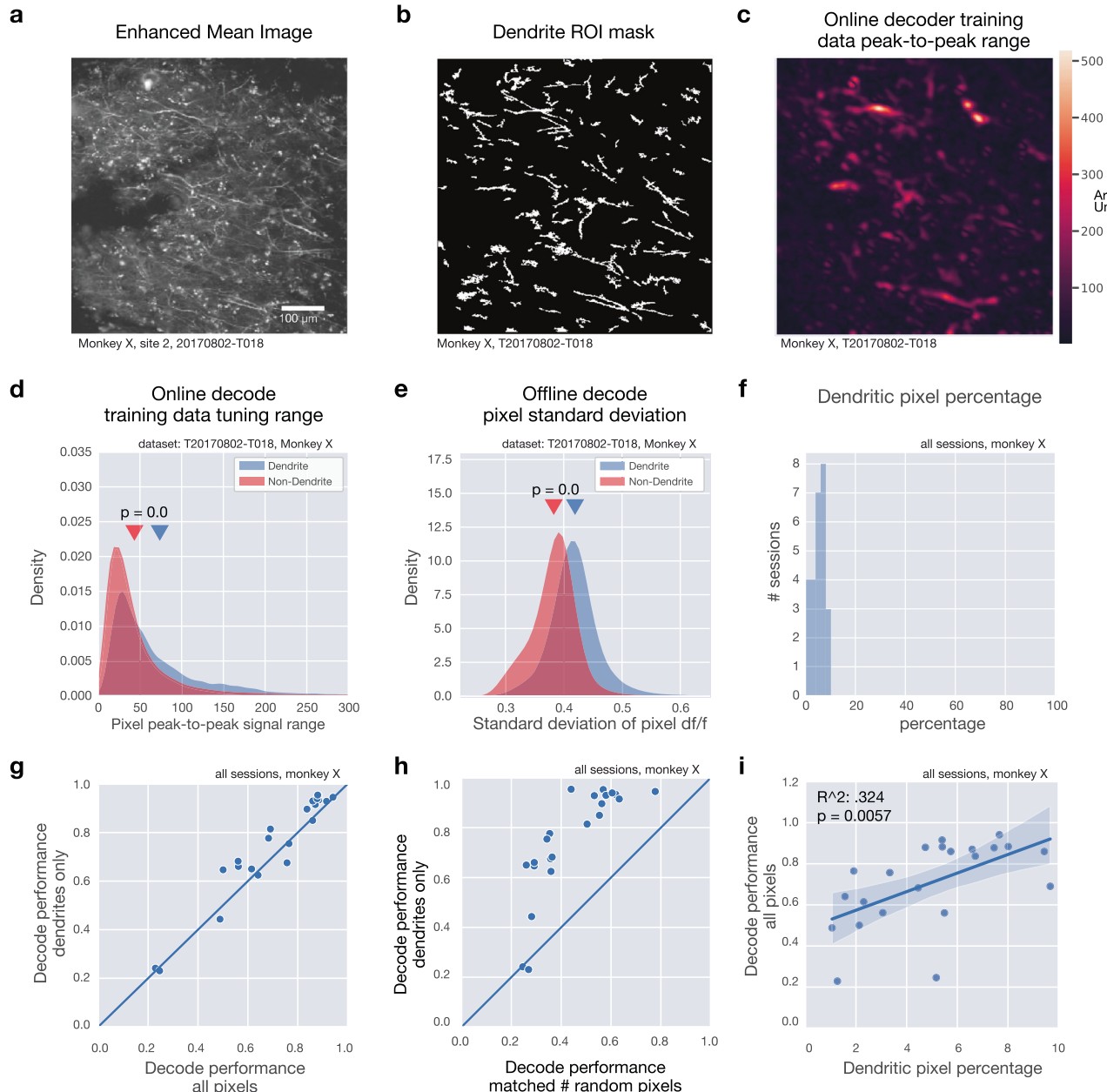

**Fig. 7 Dendritic signals drive online decode. a** Mean intensity projection (contrast enhanced) for example imaging session (one of 36 sessions shown), **b** Dendrite ROI pixel mask for example in **a**. **c** Peak-to-peak pixel signal range across four reach directions for the online imaging decoder training data for example in **a**. Large values indicate large modulation and higher variability across different reach directions. **d** Comparison of distributions of pixel peak-to-peak range between dendritic pixels and non-dendritic pixels (rank-sum test, $U$ statistic = 78.6356, $p$ = 0.0). **e** Comparison of distributions of standard deviation of pixel d$f/f$ value across all timepoints and across reach directions between dendritic pixels and non-dendritic pixels (rank-sum test, $U$ statistic = 113.6273, $p$ = 0.0). **f** Histogram of percentage of pixels inside a dendritic ROI for all sessions, monkey X. **g** Comparison of offline decode performance using dendritic pixels only (ordinate) vs. all pixels (abscissa). **h** Comparison of offline decode performance using dendritic pixels only (ordinate) vs. a random selection of the same number of pixels as those within dendritic ROIs. **i** Decode performance as a function of the percentage of pixels associated with a dendritic ROI for each session. Blue line represents regression fit; shaded area indicates 95% confidence interval.

from layer 2/3 and layer 5 neurons, including from corticospinal projection neurons.

Importantly, we do not suggest that imaging is necessarily a viable recording modality for clinical BCIs in the near term due to the need to introduce exogenous calcium reporters using viral vectors. Instead, we view optical imaging and oBCIs as an important pre-clinical tool to address critical questions that are challenging to address using electrical recording alone[42,58]. Optical imaging and closed-loop oBCIs complement electrophysiology methods by addressing several limitations of current

array technologies, such as providing the potential to accurately track large neural populations over timescales longer than several days, densely recording from 3D volumes of tissue, and merging genetic neural circuit dissection techniques with BCI designs to better understand the neural substrate for neural prosthetic control. Even high-density silicon electrodes (e.g., Neuropixels probes[17,29]) provide limited volumetric tissue coverage, cannot easily be moved once inserted, and lack cell-type information and ability to target genetically-specified cell types. Moreover, the process of engineering appropriate electrical stimuli for sensory

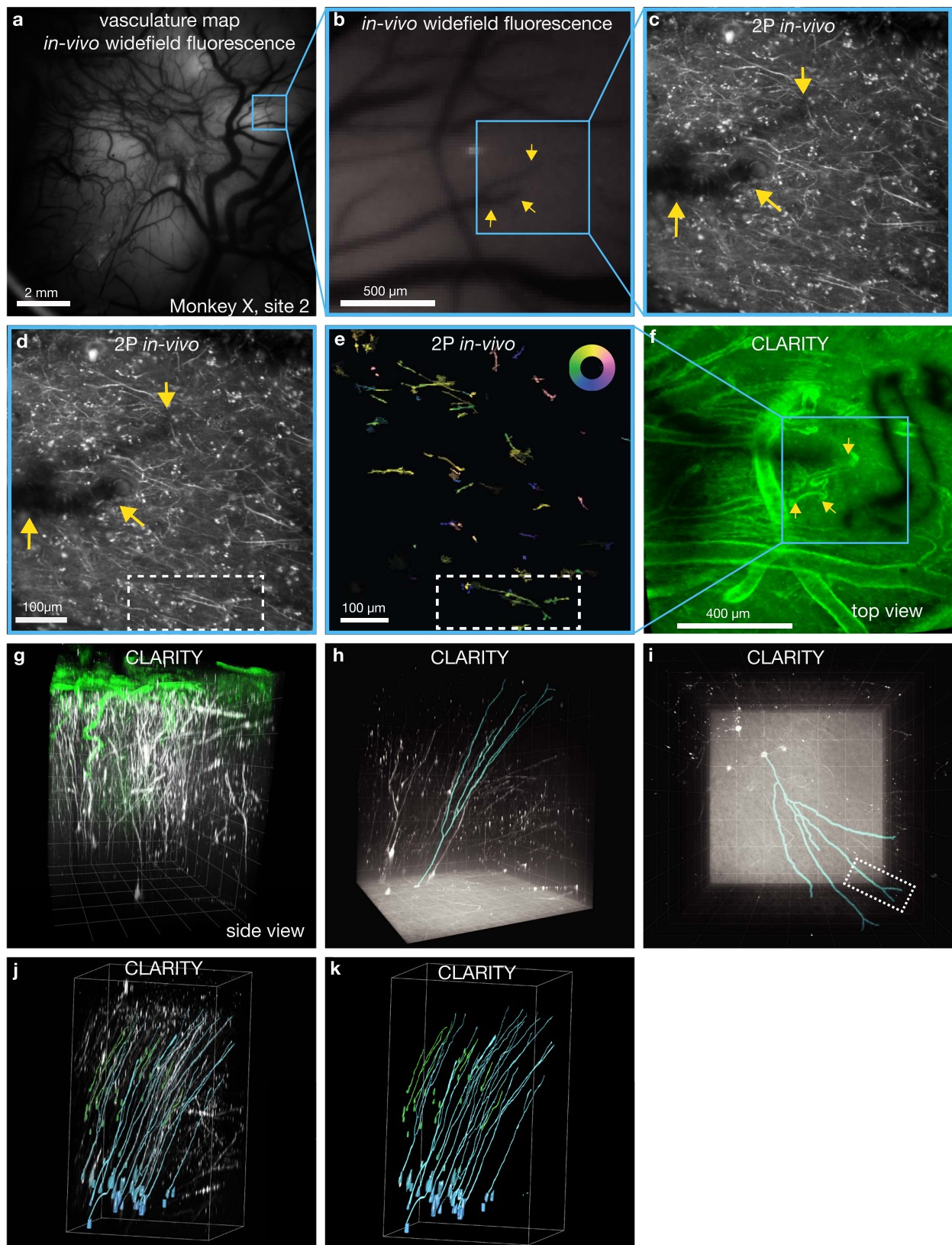

write in[59–61] would greatly benefit from direct optical observation of the complex stimulation-evoked patterns of neural activity[62,63]. Ultimately, imaging methods will deepen our understanding of the basic science of natural movement control and will help advance BCI algorithms, enabling the development of next generation BCIs for animal research and human clinical use.

Achieving stable 2P imaging in monkeys engaged in motor behaviors necessitated several technical advances. First, we developed an optical window optimized for large multiphoton lenses and an accompanying stabilization system to obtain optical access to cortex, which is more challenging in macaques than in rodents or smaller organisms. The artificial dura system

**Fig. 8 Identifying neurons in CLARITY and functional imaging. a** A widefield fluorescence image revealing vasculature landmarks was used to locate and register 2P imaging FOVs (representative example widefield image from the start of each imaging session). **b** Higher-magnification FOV from **a** with example vascular landmark indicated with yellow arrows. **c** Example mean intensity projection of all 2P frames acquired during a decode session using the FOV indicated in **b** by blue box, representative sample from one of 36 sessions). **d** The same image as **c** with selected dendritic processes are outlined in white dashed rectangle. **e** Pixel-wise tuning map, same FOV and presentation style as in Fig. 4h (Inset color wheel indicates reach direction). **f** CLARITY volume showing wide-area surface vasculature (lectin stain, green) with FOV from **c**–**e** in blue box. Vascular fiducial feature from **b**, **c** indicated with yellow arrows. The dark feature on the right is an artifact from an occluding object only present during the CLARITY imaging (images acquired in one ex vivo session). **g** Side-view rendering of CLARITY volume showing a close up of neural processes extending from the FOV in **a**, **b** down to areas below the imageable regions using in vivo 2P imaging (green: lectin, white: GCaMP stain). Grid spacing 200 μm. **h** Rendering of CLARITY volumetric imaging showing a side view of motor cortical tissue spanning cortical lamina with putative Betz cells located in layer 5, approximately 1500 μm below the surface. Blue outline indicates the traced reconstruction of the dendritic process imaged superficially in **e**, traced from the dendritic arbor down to the cell soma. The location and large size of the soma suggests this cell is likely to be a Betz cell (Grid spacing 200 μm). **i** Surface image of reconstructed arbor; white dashed rectangle indicates matching region from **e** (Grid spacing 200 μm). **j** Cell bodies identified in layers 2/3 (green) and 5 (cyan) with their dendrites extending to the superficial layer, as traced by CLARITY and highlighted in different colors. **k** Traced neurons only. Data from Monkey X, injection site 2. **g**–**k** Imaged volume size: 1.27 mm × 1.27 mm × 1.62 mm.

facilitated reliable optical access for many months by maintaining an environmental seal, and enabling frequent cleaning to minimize infection risks for chronic implants after removing the dura. Once optical access had been established, the adjustable tissue stabilization and three-point head fixation system enabled stable imaging despite substantial cardiac and respiratory-driven brain pulsation, as well as brain-in-skull motion caused by the monkey's movements. As previously reported in primary somatosensory cortex[23], direct stabilization of the brain was required for dorsally-located frontal cortex than in the occipital lobe (see e.g., [20,21]) due to the effect of gravity. Additionally, the artificial dura window designed provided access to an 18 mm diameter region of cortex (12 mm is accessible for imaging using a commercially available 16× multiphoton lens), which was considerably larger than the field of view that our 2P imaging system could simultaneously image. An implant similar to our design could also be used with large field of view imaging technologies such as the Mesoscope[16] to address an expanded field of view, thereby vastly increasing the number of simultaneously recorded neurons.

Second, we obtained functional expression of GCaMP6f throughout motor cortical neurons including apical dendrites. We designed monkey codon-optimized GCaMP constructs, some of which included the NES target peptide[64]. While we selected these NES-constructed for the purpose of achieving functional expression in dendrites, we cannot draw any conclusions regarding its efficacy in primate cortex without a systematic comparison between different constructs. The probability of achieving successful expression may also have been improved by performing a neutralizing antibody assay to pre-screen viral serotypes against subject-specific immune status, though systematic comparison is needed to validate this approach. Serotype, selection of genetic promoters, and the viral delivery protocol are all likely to play an important role in determining whether expression levels are healthy and effective. We found that small intracranial injections of AAV could elicit a humoral immune response in pre-sensitized subjects, as measured by the neutralizing antibody assay. While this information can guide the selection of candidate viral serotypes to minimize anti-AAV immune response, more work is needed to determine whether humoral responses to AAVs negatively impact neuronal GCaMP expression. We demonstrated functional expression of GCaMP using AAV1 in macaque motor cortex and validated safe and effective expression of these constructs in a second subject in V1.

Superficial neural processes which expressed GCaMP exhibited robust, direction-tuned responses during reaching movements. In contrast, responses during the instructed delay period preceding movement were surprisingly weak in contrast to preparatory responses observed with electrode arrays[65]. This could potentially

reflect a less faithful representation of motor preparation in the dendrites relative to the cell body, but it might also result from nonlinearities in GCaMP activity, an idiosyncratic expression pattern biased towards neurons with weaker preparatory signals, or other unknown factors. This underscores the need for future research to better understand the relationship between neural dynamics inferred from calcium signals versus from extracellular physiology[66].

Third, we engineered a low-latency image processing pipeline to train and execute the real-time optical decoder, which provided the movement estimates needed for the oBCI system. This pipeline leveraged memory buffers exposed by the microscope hardware and image processing code optimized to achieve low latency to achieve an online decode, which is conceptually similar to a 1D imaging decoder reported in mice[67]. This code is documented and publicly available (see "Code availability" section). This enables a large class of experiments in which neural activity is read out and used to modify the task or stimulus in real time, e.g., using a BCI decoder to study motor adaptation to visuomotor rotation[68]. Combining this capability with genetic targeting can elucidate the roles of different cell types in contributing to motor adaptation and control.

Finally, to identify the source of a subset of the dendritic signals we imaged, we optimized CLARITY to clear and immunostain a large volume of the macaque brain, encompassing the whole of motor and premotor cortices. By localizing blood vessels in both the cleared volume and functional datasets, we were able to localize functional datasets within CLARITY imaging volumes. We used this technique to validate that layer 2/3 and layer 5 neurons sourced the superficially-projecting apical dendrites which drove the oBCI decoder. In some cases, we were able to exploit neuronal morphology visualized through GCaMP expression in dendritic arbors to identify specific neurons.

While this serves as an initial proof of concept, registration of in vivo and ex vivo cleared tissue volumes remains a challenge. Confounding factors include (1) different imaging planes and angles and often different imaging modalities from in vivo to ex vivo, (2) differential fluorescence signal in vivo vs. ex vivo, specifically a much denser network of labeled processes from anti-GFP staining ex vivo, (3) potential tissue deformation from tissue extraction and the clearing process, particularly of the superficial layers. In this particular case, fiducial markers were limited to the surface vasculature and therefore did not provide any information on the depth of the functional ROIs within the CLARITY volume.

Given these challenges, we were not able to register more functional ROIs in the superficial layer with deep-layer cell bodies and dendrites identified by CLARITY besides the demonstrated

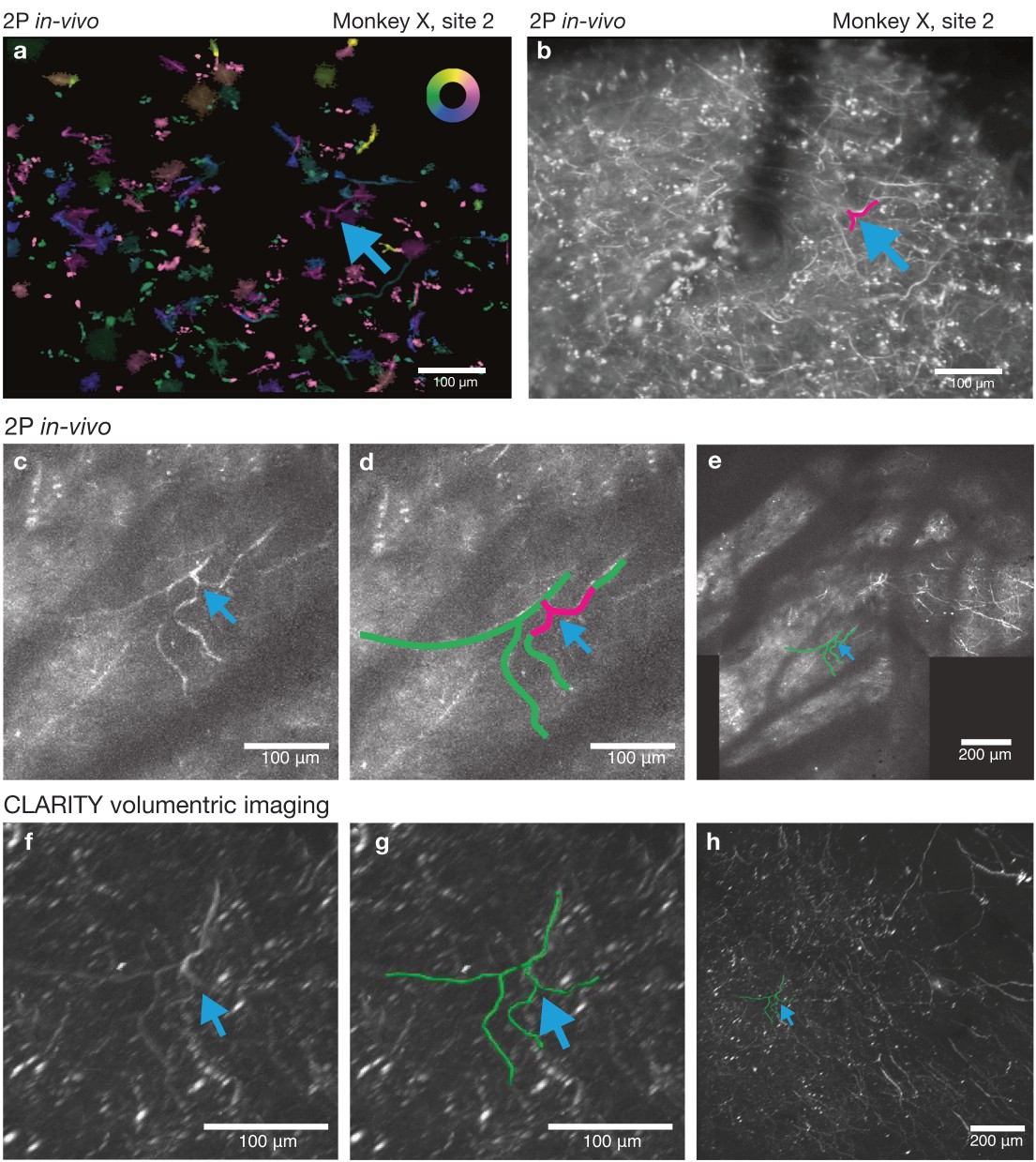

**Fig. 9 A second example of a neural process functionally imaged and reconstructed in the registered CLARITY volume. a** Pixel-wise tuning map, same FOV and presentation style as in Fig. 4h (Inset color wheel indicates reach direction). A neural feature of interest is indicated by the blue arrow. **b**. Mean intensity projection of FOV in **a**. The same process is labeled with a blue arrow. This image represents only a thin slice through the tissue volume. **c** Maximum intensity projection from in vivo volumetric *z*-stack showing projection of the neural features labeled in **a** and **b**. More structure is present in this image than in **b**, since this image is a maximum intensity projection of images acquired at multiple depths. **d** Same image as **c**, with the neural feature of interest from **a–c** traced in green. **e** Maximum intensity projection of stitched 3D volume assembled from in vivo 2P imaging. **f** Closeup view of ex vivo imaged CLARITY volume. The neural feature from **a–e** is marked with the blue arrow. **g** Same as **f** with the feature from **a–e** traced in green. **h** Wide view of CLARITY volume. Data from Monkey X, injection site 2.

examples. However, there are many ongoing efforts to register functional and ex vivo tissue volumes. For example, this has been essentially solved in larval zebrafish[69], where activity imaging is successfully registered to multiple rounds of antibody staining at cellular resolution. In rodents, registration of in vivo and cleared tissue imaging volumes has been aided by sparse labeling and fiducial markers throughout the tissue volume. For example, recordings using Neuropixels probes have been registered to their position within the Allen Brain atlas in mouse by fluorescent-labeling of the electrode tracks[70]. As these techniques are optimized for use in rodents, we expect that they will aid in similar efforts in future primate work.

**Imaging neural projections**. The majority of previous calcium imaging experiments used isolated neuronal somas as the primary signal source. However, in rodents and larger mammals, imaging neuron cell bodies below superficial layers of cortex (down to ~600 μm below the surface) at standard frame rates (~30 Hz) is not currently possible using standard 2P imaging technologies due to photon scattering and absorption. It is not currently possible to image somatic calcium signals from layer five cells in macaque motor cortex, which may be located as deep as 1.5–2.5 mm from the surface, without using a penetrating GRIN lens[71]. A microendoscopic lens provides access to neurons many millimeters from the surface, but is difficult or impossible to

move once placed. In contrast, imaging apical dendrites from the surface can provide access to deeper cells, while allowing experimenters to image different populations of cells by translating the objective lens. The approach we demonstrated here accessed calcium transients from these deeper neurons through their apical dendrites, which arborize near the cortical surface. The data we presented demonstrate that superficial processes in M1 exhibit directionally-tuned movement modulation, consistent with signals from extracellularly-recorded motor cortical action potentials[29]. This correspondence supports additional work to leverage these dendritic calcium signals to explore neural population activity during motor behaviors. Moreover, this approach could also be used to study computations in active dendrites with calcium reporters[36,37] or with fast, genetically encoded voltage indicators[72].

More broadly, imaging of superficially projecting axons could be used to study a variety of neuromodulatory influences on cortical processing. Previous studies in rodents have demonstrated that it is possible to optically record activity from GCaMP-expressing neuronal projections[73,74]. This suggests that by expressing GCaMP in subcortical regions, which source a specific neuromodulatory influence on cortex, the activity of these axonal projections may be imaged directly at their cortical target. For example, dopaminergic afferents from ventral midbrain laminate throughout cortex including superficially in layer 1[75]. Imaging of cholinergic afferents from basal forebrain, known to activate layer 1 inhibitory circuits[76], could elucidate the mechanisms of signal enhancement in primary visual cortex[77,78] and in memory function in prefrontal cortex[79]. Analogous approaches could be used to study the influence of GABAergic inputs from basal ganglia into frontal cortex on reward learning and coordination of motor actions[80], as well as thalamic afferents[81] and cortico-cortical connections[82] with superficial synaptic targets.

**Identifying cell types of recorded neurons.** Electrophysiological recordings are limited in their ability to connect descriptions of neural population dynamics to the underlying neural circuit structure. In particular, using electrophysiology alone, it is not currently possible to study how coordinated population activity arises from the constituent neuronal cell types, the fine spatial organization of cells within the circuit, and connectivity of the network[14,83,84]. Several studies have used spike waveform shape to distinguish putative excitatory and inhibitory neurons[85–87], and with increasing electrode density finer details of each neuron's electrical image onto many nearby channels can better resolve additional cell classes[88,89]. However, it is difficult to validate the accuracy of this approach. Genetic approaches can bridge this gap and provide richer information on the diversity of neurons present throughout cortex[90–92].

Once a broader panel of labeling methods are validated in rhesus macaques and tissue clearing techniques are better optimized for macaque and human tissue[93], we anticipate that the cell-type information provided by co-registered CLARITY volumes will rapidly increase. By providing access to both structure and function, optical methods can discover connections between computational-level descriptions afforded by analyzing neural population dynamics, and the underlying mechanisms that generate and shape these dynamics to drive behavior. In the motor cortex, this opens the door for further work investigating the contributions of distinct cell types to motor control, an approach which has been highly successful in mice[94,95]. We note, however, that as this method requires sacrificing the animal, there are inherent limitations to the timescale and questions that can be addressed with this approach.

**Stable tracking of neural populations across sessions.** In macaques and other NHPs such as marmosets, there is a growing interest in tracking neural populations over time to address a collection of learning-related questions. Multielectrode array recordings in M1 and PMd have been used to study population changes in motor cortex during visuomotor adaptation[68] and force field adaptation[96], and have been used to study degradation of decode performance over time[97]. Recent work has demonstrated methods for making a decoder robust to changes in tuning, and/or recording stability over timespans of months or years[98,99]. Researchers have employed carefully designed perturbations to BCI decoders—in which the behaviorally-relevant readout axes of neural activity are controlled directly by the experimenter—in order to probe constraints on learning imposed by neural circuitry[43,100,101]. Within the context of BCI, researchers are also developing "co-adaptive" BCI algorithms, in which a subject learns to improve BCI performance by generating specific patterns of neural activity via neural plasticity, while the decode algorithm concomitantly adjusts to make the BCI easier to control (see e.g., [4,102], reviewed by ref. [3]). Understanding and facilitating co-adaptation is generally believed to be an important next step in BCI design[103].

Multielectrode arrays, however, are susceptible to waveform drift caused by micron-scale shifts in electrode placement. Tracking the activity of a specific population of neurons across multiple experimental sessions spanning days or weeks is quite challenging[104–108]. Precise tracking of large populations of neurons using imaging could be leveraged to study learning-related changes at single-cell resolution across longer timescales[109]. Optical techniques can meet this need by recording from the same population of neurons across sessions. This is possible because morphological features of the neurons allow for high confidence in identifying the same neurons in multiple images. Using a combination of widefield and 2P imaging, we were able to localize the same field of view using vasculature and GCaMP expression patterns as fiducials. This capability underscores the utility of functional imaging for studying the evolution of neural population dynamics over extended timeframes[110], particularly for experiments involving across-session learning or long timescale adaptation. For example, 2P imaging has proven particularly useful in rodents for observing learning-related changes in neural population activity[111–116], and we believe a similar approach may be employed in primates. We note, however, that the data reported here were not collected with the exact same field of view or imaging depth, leading to the across-session differences visible in supplementary Fig. 6, which make it difficult to perform quantitative assessment of the tuning stability, stability of viral expression, or assessment of image decoder stability across multiple sessions.

When combined, the technical advances reported here enabled stable imaging sufficient to record direction-tuned signals in a population of dendrites and to drive a real-time direction decoder. The observed accuracy rates for the online oBCI decoder were significantly above chance, but are not yet at parity with results obtained using multielectrode electrophysiology methods (e.g., Utah arrays[8]). While our proof of concept oBCI was a discrete decoder for reaching direction, previous simulations have shown that it is possible to decode continuous hand movement trajectories from simulated optical signal, despite calcium indicators having (1) relatively slow response kinetics to action potentials, (2) various sources of noise, and (3) non-linear relationship, including saturation, between the neural spikes and fluorescent signal[117]. Additionally, our analysis in that study indicated that oBCI decoding performance should significantly increase with improvements in the temporal resolution of calcium reporters[118] or with high-SNR, genetically-encoded voltage indicators[72].

The GCaMP expression we observed was sparser than comparable results using virally-transfected GCaMP in mouse[119–121], marmoset[25,27], and primate V1[20,48] for most injection sites. Future applications would benefit considerably from denser and more reliable expression of the reporter construct. Fortunately, considerable advances in macaque optogenetics have provided insight into effective delivery vectors for achieving expression[122]. Reliably obtaining and maintaining healthy levels of GCaMP expression in motor and premotor cortex remains challenging, and is particularly important in highly trained macaques. Future experiments may benefit from subject-specific serotype pre-screening as performed here, careful consideration of serotype and promoter[123], and precision titration of expression (e.g., via tetracycline-gating) to maintain robust, functional signals[25]. We anticipate that achieving denser expression will enable more sophisticated decoding approaches capable of driving the continuous velocity of a cursor or a robotic arm.

This work demonstrates capabilities for combining emerging neural recording technologies with low-latency BCI and ex vivo volumetric, structural imaging. This pipeline provides opportunities to leverage genetic neural circuit dissection techniques within motor neurophysiology studies and BCI experiments. Such an integrative approach is needed to elucidate the neural circuits and mechanisms that control natural movements and neural prosthetic devices.

## Methods

**Animal subjects**. All procedures and experiments were approved for animals S, W, and X by the Stanford University Institutional Animal Care and Use Committee (IACUC), and for animal L by the University of Texas at Austin IACUC and were performed in compliance with the Guide for the Care and Use of Laboratory Animals. Three male rhesus macaques (X, S, and W) were used for 2P imaging experiments and a fourth rhesus macaque (L) was used for secondary validation of virus constructs in V1. Stable 2P imaging during the reaching task was demonstrated in Subjects S, W, and X. Decode experiments were performed with subject X. Subjects S and W did not exhibit functionally tuned responses due to a lack of GCaMP expression (monkey S) or static non-modulated expression (monkey W). Monkey W did exhibit healthy GCaMP expression at some injection sites, but neurons in those sites were not responsive during the task (data not shown). All four animal subjects (S, W, X, and L) were included in the Neutralizing antibody assay results.

**Assay for measuring AAV neutralizing antibodies**. Individual subjects have different immunological states based on their particular exposure history to environmental viruses[124]. Although the brain has immune-privileged properties, whether pre-injection immunological status affects CNS expression of virally delivered constructs or elicits a systemic immune response is uncertain[52]. As such, viral infection and injection-related adaptive immune response may be highly variable between individuals, leading to consequences such as neutralization before infection, low expression of GCaMP, and/or deleterious systemic immune sensitization in certain subjects but not others. We employed a neutralizing antibody assay to quantify the levels of functionally neutralizing anti-AAV antibodies in serum samples.

Our goal was to select specific serotypes of AAV with a higher probability to infect neurons and a lower probability of eliciting an immune response. As anticipated, each macaque subject had a different pre-existing antibody status; monkey S was responsive to AAV1, monkey W was responsive to AAV8, and monkey X to neither (Supplementary Fig. 3c). After viral injection, we found that one monkey subject with significant pre-injection anti-AAVs (monkey W) developed significant antibody responses to AAV viruses injected into cortex (Supplementary Fig. 3d). In contrast, a low anti-AAV monkey (monkey X) demonstrated a response profile that seemed to mimic a primary exposure (Supplementary Fig. 3e), consistent with a previous report[125]. We caution that we do not have direct evidence that our particular serotype selection ensured better expression in our subject. While these results are consistent with our expectations, further experiments and additional subjects are required to validate the efficacy of this approach. In addition, the impact of immunoreactivity on the longevity of expression at healthy levels and on the success of subsequent additional injections remains to be determined.

Blood samples were collected from four rhesus macaques. Blood draws were either performed at Stanford University (monkeys S, W, X) or at the University of Texas at Austin monkey (L), depending on the location of the monkey. Approximately 1 mL of blood was drawn at each collection date. Blood samples

were placed in an IEC Centra GP8 Centrifuge at 1292×g (2500 RPM) for 150 s, transferred to a 1.5 mL Eppendorf, and then centrifuged at 22,131×g (14,000 RPM) for 3 min in an Eppendorf Centrifuge 5430. Remaining serum was extracted and stored at −80 °C until use.

We compared neutralizing antibody titers in blood sera collected before and after intracranial injections of AAVs using a standard in vitro assay (see Supplementary Table. 1). HEK cells were seeded onto a clear bottom, black 96-well microplate (Corning Inc.) in dMEM/F12 without phenol red (Gibco Inc.) containing 10% Fetal Bovin Serum and incubated for 24 h at 37 °C with 5% $CO_2$. The next day, serum dilutions were prepared (1:5, 1:25, 1:50, and 1:250), added to diluted adeno-associated virus encoding eYFP (AAV-Ef1α-eYFP), 1:1 volume-to-volume ratio, and for 1 h at 37 °C (see Supplementary Table 2 for AAV concentrations). Next, 10 μL of the serum-virus mixture was added to each well of cultured cells, resulting in final serum concentrations or 1:100, 1:500, 1:1000, and 1:5000. Experiments were run in triplicate for each condition. Positive control (no serum) and negative control (no AAV) wells were included in each experiment. For the pre-injection neutralization experiment, two serum dilutions were prepared (1:5 and 1:25), added to diluted adeno-associated virus encoding eYFP (AAV-Ef1α-eYFP), 1:1 volume-to-volume ratio, and for 1 h at 37 °C (see Supplementary Table 2 for AAV concentrations). Next, 10 μL of the serum-virus mixture was added to six wells of cultured cells. A positive, no serum, control was included for each serotype.

Thirty-six or forty-eight hours after incubation, wells were imaged and YFP fluorescence measured. Fluorescence was quantified using a Tecan Infinite M1000 Microplate Reader. For each serum dilution, pre-injection and post-injection sera were compared using unpaired t-tests. In addition, widefield images of each well were captured using a Leica DMi8 microscope (10× objective) to visualize YFP expression. Cell lines were not authenticated.

**Surgical procedures**. In monkey X, we implanted the imaging chamber in a sequence of two surgeries in order to (1) minimize the duration of surgical procedures, and (2) allow for behavioral training with head fixation prior to opening the dura. After the dura was opened, the surface of the cortex required intermittent cleaning and maintenance, and an opaque neomembrane often began to grow over the surface of the cortex approximately 2–4 months after opening the dura. In the first surgery, we implanted the chamber over the surface of the bone, sealing the chamber to bone interface with C&B Metabond dental cement (Parkell), and cementing the chamber in place using Palacose bone cement (Heraeus Medical) and titanium bone screws (Synthes Inc.). Custom-machined headposts were implanted to allow for head fixation during behavioral training. Several months later, after completing behavioral training, we performed a second surgery to remove the skull (in the region of the cylinder) and dura from the center of the 2P chamber, to perform viral injections and to place the artificial dura (AD). Viral constructs were injected using pulled glass micropipettes (~25 μm tips), beveled using a micropipette grinder (Narishige EG-401 pipette beveller) using a nanoliter injector (WPI Inc.). To assist in visualizing viral injections, trypan blue dye (0.4% (w/v)) was diluted 1:10 in saline and mixed with the viral suspension[21].

For monkeys S and W, we implanted an earlier generation of titanium chamber design, which threads into a craniotomy. For these surgeries, the imaging chamber was implanted in a first surgery, but the dura was left intact, and virus injections were performed using stainless steel syringes (Hamilton) outside the operating room while the monkey was performing a behavioral task. After waiting for expression, the dura was then resected, and the AD placed in a second surgery. While this approach has the advantage of not exposing the surface of the brain while waiting for GCaMP expression (~8 weeks) prior to imaging, it required injections to be performed through the dura, preventing careful targeting of virus as the vasculature on the surface of cortex was not visible. This led to additional uncertainty with regards to injection location and depth, made it impossible to avoid surface vasculature while injecting, and increased the required volume of virus injected in order to be confident that a sufficient volume of virus was injected in the target lamina. As such, in monkey X we adopted the more targeted approach, performing virus injections using glass micropipettes with the surface of the brain exposed, which is the preferred approach for future experiments[21].

**Implant design and maintenance**. The imaging chamber design strikes a balance between (1) enabling imaging access to a large volume of tissue (18 mm visible, of which a 12 mm diameter region is accessible for 2P imaging down to ~1000 μm deep using the Nikon 16 × 0.8 NA lens) using commercially available multiphoton objective lenses and (2) minimizing the implant size. The implant must allow for stable head fixation during reaching behaviors, and allow for simple replacement of the silicone AD and access to the edge of the dura for routine cleaning and maintenance. These design constraints suggest a large diameter but low-profile imaging chamber and multi-point head fixation. Design files for the chamber and associated hardware are provided in supplemental materials.

After an experiment, the tissue stabilizer was removed, and a solution of agarose and vancomycin was applied to the dura edge, typically every 1–4 days. Physiosol (Pfizer Inc.) with added vancomycin was applied to fill the remaining chamber volume, and the glass window was secured in place with the retainer ring, and the chamber cap is secured on top. Under normal conditions, the AD did not need to be regularly removed or replaced for cleaning. Over timescales of several months,

we observed "neomembrane" tissue growth under the AD but over the surface of the cortex. Over time, this tissue grew in thickness, blocking optical access to the cortex, and requiring careful surgical dissection and removal. In practice, the timeline for tissue removal could be variable but was usually required every 2–4 months to retain imaging performance. The wound margin at the edge of the durotomy also experienced tissue growth and required periodic trimming to prevent excessive buildup of tissue above the artificial dura.

**Head restraint and implant immobilization**. Initial testing revealed that single-point head fixation methods, as are commonly used for electrophysiology, were incapable of restricting tissue motion at the scale of microns. We tested a three-point acrylic-free footed headpost system that uses bone screws to secure headposts to the skull, and found that this too was not sufficient to restrict micron-scale implant motion during arm-movement behavior. Instead, we found that using a single larger implant constructed with bone cement (Palacose, Zimmer BIOMET Inc.) and titanium mandible straps (Synthes Inc.) fixed to the skull with bone screws allowed for more rigid fixation by distributing the loads between the multiple fixation points through the bone cement. Design files for the head restraint and associated hardware are provided in supplemental materials and details provided upon reasonable request.

**Tissue stabilization**. Prior to imaging, the outer window and retaining ring were removed under sterile conditions, and the AD was exposed to the air. The tissue stabilizer was placed within the chamber, placing gentle mechanical pressure on the top surface of the AD. The tissue stabilizer consisted of a conical aluminum component which sat inside the chamber and held a circular glass coverslip against the artificial dura. Since the location of the surface of the brain could change over time due to tissue growth, recession, or other factors, we fabricated a set of tissue stabilizers at different fixed depths in 500 μm increments. Sterile saline was placed in the chamber prior to placing the stabilizer as an index matching fluid between the silicone and stabilizer glass.

**Widefield Imaging in V1**. As part of the development and verification process for the GCaMP constructs, we validated some constructs using widefield imaging in visual cortical areas in monkey L (Supplementary Fig. 5). We used a large ($6 \times 6$ deg$^2$) sine wave grating at 100% contrast with a spatial frequency of 2 cpd. The mean luminance of the screen was set at 30 cd/m$^2$. The grating was flashed with a temporal frequency of 4 Hz (100 ms on, 150 ms off), while the monkey was performing a fixation task. The behavioral task and widefield (1P) GCaMP data analysis in the rhesus macaque (monkey L) were performed as described previously[21].

**Two-photon imaging in premotor primary motor (M1) cortex**. Imaging was performed using a Bruker Ultima in vivo microscope with a custom motorized orbital nosepiece (Bruker Inc.) to provide off-axis imaging with a Nikon $16 \times 0.8$ NA objective lens. Images were acquired at $512 \times 512$ resolution at a single depth at 30.3 Hz using resonant-scanning galvanometers (or occasionally at lower resolution and higher framerate, while imaging a portion of the $512 \times 512$ pixel field). Laser power was adjusted as necessary to optimize SNR prior to each recording series and typical values were between 50–150 mW. Using dichroic beam splitter (555 nm) and a pair of filters, we collected both a functional green channel (520/44 nm) and a static red fluorescent channel (624/40 nm) to facilitate registration.

Typical sessions in which the subject was working in the imaging rig lasted between 90–180 min. Imaging was performed during decode blocks, typically lasting between 5–30 min, and different decoding sessions were performed in different fields of view during a single session in order to explore different injection sites and depths. We typically performed imaging (while the monkey was at rest and not performing the task) for several minutes in between decode blocks to localize distinct fields of view with neural features, or to localize an imaging field of view based on the surface vasculature. We did not observe photobleaching over the course of a decode block or across blocks within a session in the same region.

2P imaging sessions were collected over sessions spanning the following number of days in the three subjects included here: monkey S, 122 days; monkey W, 30 days; monkey X, 144 days. We note that for all three subjects, degradation of imaging quality was not a driver for terminating experiments, and imaging data were collected until we had sufficient data within the imageable injection regions. In particular, for monkey W, imaging quality remained excellent but as we did not observe functional tuning in the neurons we imaged, we did not continue to collect data beyond the first month of exploration after observing virus expression.

The positioning of the imaging plane for each decode session was chosen to maximize the number of modulated processes observed in the field of view. We did not attempt to optimize correspondence of the imaging field of view across sessions.

**Behavioral task**. Monkey X was trained to make point-to-point reaches with the arm contralateral to imaging implant on a delayed center-out-and-back task in a vertically-oriented 2D plane as described previously[126]. The monkey initiated each trial by holding a point at the center of a display screen, placed approximately 30 cm from the eyes. Next, one of four radially arranged targets appeared 10 cm from the center and jittered around the target location (5–10 mm std. deviation)

during the delay period (randomized period, ranging from 100 to 700 ms). Next, the target ceased to jitter, and the central fixation point disappeared, thereby indicating a "go cue". The monkey was free to initiate a reach following the "go cue". Movement onset was defined as the time when in-plane hand speed exceeded 5% of peak speed for each trial. We note that movement onset is defined entirely behaviorally, independently of neural responses. Reaction time (RT) was defined as the time between the "go cue" and movement onset. To exclude rare trials where the monkey may have anticipated the "go cue" (i.e., RTs < 180 ms) or where the monkey may have been distracted (i.e., RTs > 620 ms), we aborted trials in real time by blanking the screen and withholding reward if the RT was outside of the 180–620 ms RT range. Hand position was measured in 3D and in real time with a Polaris infrared bead tracker (Polaris, Northern Digital, Ontario, Canada) which samples 60 times/s with submillimeter accuracy. Liquid rewards were delivered automatically upon successful target acquisition and hold in the delayed-reach task, or upon successful target decode in the oBCI task, which operated in the same way but rendered a magenta annulus around the decoded target upon successful decode, or gray for unsuccessful decode.

Task timing, stimulus control, and behavior monitoring were performed using Simulink with real-time xPC target (Mathworks Inc., Natick, MA) while microscope control, image acquisition, and online decode were performed by a separate PC (Fig. 6a). Behavioral and task data were serialized and sent via UDP to a data logging PC running custom data logging software available online (https://github.com/djoshea/matudp). Images were acquired continually throughout the duration of the task (not triggered on individual trials) using PrairieView software (Bruker Instruments, Inc.).

**Offline ROI identification and analysis of neuronal responses**. We processed each imaging session offline using Suite2P[53] (https://github.com/MouseLand/suite2p) and analyzed 36 imaging decode sessions collected on 8 days. Imaging datasets were aligned using the red, static fluorescence channel using a rigid coarse alignment step followed by a non-rigid block-wise alignment step. Standard settings for the algorithm were used with the following exceptions to optimize for dendritic ROIs and GCaMP6f (connected = False, tau = 0.7). We identified putative dendritic/axonal ROIs in the datasets as those ROIs with a computed aspect ratio greater than 2 (ratio of long axis to short axis of ROI shape), and manually verified that this selection criterion identified only ROIs that appeared to be neuronal processes.

We next performed a clustering analysis on individual functional ROI responses to check whether automatic ROI detection approach identified distinct neuronal signals rather than signals from a small set of individual neurons split into many separate ROIs. First, we assembled for each ROI a vector of trial-averaged responses aligned to movement onset for each reaching direction. These response vectors were then clustered using DBSCAN, a non-parametric density-based clustering algorithm. DBSCAN which looks for clusters of points (ROIs), which are packed more closely together in feature space (have highly similar response profiles during reaching). DBSCAN can automatically determine the number of clusters present in the data. For all 36 datasets, DBSCAN identified only a single cluster including the vast majority of ROI responses, excluding a small fraction of ROIs, containing noise, that were highly dissimilar to all of the other ROIs and to each other. This indicates that in our imaging datasets, ROI responses form a continuum in the space of peri-movement responses, rather than distinct, separable clusters. The latter would be expected if the signals originated from the extended arbors of a small number of neurons. We also directly verified this non-clustered continuum in response space by using a t-SNE visualization (Supplementary Fig. 8d, e).

We next performed a pairwise correlation analysis to identify pairs of putative dendritic/axonal ROIs which likely originate from the same neuron following the method described in ref. [30]. At the threshold of $\rho \geq 0.8$, 187/4365 ROIs were identified as likely recorded from the same neuron as another ROI in the datasets. At a more conservative threshold of $\rho \geq 0.6$, 828/4365 possible duplicates were detected. All dendritic ROIs were retained for subsequent analysis.

Enhanced mean images were computed by averaging all frames of the registered data tensor, and then computing the inverse hyperbolic sine of the value of each pixel to enhance contrast (only for visualization). $\Delta F/F_0$ was estimated from extracted fluorescence traces using an iterative time-varying baseline estimate following Peters et al. (2014)[115]. Single trial rasters were visualized by sorting ROIs using rastermap (https://github.com/MouseLand/rastermap), an embedding algorithm which groups similar ROIs nearby to reveal structure in the traces. Z-scored responses were computed relative to the mean and standard deviation of fluorescence levels observed during a pre-trial baseline period when the hand was stationary. To assess individual ROI's responsiveness to task events, we extracted on each trial, the fluorescence level of each ROI within a time window just prior to the task event and immediately following the task event, and compared the distributions across trials using a rank-sum test. For the target cue and go cue, we compared frames within the window of −150 to 0 ms prior with frames in the window 0–150 ms after. For movement onset, we compared 450–100 ms prior with 0–350 ms after, as neural responses accompanying movement typically begin in advance of movement persist throughout the reach. We computed the onset time of directionally-tuned responses in each ROI by performing a Kruskal–Wallis nonparametric one-way analysis of variance on the single-trial fluorescence

measurements at each time in a sliding window. The onset time is taken when significance at $p < 0.01$ is reached for five consecutive 10 ms windows.

Direction-tuning was assessed by collecting the single trial average fluorescence levels of each ROI in the 350 ms following movement onset, and grouping these values by reach direction condition. ROIs were considered direction-tuned if an ANOVA test for a main effect of reach-direction significantly affected the mean fluorescence level at $p < 0.01$. For these tuned ROIs, we then constructed a tuning map by regressing each ROI's activity onto 2D hand velocity using an empirically determined 50 ms lag between neural activity and hand velocity (neural leads kinematics). We converted these coefficients into a preferred direction angle, which we mapped into a perceptually uniform circular colormap (CET-C2, Peter Kovesi, https://peterkovesi.com/projects/colourmaps/). Colored ROI masks then were painted onto the mean image using a transparency map derived from the product of the strength of the directional tuning of the entire ROI and the ROI mask weights onto pixels.

To visualize neural trajectories, we first computed the trial-averaged $z$-scored ROI responses for all direction-tuned ROIs in a window 300 ms before to 400 ms after movement onset. We used demixed PCA (https://github.com/machenslab/dPCA) to identify a condition-independent signal (CIS) dimension exhibiting variance over time but with minimal cross-condition variance (corresponding to reach direction). For DPCA, the lambda hyperparameter was determined automatically for each session using cross-validation. To identify a subspace of dimensions exhibiting variance related to X and Y reach direction, we used targeted dimensionality reduction (TDR)[55]. Essentially, we compute the pseudoinverse of the matrix of X and Y velocity regression coefficients for each direction-tuned ROI to obtain these dimensions. We then successively orthonormalize the CIS, TDR X, and TDR Y dimensions. Trial-averaged and single trial ROI trajectories are then projected into these dimensions for visualization.

To assess the stability of tuning individual ROIs within imaging sessions, we separately assessed preferred directions for each ROI for the first 25% and last 25% of each session. We computed the difference of preferred directions and a null distribution computed as the difference of preferred directions computed from 1000 random reshufflings of these early/late imaging frame labels. Drifts of preferred directions was considered significant if the absolute change in preferred direction exceeded that of 99% of the random shuffles ($p < 0.01$). To assess the heterogeneity of tuning within individual ROIs, we regressed each individual pixel within an ROI onto hand velocity, producing a set of pixel-wise regression coefficients. We then performed K-means clustering on these coefficients and identified the optimal cluster count which maximized the Calinski-Harabasz criterion values. We considered an ROI to have multiple meaningful clusters if the optimal cluster count was greater than 1, and if at least 30% of the pixels belonged to at least two such clusters, and the difference in preferred direction corresponding to the centroid of these clusters were at least 60 degrees.

**Real-time oBCI decoding**. During oBCI decoding sessions (36 decode sessions collected on eight days), image data were accessed using PrairieLink (Bruker Instruments, Inc.), processed using a soft real-time C++/Cython/NumPy pipeline, which aggressively avoids memory copies to reduce latency. The code for this real-time decoding pipeline is documented and available at https://github.com/djoshea/obci/. Frames were ignored until a fixed period of time after the go cue ($T_{skip}$), after which frames are integrated for a fixed duration ($T_{int}$) (Fig. 6b and Supplementary Fig. 14). $T_{skip}$ was selected to roughly match the average reaction time of the subject to begin movement, so as to avoid acquiring (for decode purposes) neural activity that is unlikely to be related to the reach target. Then, starting right after $T_{skip}$, we acquired peri-movement neural activity during $T_{int}$. This decoding nomenclature was used in a prior study from our group[8]. While it is possible to develop a mathematically optimal procedure for selecting $T_{skip}$ and $T_{int}$, in practice, we found that it was helpful to use a permissive integration window to capture the natural behavioral variability of an animal, while also minimizing the number of non-informative imaging frames that contribute to the average frame calculation. We experimented with a range of values to improve decode performance and speed (Supplementary Fig. 14): $T_{skip}$ values ranged from 200 to 400 ms and $T_{int}$ values ranged from 70 to 200 ms. These ranges were selected based on several factors, including:

1. Variability in the reaction time of the animal. The integration time should be centered around the beginning of the attempted movement. On trials in which an animal had a slow reaction time, the integration window could occur too early, before a decodable signal was present (in our data). As such, there exists a tradeoff between decoder speed and accuracy, taking behavioral variability into account.
2. The typical timing of divergence of calcium signals within the imaging region, based on offline analysis, as shown in Fig. 6h, i. We note that this timing may be specific to the recording location and subpopulation of cells recorded.
3. Image acquisition time (33 ms). Only full frames are considered during decode, so values of $T_{int}$ and $T_{skip}$ in practice are discretized by the frame period.

Rather than create a multi-objective cost function to optimize these values, in practice, we chose to bias towards increased robustness by using a permissive $T_{int}$

(long window) and short $T_{skip}$ to increase the likelihood of movement onset and peri-movement neural activity falling within the integration window.

Each imaging frame was processed by applying an approximate Gaussian spatial blur (standard deviation of between 3 pixels) and accumulated into a running average (Fig. 6c). During training trials, after $T_{int}$, the average image was accumulated into the running average for the current target, $Y_{i,j,k}$, where $k \in 1, 2, 3, 4$ indexes target location, and $i, j$ index pixel $[x, y]$ location. During decode trials, after $T_{int}$, the reach target was decoded using a pixel-wise minimum mean square error (MMSE) decoder as described in Eq. (1):

$$\underset{k}{\arg\min} \sum_{i,j} \left( X_{i,j} - Y_{i,j,k} \right)^2, \qquad (1)$$

$X$ is the image formed by averaging frames acquired during the $T_{int}$ on the current trial. oBCI experimental sessions were often terminated to explore different imaging depths or fields of view the same day, as opposed to ending a session after observing roll-off of decode performance. In many if not most cases, we anticipate that the image decoder would continue to operate successfully for much longer than the acquired series of data.

For online decode experiments, we sought to ensure that motion artifacts were not responsible for driving tuning and decode via several approaches:

1. In post-hoc analysis of the data obtained during online decode experiments, we did not observe characteristic patterns of symmetric signal increase/decrease for opposite movements that we might expect if fluorescence modulation were primarily resulting from motion. In virtually every field of view imaged, only a sparse subset of neural features exhibited tuning, while the other fluorescent objects with a similar morphology did not. It would be quite unlikely, if not impossible, to obtain this result if the observed tuning were the result of motion artifacts.
2. We performed online control experiments, imaging in areas with autofluorescence but no GCaMP expression and did not decode at above chance levels.
3. For online decode experiments, we lightly blurred the images with gaussian smoothing to minimize the impact of X/Y translational movement.
4. While Z-axis motion is not directly observable, during initial experimental characterization, we imaged dendrites during behavior at several focal planes, offset in depth in sequences of 5–10 μm. In all cases, we observed consistent tuning regardless of imaging plane position, further indicating that the observed tuning was not the result of z-axis motion artifact (data not shown).

We minimized the impact of background light via several methods:

1. The monkey and apparatus were in the dark throughout experimental sessions, with the only source of light being the illumination of targets displayed on the screen. Targets were either red or blue (not green) to minimize the likelihood of interfering with GCaMP signals, and LEDs or light sources were taped over.
2. The geometry of the tissue stabilizer naturally blocks most residual background light due to the small space between the lens and the wall of the tissue stabilizer.
3. If background light was ever found to be an issue, we placed blacked foil around the implant and objective.

In practice, we found that background light did not present a major challenge during the course of these experiments.

**Offline analysis of decoder performance**. Offline classification was performed using scikit-learn[127] to implement multiclass support vector machine classification with 5-fold cross validation. Trials were filtered for successful reaches and aligned to either go cue or movement onset for separate analyses. Images were processed to calculate $\Delta F/F$ and integrated using a causal sliding 200 ms window, and all frames within this window were averaged prior to inclusion and preprocessed by removing the mean and scaling to unit variance for each feature (pixel). Imaging data was registered using NoRMCorre[128] prior to analysis. Offline analysis was performed using custom MATLAB (Mathworks Inc, Natick, MA) and Python code, available upon reasonable request.

For the eigenvalue screen plots presented in Supplementary Fig. 12, eigenvalues were calculated by running PCA on a 2d matrix formed by concatenating all trials and time points along dimension 1, and unraveling all pixels from single frames along the second dimension. The final matrix has dimensions (# trials * # timepoints) x (# xPixels * # yPixels). The offline pixel dropping analysis presented in Supplementary Fig. 15 was performed by selecting 15 square regions of a given size, sampled at random from the full field of view and masking all pixels outside of that square. The masked data was then decoded using the same approach described above for multiclass classification.

**CLARITY tissue clearing and imaging**. The area of interest was dissected from the whole brain and embedded in 1% hydrogel. The tissue was cleared in the SmartClear (Lifecanvas Inc.) for 2 weeks, then stained with anti-GFP conjugated to alexa-647 (Invitrogen Inc.) and lectin conjugated to DyLight 488 (antibody concentrations 1:100 in PBST with 1% triton-x and 0.2% sodium azide) for 1 week, and finally washed in PBST for 1 week prior to imaging. The tissue was immersed in

RapiClear (SunJin Lab Co) for 2 days and then imaged using an Olympus two-photon microscope with 10 × 0.6 NA CLARITY objective.

**Open-source software**. We thank the authors and contributors of the following open-source projects utilized in this project:

- Cython—https://cython.org/
- Numba—http://numba.pydata.org/
- SciPy[130]/NumPy/Matplotlib—https://www.scipy.org/    https://numpy.org/ https://matplotlib.org/
- Jupyter—https://jupyter.org/
- Sci-kit learn[127]—https://scikit-learn.org/
- Suite2P[53]—https://github.com/cortex-lab/Suite2P
- Rastermap—https://github.com/MouseLand/rastermap
- dPCA[54]—https://github.com/machenslab/dPCA
- CET Perceptually Uniform Color Maps[131]—https://peterkovesi.com/projects/colourmaps/
- Libuv—http://libuv.org/
- Dear ImGui—https://github.com/ocornut/imgui
- Cygwin—https://www.cygwin.com/

**Reporting summary**. Further information on research design is available in the Nature Research Reporting Summary linked to this article.

## Data availability

The datasets generated during and/or analyzed during the current study have been made available on DRYAD at https://doi.org/10.5061/dryad.cnp5hqc4k. Supplemental figure data are provided in the accompanying excel file: "Supplemental Data 1.xls"

## Code availability

Offline analysis was performed using custom MATLAB (Mathworks Inc., Natick, MA) code, available upon reasonable request. The code for the low-latency data access and real-time decoder is available at https://github.com/djoshea/obci and have been published via Zenodo at https://doi.org/10.5281/zenodo.4702559[129].

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

## Acknowledgements

We thank M. Risch, M. Wechsler, L. Yates, and R. Reeder for surgical assistance and veterinary care and B. Davis for administrative support. This work was supported by NIH NRSA grant 1F31NS089376-01 (E.M.T.), a Stanford Graduate Fellowship (E.M.T.), NSF IGERT grant 0734683 (E.M.T.), DARPA BTO "NeuroFAST" award W911NF-14-2-0013 (K.D. and K.V.S.), the Howard Hughes Medical Institute (K.D. and K.V.S.), an NIH Director's Pioneer Award 8DP1HD075623, and the Simons Foundation Collaboration on the Global Brain awards 325380 and 543045 (K.V.S.). We thank John Rafter, Aaron Statz, and Michael J. Fox at Bruker Corporation for their assistance with the 2P imaging microscope and guidance in achieving real-time access to microscope data.

## Author contributions

E.M.T. and D.J.O. designed and constructed the 2P imaging system and designed the implant. E.M.T., D.J.O., and X.S. performed virus injections, designed and conducted the in vivo imaging experiments. D.J.O. implemented the real-time decoding system. E.M.T., D.J.O., and G.B. performed analysis of 2P imaging. S.I.R., E.M.T., and D.J.O. performed the surgeries. J.H.M. provided extensive assistance with all aspects of imaging, experimental design, analysis, and manuscript preparation. A.C. performed CLARITY imaging. B.H. performed histology and CLARITY sample preparation. S.V. performed neutralizing antibody assay and assisted with manuscript revision. L.C. assisted with implant and equipment design and testing. G.B. developed 2P imaging analysis code. W.A. and I.K. built the widefield imaging microscope and provided significant assistance with 2P imaging. S.Q. assisted with 2P imaging. M.M. performed surgery and assistance with virus injections in Monkey X. Y.C., M.W., and E.S. assisted with implant design and performed widefield imaging and GCaMP validation in Monkey L. C.R. assisted selecting and validating viral constructs. M.S. assisted with 2P analysis. K.D. and K.V.S. provided guidance and assistance with all aspects of the work. E.M.T., D.J.O., X.S., and S.V. wrote the manuscript with input and editing from all authors.

## Competing interests

K.V.S. is a consultant to Neuralink Corp. and CTRL-Labs Inc. (now a part of the Facebook Reality Labs division of Facebook) and on the Scientific Advisory Boards of Mind-X Inc., Inscopix Inc. and Heal Inc. K.D. is on the scientific advisory board of Maplight Therapeutics. These entities did not support this work. Following this study, J.H.M. is now a member of the scientific advisory board of Bruker. M.M. is employed by Neuralink Corp. The remainder of the authors declare no competing interests (E.M.T, D.J.O, X.S, G.B., S.I.R, A.C., B.H., S.V., L.C., W.A, I.K., S.Q., M.M., Y.C., M.W. E.S., C.R., and M.S.).
