## [Peer Review File · Nature Communications]

Reviewer #1 (Remarks to the Author):

The paper by Trautmann, O'Shea, Sun et al. presents impressive technical advances to provide the first demonstration of an all optical BCI in macaque monkeys. I found the paper interesting and well written. The technical advances are undoubtedly novel and important –they provide a proof of concept of powerful tools in macaques. However, I do have several comments; most importantly, I'd like the authors to present more "raw data" and a more in-depth analysis of their decoder (see my comments 5 and 6 below).

Comments:

1. Maybe I missed this in the text, but was the chamber designed so the three-point fixation always locked in the same exact position? I imagine so, because the authors show a few examples of their ability to record from the same ROIs on different days, but I'd welcome a bit more detail on this. Perhaps it could be useful to add some example CAD drawings and photos in the Supplementary Figures (I couldn't find them in this version). Moreover, I'd like to see a more detailed quantification of the data to support this statement (see the additional analyses I suggest in #5 below)
2. "As anticipated, for activity in M1, as opposed to in PMd, we did not observe sufficient preparatory activity in this region to facilitate decoding the upcoming reach target (...)" -> In a recent study by Dekleva et al. they report that "On a single-trial level, we saw no substantial differences between the direction of the reach plan decoded separately from PMd and M1." Thus, if the authors were not able to decode the upcoming movement from M1 signals, it is possible that the movement-related information in their recordings is significantly worse than in M1 neural population firings recorded with standard Utah arrays (a fact they already recognize in the discussion). Please, update the manuscript accordingly
3. The values of Tint and Tskip should be indicated in the Results —the authors only cite a previous paper in the Methods. Were they identified on a session-by-session basis during decoder training? If so, how? It would also be useful to see these intervals graphically, perhaps in Fig 4K or in the new figure I suggest the authors to make in #5 below.
4. I may be wrong, but there seems to be a trend toward decoder performance degrading during a session (Fig 5E) (it would be helpful if the authors added the mean +/- SEM accuracy to these traces). Is that case? If it is indeed happening, do the authors think it is due to the progressive drift of the imaged ROIs? In relation with this, what was the total duration of one experimental session?
5. I'd like to have a better sense of the data: a new section and figure on it before or after the decoding results would really strengthen the paper. I'd like to see things like:
 - the number of ROIs identified and used for decoding for each monkey across days (in case they don't match)
 - how homogeneous is the tuning within each identified ROI?
 - how is the information distributed across layers (more superficial processes vs somata in superficial layers)?
 - how stable are the identified ROIs during a session (e.g., based on the correlation of the activity in the imaged ROIs between the first half and the second half of trials after matching targets; a few

examples of mean image for the first 25 % of the trials and the last 25 % of the trials would be useful too)

- a more detailed quantification of how well the ROIs match across days
- raster versions of Fig. 4K for all the ROIs across a few example sessions (e.g., mean delta F/F traces for each target and ROI), etc.

6. I'd also like to see a more thorough offline analysis of their decoder experiments. For example: how is the information distributed across ROIs? And across layers? How does decoder performance degrade as ROIs are dropped? What would be the cross-validated performance of a decoder build on single trial data compared to their approach? And even, how does the eigenvalue distribution of the DeltaF/F look? (this could help have a crude sense of the "dimensionality")

7. I'm obviously impressed by the authors' identification of a Betz cell. Yet, I wonder: could the authors replicate this result and identify more neurons? Their result is great, but it's an $n = 1$... and besides this result is quite prominently features in the abstract. Note that I'm not an expert in CLARITY so I don't have a good sense of whether this is feasible.

8. I'm curious as to whether the authors tested any of the calcium imaging processing pipelines that have become standard in the field like Suite2p (Pachitariu et al 2016) or CNMF (Pnexamatikakis et al 2016) on their data, rather than developing their own. Also, did they compare their online processing to offline "sorting" or is this not necessary because the online implementation didn't require any algorithmic "shortcuts"?

9. As I said above, I like how the paper is written. However, in some parts I had the impression that the authors were trying to spin a very impressive methods paper into a more scientific paper, whereas the overall tone is focused on the methodological advances. I suggest they take this latter view.

Minor:

- I am curious as to whether the
- Supplementary Figures are not cited in order
- Line 6: "...from in..." is a mistake
- Line 29: "arm and hand movements". Understanding hand motor control will be critical for FES- or spinal cord stimulation-based neuroprosthetics. With regards to neuroprosthetics, I thought the recent study by the Courtine group showing restoration of walking after SCI with BCI-controlled spinal stimulation was missing (Capogrosso et al 2016)
- Line 32: perhaps cite some tools to measure large neural populations?
- Line 42: while I appreciate the potential advantage of recording from different brain areas by just moving the objective lens, doing so also requires having a larger imaging window, with the additional complications and risks this poses. Despite the impressive advances shown by the authors (e.g., 18 mm window), I think this should be discussed
- Line 146: "comment" should be "common"
- Fig 2A,B,G: make the text larger. Same for Fig. 3H-K
- Fig 4H: is the colored circle on the top left a legend that indicates the directional tuning of the ROI?
- Fig 4M: Make larger; it's hard to appreciate in the current version.

- Fig 5G: aren't the "incorrect" and "correct labels of the top figure swapped?
- Line 308: As shown in Fig 5 -> Fig 6?
- Fig 7B,D: Did the authors use yellow because the neural feature of interest was tuned to the yellow direction in A? I'm intrigued as to whether a part of the neural process is tuned to the yellow direction and the other to the nearby green direction?
- Lines 339-340: this needs further quantification; my proposed new figure (Comment 5) would help.
- Lines 459: the authors may find useful a recent preprint by Gallego, Perich et al, where they quantify the large changes in movement tuning parameters of Utah array recordings over up to two years, and show how they can stabilize a decoder using an idea similar to that in Trautmann et al 2019 despite these changes. The same group has a follow-up publication using deep learning methods to accomplish a similar goal (Farshchian et al 2018). Moreover, the Batista, Chase & Yu groups are working on a similar approach, but I do not believe it is available yet other than as SfN abstract. I also noticed a paper from the Shenoy group and another by Wu & Hatsopoulos that I think could be worth citing (Chestek et al 2011; Wu & Hatsopoulos 2008).

References:

- M Capogrosso et al. A brain–spine interface alleviating gait deficits after spinal cord injury in primates. *Nature* 2016
- C Chestek et al. Long-term stability of neural prosthetic control signals from silicon cortical arrays in rhesus macaque motor cortex. *J Neural Eng* 2011
- BM Dekleva, KP Kording, LE Miller. Single reach plans in dorsal premotor cortex during a two-target task. *Nature Communications* 2018
- A Farshchian, et al. Adversarial Domain Adaptation for Stable Brain-Machine Interfaces. arXiv:1810.00045
- JA Gallego, MG Perich, RH Chowdhury, SA Solla, LE Miller. A stable, long-term cortical signature underlying consistent behavior. *bioRxiv* 2018
- M Pachitariu, C Stringer, M Dipoppa, S Schröder, LF Rossi, H Dalglish, M Carandini, KD Harris. Suite2p: beyond 10,000 neurons with standard two-photon microscopy. *bioRxiv* 2017
- EA Pnevmatikakis et al. Simultaneous Denoising, Deconvolution, and Demixing of Calcium Imaging Data. *Neuron* 2016
- W Wu, NG Hatsopoulos. Real-time decoding of nonstationary neural activity in motor cortex. *IEEE Trans. Neural Syst. Rehabil. Eng.* 2008

Reviewer #2 (Remarks to the Author):

Trautmann et al. used two-photon (2P) imaging of the rhesus macaque premotor and primary motor cortices to detect calcium signals from dendrites in the superficial layer that displayed stable tuning for different directions of arm movement across several weeks. In addition, they developed an optical brain-computer interface (oBCI) with these calcium signals to successfully decode the movement direction online. After imaging, the authors treated the imaged tissue with CLARITY, which showed that one of the imaged dendrites originated from a deep layer 5 neuron. This is the first demonstration of 2p calcium imaging in the macaque motor cortex and online decoding of movement direction. This method has some advantages over electrical recording because 2p imaging through a large cranial window can reveal the activity of large populations of densely localized neurons over several weeks and is not invasive. Also, it can be used to study neural

population activity, which will be critical for the design of next-generation BCIs. The current study provides an important advance in the field of motor control. I recommend the paper be published in Nature Communications if the following concerns are properly addressed.

1) It is crucial to show clearly the reliability of 2p calcium imaging. How many ROIs morphologically corresponded to dendritic branches in each field of view? Did some ROIs originate from the same neuron and show highly correlated spontaneous activities? Although the authors indicated that they could access neurons in L2/3 and showed an L2/3 cell body image in Figure 4E, F, they did not show these activities. The authors should show some examples of raw fluorescence traces of the dendrites and cell bodies over several minutes. What proportion of ROIs showed movement-related activity and direction tuning? Was there any neuron that responded to the go-cue presentation or the reward? Figure 4M and Supplementary Figure S4 elegantly showed that a subset of dendrites acquired tuning for the movement direction that remained stable over several days. However, it is not clear whether these dendrites showed stable tuning over all imaging days or over some imaging days. It is possible that the tuning of other dendrites was unstable, i.e., their tuning changed or even disappeared on some imaging days. Please show changes in the proportions of the task-related neurons over several days.

2) As the authors repeatedly pointed out, the major advantage of 2p imaging is the ability to identify a population of recorded neurons. However, the fluorescent changes in the whole field of view were used to drive the oBCI decoder. Thus, it is critical to show how many ROIs and different neurons were included in each oBMI decoding and the dependency of decoder accuracy on the number of neurons. It is also important to show that the field pixels, except for those of dendrites, could not decode the direction. These analyses could be done offline.

3) How many ROIs with calcium transients were reconstructed by CLARITY? Could any ROI that originated from the superficial layer neuron not be identified? How many cell bodies were found in layers 2/3 and 5? Although 3D reconstruction of neuronal morphology is improving, the limitations of the current method and how these could be solved should be discussed.

Others:

Because corticospinal neurons show high spontaneous activity, it would be possible to distinguish differences in spontaneous activity between dendritic branches. The authors should determine whether reaching direction preference was different between different types of neurons.

In line 217. The authors should show the image and fluorescent signals of neurons at a depth of around 550 μm from the cortical surface.

The numbers and periods of imaging sessions were not clear. For example, "multiple times" over "several" experimental sessions in line 238 should be replaced with concrete figures.

The authors mentioned that M1 neurons did not show preparatory activity in line 258. It would be better to show calcium transients in both PM and M1 neurons during the preparatory period.

The authors stated "Behavioral sessions lasted between 90-180 minutes and no bleaching of GCaMP6f was observed over time" (in line 573). Was imaging performed between 90-180 min and were all of the image data used?

In Figure 4J, K, the range of the go cue onset should be added. In Figure 4K, please show the amplitude of the Y axis, and show these ROI structures in the field of view.

Figure 5G is a very important panel showing the time course of decoder accuracy. However, the explanation is unclear and requires further clarification. Is time 0 the movement onset or the onset of Tint? Was the Y-axial value at each time point calculated from each frame or from accumulated frames?

Neutralizing antibody assay to seek the AAV that would not induce the immune response in each animal may be important. However, in the current study, the effectiveness of AAV selection based on this assay was not demonstrated. Was non-standard GCaMP expression related to specific combinations of AAV and marmoset? The authors should tone down the description regarding the neutralizing antibody assay.

In line 468, the authors should also cite Masamizu et al. Nat Neurosci (2014).

Recently, Shiming Tang's group in Peking University reported 2p calcium imaging of a large population of neurons in the macaque visual cortex. A discussion of the differences between the imaging techniques used in the current study and those used in Tang's study should be added because many readers might find this information useful.

The laser power should be provided.

How was light shielding performed around the objective?

How was the motion artifact corrected? When did the artifact occur during the movement? The width of a dendrite is less than 2 μm , so even a few μm shift in the dendrite shown in Supplementary Figure S3 would distort the calculation of neural activity. What about the shift along the Z axis?

Reviewer #3 (Remarks to the Author):

Trautmann, O'Shea, Sun and colleagues show a non-human primate calcium imaging system for optical brain-computer interfaces. The abstract emphasizes key, primarily technical, achievements: implant system for (a) chronic 2p imaging access and (b) motion-stabilized 2p access; (c) ability to image activity of deeper neurons in layer 5 via apical dendrites visible from the surface, (d) functional tuning of apical dendrites that can be used to decode arm movements, and (e) online decoding pipeline for images. Overall the work represents impressive technical contributions, though I feel the manuscript needs significant revision to more precisely quantify their results to more precisely support key claims/contributions, and to focus and clarify the presentation.

1. My primary concern is that, given the manuscript's focus on technical achievements/contributions, it is severely lacking in analyses to quantify claims and overall results. Some specific examples:

A: image stability claim (via implant/head-fixation design): The only quantification performed

(supplemental fig S3) is for a single example video snippet of unknown length. To claim stability, please provide more documentation of alignment error over time (e.g. a distribution of alignment shifts for multiple days)

B: Registration with anatomical landmarks and across datasets: The manuscript repeatedly makes statements about registration of data across days and different types (e.g. “These vascular features provide reliable landmarks allowing for imaging localization and reliable identification of individual neurons across multiple imaging sessions. These landmarks facilitated registration with ex vivo CLARITY tissue clearing”) without any quantification of registration accuracy. Specific claims of “within a few tens of microns” for CLARITY registration is made without any evidence to support this. Most importantly, the manuscript includes no methodological detail on how these registrations were performed.

C: More thorough quantification of neural signals: The authors show examples of movement-tuned ROIs but do not include any information on the total # of ROIs measured, how many were tuned, etc. Given that they report (but don’t show) many cells with filled nuclei that did not have behaviorally-modulated fluorescence, these details seem particularly important. The intro and discussion also repeatedly emphasize how useful Ca imaging is for recording large populations, but the data shown do not speak to this point at all.

D: Chronic access claim: The claim of chronic imaging is supported by a single example neuron image at 3 time-points, without any quantification (e.g. statistical tests). This is quite minimal support for this claim. I would also recommend more precisely stating these claims to specify what is meant by “chronic” (# of days, ability to image vs. ability to image the same neurons, etc.)

2. Substantial portions of the manuscript are devoted to documenting varying viral injections and immunoreactivity characterization of subjects. These data are interesting and valuable to the community. But, as currently presented the authors do not provide sufficient data to prove particular points or make any claims. There are a wide variety of viral injections (AAVs, promoters) used and general statements about what seemed to work vs. not, but do not directly support strong claims or interpretation. I would suggest the authors shift these data into supplementary materials and discussion or revise the analyses to more quantitatively analyze the phenomena they qualitatively describe (e.g. differences in expression across constructs; progression of expression/nuclear filling).

I want to emphasize that I think these data are very valuable for the community, as they contribute to ongoing understanding of how to optimize viral delivery of opsins and fluorescent reporters in primates. But including these data as key focus of the manuscript when they cannot directly support rigorous conclusions has a diluting effect. In some places the authors also appear to make strong claims that are not well-supported by data about the safety and efficacy of their expression protocols (e.g. achieving “robust functional expression...aided by designing a macaque codon-optimized GCaMP which included a nuclear export signal”. This claim is particularly confusing, as the reaching data is stated to primarily come from an injection site that does not use the NES construct. The main data shown using the NES construct, as I understand it, is from supplemental wide-field imaging data in fig. S2).

3. The paper emphasizes the achievement of closed-loop BCIs with optical signals. However this is both not fundamentally novel (has been done in rodent models by several groups) and the presented experiment is not a truly compelling example of a closed-loop BCI. Given that other online imaging read-out systems have been implemented before, the authors should take effort to highlight any differences/strengths of their approach in order to be able to emphasize this as a key technical contribution. More generally, the latency and significant temporal binning used in the 'online' experiment overall serve to shift the experiment into a discrete read-out that runs in parallel to the animal performing the natural reaching behavioral task, more than an online, closed-loop BCI task. Thus, while certainly a technical feat, it is unclear to me what specifically this experiment demonstrates above and beyond the offline decoding analyses. The authors also state that their work provides "the ability to make closed-loop experiment adjustments based on real-time neural read-outs", but I do not see specific evidence of this in the existing experiment.

4. Overall, I think the paper could benefit from an effort to make it clear what exactly was done across all the animals and for specific data presented. Several things leave me a bit confused. For instance, methods state that immunological assays were performed on 5 monkeys, but then data is only shown for 3. It seems that reaching data and BCI experiments are performed primarily with monkey X? Are other data beyond immunological response included for monkeys S and W?

Minor/specific comments:

Error in abstract: "...imaging of calcium signals from in macaques..."

I would suggest making the statement that rhesus macaques have been important for clinical BCIs (intro 1st paragraph) stronger. Motor BCIs are an example where techniques developed/demonstrated in NHPs directly translated to humans multiple times.

The logic of the argument in the introduction's second paragraph is not entirely clear to me. The authors argue that optical techniques, though not necessarily clinically translatable, will better inform our understanding of arm movement. This seems like a motivation for studies of arm movement representations via optical imaging in NHPs. But the authors then say they present an optical BCI.

Please clarify the imaging area and implant size. Sizes of 1.2cm and 1.8cm are cited in different places in the manuscript.

I am unclear on what specifically the role of supplemental fig. S2 is, particularly because these injections use a different construct (AAV1-CaMKIIa-GCaMP6f vs. AAV-CaMKIIa-NES-GCaMP6f) than the site selected for data from monkey X

Please explain what motivated the specific focus on a single injection site for data from monkey X

Please specify the time-point (relative to implant date) for fig. 4a.

I do not understand the value added by the analyses for fig. 4L.

How many closed-loop BCI sessions are presented for data used in fig. 5?

While the registration with CLARITY is quite exciting and opens new experiments outlined in the discussion, it is important to acknowledge that this technique specifically requires sacrificing the animal and therefore both increases the number of animals needed and is limited to a single time-point. These limitations are not acknowledged.

It is a bit strange to me that the low levels of expression achieved, compared to other animal models is not discussed or even alluded to until the penultimate paragraph of the paper.

Behavioral task: please clarify how the target was jittered during the delay (i.e. hopping between the different possible targets? Vs. position jitter around the target location?)

Some relevant citations appear to be missing:

- Artificial duras designed for NHP: Chen et al., *J Neurosci methods* 2002
- Other techniques that achieve highly stable 2p imaging in NHP: Choi et al., *IEEE EMBS* 2018

Response to reviewers

Introduction

We thank the reviewers for their time and feedback and appreciate their insightful comments and suggestions for how to better support the conclusions presented in this work. Taken together, the reviewers' comments primarily focus on several broad areas which we have worked hard to clarify. These are as follows:

- 1) Showing more raw data and highlighting the limitations of GCaMP expression quality in the reported data. In addition, focusing the manuscript on the technical advances and potential for the method, as opposed to emphasizing the scientific results.
- 2) Providing additional quantification throughout the entire manuscript, including:
 - a) Further quantification of cells and projections in CLARITY dataset
 - b) Control experiments and analysis for our decoding results, including online negative controls and offline pixel-dropping experiments.
 - c) Further analysis and quantification of motion artifact across datasets
 - d) Further reporting of number of neurons, datasets, time ranges, injections, and monkeys.
- 3) Significantly reworking the analysis of functional responses using a standard calcium imaging pipeline (Suite2P) and including a systematic quantification of ROI responses.
- 4) Providing additional details on the methods, including design files for hardware, and code for analysis and online decode software. While we already committed to sharing tools and data, we are now explicitly sharing:
 - a) CAD models for implant, head stabilization, and custom tooling
 - b) Code for low-latency microscope control, data access, and online decoding
 - c) Imaging data upon request.
- 5) Further clarifying where CLARITY results serve (a) to demonstrate the specific point that apical dendrites were imaged versus (b) a proof of concept demonstration that a small number of specific dendrites could be localized and reconstructed back to their somas.

We believe that we have kept these five major areas squarely in mind when revising the manuscript and when describing these changes, and our rationale, in the point by point responses directly below.

Point by point responses

Key:

1. Reviewer comments are highlighted in blue,
2. Author responses are in black.
0. *Text included in the revised manuscript is italicized and indented*

Reviewer #1:

The paper by Trautmann, O'Shea, Sun et al. presents impressive technical advances to provide the first demonstration of an all optical BCI in macaque monkeys. I found the paper interesting and well written. The technical advances are undoubtedly novel and important – they provide a proof of concept of powerful tools in macaques. However, I do have several comments; most importantly, I'd like the authors to present more “raw data” and a more in-depth analysis of their decoder (see my comments 5 and 6 below).

We thank the reviewer for the kind summary and helpful suggestions. As described below, we have performed numerous new analyses and created several new figures. We believe these better quantify the activity exhibited by ROIs in the raw data, as well as characterize offline analyses of the decoding algorithm.

Comments:

1. Maybe I missed this in the text, but was the chamber designed so the three-point fixation always locked in the same exact position? I imagine so, because the authors show a few examples of their ability to record from the same ROIs on different days, but I'd welcome a bit more detail on this. Perhaps it could be useful to add some example CAD drawings and photos in the Supplementary Figures (I couldn't find them in this version). Moreover, I'd like to see a more detailed quantification of the data to support this statement (see the additional analyses I suggest in #5 below)

We thank the reviewer for highlighting this important detail which was not sufficiently clear in the original text. As the reviewer noted, the halo head fixation system position was consistent from day to day. While we believe that this positioning accuracy would be sufficient for localizing an imaging field of view across sessions, the Bruker 2P microscope cannot store absolute coordinates for its micrometer stages, requiring us to recalibrate the microscope position each session. We performed this position calibration using vascular fiducials which served as consistent landmarks within the imaging chamber. On a given session, we would identify consistent large vasculature features (millimeter scale), then follow these to a specific vascular branch point, where we zeroed the stage coordinates of the microscope. From there, it was possible to navigate using stage coordinates to injection sites based on coordinates alone, though we were also able to use small capillary features to ensure that we were

targeting a specific field of view. We have attempted to make this clearer in the text and in the revised Fig. 4, the relevant panels of which are copied below:

Using vascular features identified via widefield imaging as fiducial landmarks, we located the sites of GCaMP expression with 2P imaging and observed neurons expressing GCaMP (Fig. 4E-G). These vascular features provided reliable landmarks across sessions, allowing for relative localization of the imaging field of view within the imaging chamber and subsequent identification of individual neurons across multiple imaging sessions. These landmarks also facilitated identification of neurons in ex vivo imaging using CLARITY (Fig. 4D and shown later in Figs. 7, 8).

Fig. 4: Multiscale, multi-modal imaging. (A) Imaging chamber with stabilizer in place under ambient illumination. (B) Cortical surface imaged using widefield (1P) imaging. (C) zoomed in region highlighted in yellow box in B. Vascular landmark used to calibrate microscope stage positions indicated with blue arrow. (D) Further zoomed widefield image showing microvascular features used for localizing 2P FOVs and aligning 2P imaging with CLARITY marked with magenta arrows. (E) 2P image acquired from the region highlighted in (D) showing consistent landmarks. (F) CLARITY volume from the same FOV in D and E using anti-GCaMP antibody labelling (vasculature in white).

As a side note, we feel that it is important to highlight a distinction between measurement precision and accuracy in this context. The macaque brain is somewhat mobile within the skull

and can translate from day to day, such that micron-scale positioning *accuracy* in the experimental setup cannot effectively ensure micron-scale *precision* when localizing neural features across sessions. While our system did afford reasonable levels of positional repeatability, we instead relied upon our ability to identify static constellations of vasculature (and the same neural features within) to localize imaging fields of view.

In practice, we found that this approach was straightforward, and navigating throughout the addressable volume of cortex to find a specific field of view was not a challenging task. This method enables alignment of imaging fields of view across sessions at micron scale, where the same neural features are unambiguously identifiable through those multiple sessions. This method, however, does not permit a direct measurement of the absolute positioning of the halo system across sessions and we do not have a ground-truth measurement for day to day positioning error.

Halo Design

We appreciate the reviewer's point that additional detail regarding the Halo design would be beneficial. We have added additional detail to Supplementary Fig. 1, copied below. We have also included a repository of CAD files for the Halo components and assembly.

Supplementary Fig. 1 Halo and imaging stabilization. (A, B) Stable imaging during motor behaviors requires stabilizing the implant with respect to the optical table. We developed a custom three-point head fixation system which uses sockets to engage three stainless steel balls on posts that are embedded in bone cement in the implant. The three sockets (red) are threaded and are driven towards the implant to secure

the implant in place using a hex socket on the back end of each post. (C) The halo is fixed to the optical table via a rigid metal riser plate, consisting of a 3" aluminum plate. A 2" aluminum plate provides additional support to one side of the halo. (D) Implant for monkey X mounted in the Halo with the tissue stabilizer placed in the implant in the imaging configuration.

2. "As anticipated, for activity in M1, as opposed to in PMd, we did not observe sufficient preparatory activity in this region to facilitate decoding the upcoming reach target (...)" -> In a recent study by Dekleva et al. they report that "On a single-trial level, we saw no substantial differences between the direction of the reach plan decoded separately from PMd and M1." Thus, if the authors were not able to decode the upcoming movement from M1 signals, it is possible that the movement-related information in their recordings is significantly worse than in M1 neural population firings recorded with standard Utah arrays (a fact they already recognize in the discussion). Please, update the manuscript accordingly

We thank the reviewer for highlighting this important point. We do indeed find delay period activity when recording with Utah arrays implanted in gyral M1, and not only when recording in PMd. In general, in other experiments not reported here, we tend to observe a lower proportion of cells with delay activity in M1 than in PMd, though we are not aware of any rigorous quantification of number of delay-active cells distributed along the rostral-caudal axis moving from PMd down to sulcal M1. As the reviewer noted, however, there are often still enough delay-modulated neurons to decode reach from pre-movement activity when recording from Utah arrays in gyral M1. The lack of delay period tuning observed in these experiments could have resulted from several factors. One possibility might be that this is a consequence of nonlinearities in the GCaMP signal, where weaker delay period activity might be harder to detect than stronger peri-movement activity. A related possibility might be that calcium concentrations recorded in apical dendrites might less faithfully represent the delay period spiking activity relative to the soma, though we have no evidence for this speculation. In general, we agree with the reviewer that delay period preparatory signals are significantly reduced in our calcium imaging datasets than in array recordings.

It is also possible that array recordings preferentially target a different subset of cells than imaging, perhaps owing to a bias for larger, more active neurons to show up extracellularly. Or perhaps due to some unknown genetic or physiological factor that results in better SNR of GCaMP in cells with certain kinds of responses (e.g. low spontaneous firing rates). Some of these cells could be corticospinal projection cells, which serve as a major output of M1 and project to the spinal cord and muscles. It is natural to expect that these cells would be more likely to contribute to output potent dimensions of neural activity [Kaufman et al. 2014]. While Kaufman and colleagues found that the output null and output potent dimensions of neural activity did not map directly onto distinct subpopulations of neurons in their recordings, it is not unexpected that we might observe predominantly output-potent activity when preferentially recording from CST projecting cells. Certainly understanding the circuit-level organization of M1 dynamics is a rich area for future research, which helps to motivate the use of imaging methods in future studies to better understand the role of different cell types in motor preparation and control.

We have added the following note to the manuscript to touch on this discussion:

Superficial neural processes which expressed GCaMP exhibited robust, direction-tuned responses during reaching movements. In contrast, responses during the instructed delay period preceding movement were surprisingly weak in contrast to preparatory responses observed with electrode arrays (Dekleva et al., 2018). This could potentially reflect a less faithful representation of motor preparation in the dendrites relative to the cell body, but it might also result from nonlinearities in GCaMP activity, an idiosyncratic expression pattern biased towards neurons with weaker preparatory signals, or other unknown factors. This underscores the need for future research to better understand the relationship between neural dynamics inferred from calcium signals versus from extracellular physiology (Wei et al. 2019).

3. The values of T_{int} and T_{skip} should be indicated in the Results —the authors only cite a previous paper in the Methods. Were they identified on a session-by-session basis during decoder training? If so, how? It would also be useful to see these intervals graphically, perhaps in Fig 4K or in the new figure I suggest the authors to make in #5 below.

This is an important point which we thank the reviewer for addressing. We have added the following text in the methods section to clarify our process for selecting these parameters:

While it is possible to develop a mathematically optimal procedure for selecting T_{skip} and T_{int} , in practice, we found that it was helpful to use a permissive integration window to capture the natural behavioral variability of an animal, while also minimizing the number of non-informative imaging frames that contribute to the average frame calculation. We experimented with a range of values to improve decode performance and speed (Supplementary Fig. 14): T_{skip} values ranged from 200-400ms and T_{int} values ranged from 70-200ms. These ranges were selected based on several factors, including:

- 1. Variability in the reaction time of the animal. The integration time should be centered around the beginning of the attempted movement. On trials in which an animal had a slow reaction time, the integration window could occur too early, before a decodable signal was present (in our data). As such, there exists a tradeoff between decoder speed and accuracy, taking behavioral variability into account.*
- 2. The typical timing of divergence of calcium signals within the imaging region based on offline analysis, as shown in Fig. 6h,i. We note that this timing may be specific to the recording location and subpopulation of cells recorded.*
- 3. Image acquisition time (33 ms). Only full frames are considered during decode, so values of T_{skip} and T_{int} in practice only have temporal resolution of the frame period.*

Rather than create a multi-objective cost function to optimize these values, in practice, we chose to bias towards increased robustness by using a permissive T_{int} (long window) and

short T_{skip} to increase the likelihood of movement onset and peri-movement neural activity falling within the integration window.

The scatterplot below shows the T_{skip} and T_{int} parameter values (with minor x/y jitter to visualize multiple overlaid points) used for the 36 decode sessions reported here. We have added this as Supplementary Fig. 14 and copied below:

Supplementary Fig. 14: Parameter values used for online decoding sessions.

In addition, we performed additional offline analysis across decode sessions to better understand the timing of calcium signal divergence from baseline in this data. A more thorough discussion, including figures, is presented in response to this reviewer's question 6 below.

4. I may be wrong, but there seems to be a trend toward decoder performance degrading during a session (Fig 5E) (it would be helpful if the authors added the mean +/- SEM accuracy to these traces). Is that case? If it is indeed happening, do the authors think it is due to the progressive drift of the imaged ROIs? In relation with this, what was the total duration of one experimental session?

Thank you for the helpful suggestion. We've now added mean +/- SEM shaded bars to the revised figure panel (now Fig. 6e), copied below. In addition, we've added the following text to results:

We observed a slight but significant decrease in decode performance over the duration of decode sessions (2 condition: -0.05%/ trial, $p = 1.40e-28$; 4 condition: -0.09%/ trial, $p = 2.13e-09$). It is not the case, however, that performance decreases in all sessions or that if performance has decreased that it cannot later recover.

We have also added the following to the methods section:

oBCI experimental sessions were often terminated to explore different imaging depths or fields of view the same day, as opposed to ending a session after observing roll-off of decode performance. In many if not most cases, we anticipate that the image decoder would continue to operate successfully for much longer than the acquired series of data.

As the reviewer noted, small shifts or drift in the imaging plane could cause a decrease in performance. In practice, additional factors such as the animal's motivation and reaction time also impact performance, but it is difficult to isolate the specific causes in each case. As a technical note, in future work, it may be possible to make imaging less sensitive to X/Y axis shifts or drift using an online image registration algorithm, and by stretching the excitation laser point spread function in the z-dimension to imaging a thicker optical section, thereby making z-axis position less sensitive. We chose not to implement these here as we wanted to test performance using standard imaging techniques and demonstrate the stability over hundreds of trials without these measures.

5. I'd like to have a better sense of the data: a new section and figure on it before or after the decoding results would really strengthen the paper. I'd like to see things like:

We thank the reviewer for this suggestion. We have added a new set of analyses on ROIs automatically extracted from each imaging session, as well as new main Fig. 5 and Supplementary Fig. 10 that demonstrate these responses for 3 example sessions. Please find our responses to each of the specific questions posed inline below.

- the number of ROIs identified and used for decoding for each monkey across days (in case they don't match)

We have added the following to the results section to quantify the number of ROIs:

To identify ROIs within each of the 36 imaging datasets, we used Suite2P, an open-source tool which does not make assumptions about the shape of ROIs and can readily identify elongated structures belonging to neural processes (Pachitariu et al. 2017). Suite2P identified 596 ± 126 ROIs (mean \pm s.d.) across the 36 imaging datasets (total 21,451 ROIs).

The imaging sessions used for these offline ROI analyses are the same as those used for online decoding sessions. Note however that the online decoding system fundamentally operates at the level of individual pixels, comparing each image frame to a condition-specific template computed during training. The decoder itself has no concept of ROIs, only pixels. All ROI analysis was performed offline on the same imaging data.

- how homogeneous is the tuning within each identified ROI?

We have devised a pixel-wise clustering approach within ROIs to address this question and added the following to the manuscript:

Pixel-wise direction tuning within individual ROIs was generally homogenous; only 3.1% of ROIs and 2.8% of putative dendritic/axonal ROIs exhibited significant tuning heterogeneity (assessed using k-means clustering, see Methods).

And we have described the methodology in the Methods:

To assess the heterogeneity of tuning within individual ROIs, we regressed each individual pixel within an ROI onto hand velocity, producing a set of pixel-wise regression coefficients. We then performed K-means clustering on these coefficients and identified the optimal cluster count which maximized the Calinski-Harabasz criterion values. We considered an ROI to have multiple meaningful clusters if the optimal cluster count was greater than 1, and if at least 30% of the pixels belonged to at least two such clusters, and the difference in preferred direction corresponding to the centroid of these clusters were at least 60 degrees.

- how is the information distributed across layers (more superficial processes vs somata in superficial layers)?

As our primary goal was to ascertain whether superficial dendritic and axonal processes carry movement direction information, we did not optimize our imaging sessions to collect

signals from expressing somas. Also, all of our decoding imaging sessions collected functional signals close to the cortical surface, and we did not systematically vary imaging depth as a parameter of interest. For these reasons we are not able to comment on the distribution of information across layers in this report. We do agree that this is an important and interesting question, and one which can potentially be answered with imaging, as superficial processes are challenging to isolate using electrophysiology. However, definitively answering this question is beyond the scope of this report.

We did attempt to identify dendritic / axonal ROIs among the Suite2P identified ROIs. We did not observe any strong differences between the dendritic / axonal relative to the other ROIs, except that a higher fraction of dendritic / axonal ROIs exhibited significant tuning. However, the set of other ROIs includes many non-somatic bits of fluorescence, including weakly fluorescing patches of neuropil. This should not be regarded as an ideal comparison between responses in superficial processes vs. somatic responses. We agree with the reviewer that this question is ripe for future research to investigate, using both optical tools as well as linear electrode arrays like V-probes and Neuropixels.

- how stable are the identified ROIs during a session (e.g., based on the correlation of the activity in the imaged ROIs between the first half and the second half of trials after matching targets; a few examples of mean image for the first 25 % of the trials and the last 25 % of the trials would be useful too)

This is a great suggestion. We have implemented exactly this analysis to address this, as described now in the Methods:

To assess the stability of tuning individual ROIs within imaging sessions, we separately assessed preferred directions for each ROI for the first 25% and last 25% of each session. We computed the difference of preferred directions and a null distribution computed as the difference of preferred directions computed from 1000 random reshufflings of these early / late imaging frame labels. Drifts of preferred directions were considered significant if the absolute change in preferred direction exceeded that of 99% of the random shuffles ($p < 0.01$).

And reported the result in the main text:

Within each experimental session, estimated tuning remained stable for the majority of ROIs; only 11.3% or 752 / 6653 direction-tuned ROIs exhibited a significant drift in preferred direction (shuffle test, $p < 0.01$, see Methods and Supplementary Fig. 13).

Although statistically significant, we expect that these significant drifts are still likely to be due to estimation noise in the regression rather than indicative of systematic shifts in the underlying physiology of these processes. Having more finely spaced reach direction conditions might also help to improve the precision of estimating the preferred direction as well in order to increase confidence in tracking small shifts of this quantity. In Supplementary Fig. 6,

we also focus on tuning stability specifically in the putative dendritic ROIs, which do not exhibit significantly more volatile tuning relative to non-dendritic ROIs.

- a more detailed quantification of how well the ROIs match across days

We have demonstrated a small number of examples of ROIs which exhibited consistent tuning on sequential imaging days. We were able to match these specific ROIs based on being reasonably confident that we had returned to roughly the same FOV with the objective lens and identifying idiosyncratic features of the neuronal processes that could be reliably distinguished from others in the close neighborhood. However, we recognize that this result is preliminary, as it was not our primary goal to return to the same ROIs repeatedly. Rather than focus in on a few isolated dendrites, we have decided that displaying the direction tuning of the same FOV over multiple sessions would more effectively convey this point. To this end, we added Supplementary Fig. 6:

Supplementary Fig. 6: Consistent tuning across experimental sessions. Each pair of images corresponds to an individual imaging session, all of which were collected at highly overlapping fields of view. Left panels show enhanced mean images; right panels show ROI tuning maps. Individual dendritic / axonal processes can be identified using a combination of their idiosyncratic morphology and their location relative to the consistent shadow created by superficial vasculature. Two examples are identified with yellow and green stars in each image where they are visible, where the tuning maps show preferred directions of up (yellow) and left (green) in each session.

- raster versions of Fig. 4K for all the ROIs across a few example sessions (e.g., mean delta F/F traces for each target and ROI), etc.

Thank you for this suggestion; we agree these plots would be useful. We have included in our new Fig. 5 a raster version showing the responses of each movement-modulated ROI, trial-averaged for each target direction. We have also included two additional datasets in the same format in Supplementary Fig. 10. We have also included single trial rasters in Supplementary Fig. 8, both copied below:

Fig. 5: Functional responses during reaching behavior. (A) Contrast enhanced mean image of example FOV. (B) Reaching kinematics observed during behavioral task. Move indicates movement onset; Acq indicates target acquisition. (C) ROI tuning map showing preferred direction of ROIs with significant direction tuning. Inset color wheel indicates reach direction. (D) Trial-averaged responses of dendritic ROIs, normalized to baseline fluorescence, for the four reaching directions indicated by the colored arrow in the bottom left of each raster. ROIs are sorted by preferred direction beginning with rightwards and proceeding counterclockwise; triangular ticks at left edge indicate locations of preferred directions of up-right, up-left, down-left, down-right. (E) Single-trial (thin lines) and trial-averaged (thick lines) population trajectories, projected along condition-independent signal (CIS) dimension and condition-dependent X and Y dimensions (see Methods), color coded by reach direction condition.

Supplementary Fig. 10: Functional responses during reaching behavior. Same format as Fig. 5 in main manuscript but for two additional fields of view.

Supplementary Fig. 8: Single trial rasters of all ROIs showing significant modulation during movement (see Methods). (A) Each row represents the activity of a single ROI over time as DF/F relative to baseline. The rows of the raster are sorted vertically using rastermap (<https://github.com/MouseLand/rastermap>), a multi-dimensional embedding algorithm that groups similar ROIs close together. Below the raster are the hand velocity in X and Y, including reaches from the center to the target and the reach back to the center position. Short breaks in the hand trace represent single unknown hand position samples; time runs continuously in this plot. (B) Same as (A) but for a second imaging session at a different field of view.

6. I'd also like to see a more thorough offline analysis of their decoder experiments. For example: how is the information distributed across ROIs? And across layers? How does decoder performance degrade as ROIs are dropped?

Thank you very much for the helpful suggestions, which we've addressed individually below. With respect to ROIs, the modified Fig. 5, shown above, provides considerably more detail regarding the distribution of information and tuning across the population. For the decode analysis, it is important to stress that online decode was based on pixel values and not ROI calculated traces. ROI calculated traces are given by weighted linear combinations of raw pixel values inferred from correlation structure, whereas the decoder is trained and tested solely using raw pixel values. To address the spirit of this question, we performed a pixel-dropping experiment using offline movement-aligned imaging data where we masked out squares of pixels representing different fractions of the overall image. We then swept the fraction of masked pixels. As anticipated, we found that decoder performance degrades as the percentage of masked pixels increases. Notably, however, there is considerable redundancy in the information in pixel space, so even when 90% of pixels are dropped from a contiguous region, there are still > 26,000 pixels remaining and the performance is still above chance in this case. We observe this in the data as well, with the decoder performance decreasing gradually until most of the pixels are masked. This is now shown for an example session in Supplementary Fig. 15 and copied below:

Supplementary Fig. 15: Impact of pixel dropping on offline performance for one example session.

What would be the cross-validated performance of a decoder built on single trial data compared to their approach?

This is also a great suggestion. We've performed additional offline analysis and included the following in the results (Fig. 6 H, I). We note that this analysis also informs the selection of T_{skip} and T_{int} parameter values, as the reviewer previously noted.

We also compared this online MMSE decoder with an offline decoder using multi-class support vector machine classification with 5-fold cross validation to understand the timing of signal divergence from these signals and to benchmark our online performance against an optimal linear classifier. We observed offline decode performance diverge from chance levels ~250 ms after the go cue presentation (Fig. 6h) and ~100ms prior to movement onset (Fig. 6i), reaching a mean decode accuracy of 61.3% 200 ms after movement onset. As anticipated, this is somewhat better performance than the mean online decode performance for 4-condition decode (45.4%, 9 sessions).

And even, how does the eigenvalue distribution of the $\Delta F/F$ look? (this could help have a crude sense of the “dimensionality”)

We thank the reviewer for the helpful suggestion. We plotted the eigenvalue distribution for a number of datasets, which we've added to Supplementary Fig. 12 and copied below (left panel). In order to get a sense for the dimensionality of the data, we also plotted the eigenvalue spectrum normalized by the first eigenvalue for a number of datasets (right panel). Here, we see the knee in the eigenvalue spectrum range from ~4-10 components. This is in the ballpark of the dimensionality of data that we would expect from electrophysiological recordings for a point to point reaching task with several conditions.

Supplementary Fig. 12 *Left: Eigenvalue scree plot of dF/F imaging data, multiple sessions. Right: normalized eigenvalues shown with restricted x-axis range to illustrate knee in the scree plot.*

7. I'm obviously impressed by the authors' identification of a Betz cell. Yet, I wonder: could the authors replicate this result and identify more neurons? Their result is great, but it's an $n = 1$... and besides this result is quite prominently featured in the abstract. Note that I'm not an expert in CLARITY so I don't have a good sense of whether this is feasible.

We agree with the reviewer that this is a small- n demonstration of a particular idea, and the presentation of the limited example must be appropriately cautious. Our intent in including this example alignment was *not* to indicate that this is now a solved problem in that it is robust or easily achieved as a turnkey solution for aligning functional data with anatomical data. Rather, we included this example to illustrate that in at least two examples, we can find the exact cell in layer 5 that sends a dendrite to the superficial layer where *in vivo* imaging was performed. We have also added additional quantification of the CLARITY imaging volume, included tracing many additional cells. While these cells are not individually identified across *in vivo* and *ex vivo* imaging, the expression patterns argue that many of the neural processes we imaged during behavior were apical dendrites that projected into the superficial region accessible to 2P imaging. This is essential to validate the primary point, that is indeed possible to gain optical access to functional signals from many cells with somas in deeper layers.

While we wholeheartedly agree with the reviewer that we do not want to oversell these particular examples, we feel that this is an important demonstration for several reasons. First, it grounds the suspicion that one should be able to access signals from Betz cells by imaging apical dendrites in primates. This is relatively new territory regardless of animal model and has extra utility in primates and the motor system in particular where layer 5 cells serve as the output. Second, it motivates further work to establish genetic promoters or combinatorial targeting strategies to selectively express GCaMP in layer 5 corticomotoneuronal cells. Even this one example provides an existence proof that the fluorescent traces from dendrites identified in CLARITY were functionally tuned and observable during behavior, as many of the

neural features identifiable via fluorescence in our imaging datasets were not modulated or functionally tuned.

To demonstrate the broader utility of CLARITY imaging beyond the two identified examples, we performed additional analysis to quantify 1) the number of neuron cell bodies that express GCaMP6 indicators in cortical layers 2/3 and 5 in the CLARITY imaging volume, and 2) the number of layer 5 neuron cell bodies that project apical dendrites superficially.

We added these results to Fig. 7j,k and copied below:

Fig. 7 - Identifying neurons in CLARITY and functional imaging. (A-I not shown here). (j) Cell bodies identified in layers 2/3 (green) and 5 (cyan) with their dendrites extending to the superficial layer, as traced by CLARITY and highlighted in different colors. (k) Traced neurons only.

We have added the following text to the results section to reflect this revised emphasis:

The expression patterns observed in the ex vivo tissue volume demonstrated that many neurons in layer 2/3 and 5 cells expressed GCaMP6 both somatically and in their apical which extended towards the cortical surface and arborized within the superficial region imaged in vivo. We quantified the number of neuron cell bodies that expressed GCaMP6 indicators in cortical layers 2/3 and 5 in one CLARITY imaging volume (approximately 1.3 mm x 1.3 mm x 2.2 mm pictured in Fig. 7j-k). We identified 22 pyramidal cells in layers 2/3 and 31 in layer 5 (Fig. 7j-k, segmented somas and their dendritic arbors are highlighted). We were able to trace seven layer 2/3 and 12 layer 5 somas and their apical dendrites extending to the surface (Fig. 7j). In some instances, we could identify a specific neuron in the CLARITY volume that was also imaged in vivo during oBCI decoding. This was enabled by the presence of clearly identifiable features observed across 2P in vivo functional imaging, 2P in vivo volumetric z-stacks, and ex vivo 2P CLARITY imaging. One example was a directionally-tuned apical dendrite of a cell originating approximately 1500 μm below the surface of cortex (white dashed rectangle in Fig. 7d-e and reconstructed in 7h-i). The cell body of this neuron was approximately 60 μm in diameter, suggesting that this cell is likely to

be a Betz cell, a class of large upper motor neuron which projects to the spinal cord⁶⁰. A second example is illustrated in Fig. 8, facilitated by the idiosyncratic shape of the expressing dendrite.

With respect to registration of imaging volumes across modalities, we added this text to the discussion section:

Registration of in vivo and ex vivo cleared tissue volumes is a known challenge. Confounding factors include 1) different imaging planes and angles and often different imaging modalities from in vivo to ex vivo, 2) differential fluorescence signal in vivo vs ex vivo, specifically a much denser network of labeled processes from anti-GFP staining ex vivo, 3) potential tissue deformation from tissue extraction and the clearing process, particularly of the superficial layers. In this particular case, fiducial markers were limited to the surface vasculature and therefore did not provide any information on the depth of the functional ROIs within the CLARITY volume. Given these challenges, we were not able to register more functional ROIs in the superficial layer with deep-layer cell bodies and dendrites identified by CLARITY besides the demonstrated examples. However, there are many ongoing efforts to register functional and ex vivo tissue volumes. For example, this has been essentially solved in larval zebrafish (Lovett-Barron et al. 2017) where activity imaging is successfully registered to multiple rounds of antibody staining at cellular resolution. In rodent, registration of in vivo and cleared tissue volumes has been aided by sparse labeling and fiducial markers throughout the volume. For example, recordings using Neuropixels probes have been registered to their position within the Allen Brain atlas in mouse by using fluorescent-labeling of the electrode tracks (Allen et al. 2019). As these techniques are optimized for use in rodents, we expect they will aid efforts in primate in future work.

In addition, we have now added a supplemental movie showing the distribution of cell bodies and neural processes at each imaging plane (i.e. depth) as the plane moves from deeper to superficial layers (Supplementary Movie 1), which helps to highlight the density apical dendrites in superficial layers which originate from large cell in Layer 5.

Finally, we tempered the discussion of CLARITY findings in the main text, copied here:

We also localized functionally-imaged neural sources in a post mortem 3D CLARITY volume^{38,39} and demonstrated that layer 5 Betz cells with GCaMP-labelled apical dendrites were densely represented within this tissue volume and that in one case, dendritic signal sources originated from a layer 5 projection neuron.

8. I'm curious as to whether the authors tested any of the calcium imaging processing pipelines that have become standard in the field like Suite2p (Pachitariu et al 2016) or CNMF (Pneumatikakis et al 2016) on their data, rather than developing their own. Also, did they compare their online processing to offline "sorting" or is this not necessary because the online implementation didn't require any algorithmic "shortcuts"?

As discussed above, at this reviewer's suggestion, we have completely redone and greatly expanded our offline analysis of functional ROI responses using Suite2P to perform registration and ROI extraction. Initially, our rationale for implementing our own rather simple ROI extraction algorithm was to be maximally conservative, to isolate a handful of very clearly dendritic responses for demonstration. However, using Suite2P allows us to perform thorough and systematic analyses of the responses across all of the imaging sessions.

For the analysis now presented in this manuscript, we focused on algorithms that make no assumptions about the geometry or morphology of a signal source in order to identify contributions from dendritic structures. We selected Suite2P to identify ROIs, which effectively located both dendritic and non-dendritic regions within each FOV.

Specifically, for the decode, we do not anticipate that first isolating specific ROIs and then constructing a decoder from ROI averaged signals would improve (or hurt) decode performance. ROIs are weighted linear combinations of the raw pixel traces that are defined based on the spatial correlation structure. As a preprocessing step in the decoding pipeline, this could potentially serve as a denoising and dimensionality reduction step. However, learning the ROIs is time consuming, and the decoder that we use is very simple and unlikely to overfit the data. We found empirically that the mild approximate Gaussian blur suffices to achieve some denoising while also increasing robustness to small translations of the tissue.

While there is considerable room for further optimization of the entire pipeline, we anticipate that there are much greater gains in performance to be had through further optimizing viral expression to improve the reliability of signal quality, which would enable more complex continuous BMI control tasks such as continuous cursor control. These more complex tasks would enable a richer and more thorough investigation of the signal processing requirements and impact of mixing information from multiple cellular sources or preprocessing data with ROI masks on BMI information throughput as bottlenecked through a decoder.

9. As I said above, I like how the paper is written. However, in some parts I had the impression that the authors were trying to spin a very impressive methods paper into a more scientific paper, whereas the overall tone is focused on the methodological advances. I suggest they take this latter view.

We appreciate the reviewer's perspective and agree that it is imperative not to overstate the scientific claims while highlighting the technical demonstrations. We have attempted to more clearly delineate the scientific claims being advanced in this paper throughout the introduction, results, and discussion.

We have chosen to include important details of the methodology in the manuscript. We feel these details are important primarily because (a) 2P imaging in the macaque motor cortex has not previously been reported and consequently a reader must be convinced the approach is viable to believe the scientific conclusions of this work, (b) imaging neuronal processes is particularly difficult given their small size and given the amount of tissue movement we observed before developing our stabilization approach, and (c) the capability of locating fields of view for individual neurons across 2P imaging and CLARITY required significant time and effort to develop, which then allowed us to conclude definitively that we were imaging from

deeper cells as well as superficial ones. We respectfully note that to our knowledge, none of the reviewers has brought into question these core scientific claims, which reassures us that our methodology is sufficiently well described.

We respectfully assert that we feel that our primary findings—that superficial dendrites/axons in primary motor cortex exhibit directional tuning of sufficient single-trial SNR to drive a real-time BCI system—is scientifically important and not merely spin. These superficial processes, though quite difficult technically to record electrically, provide a readily optically accessible window to the population activity of deeper neurons including the projection cells that drive downstream motor circuits. We have also validated that it is possible to identify individually labeled neurons in both functional recordings and post-mortem CLARITY volumes.

Moreover, the broader landscape of scientific questions that can be addressed in the primate motor system by engaging the toolkit of modern genetics is vast. We have attempted to explore a set of questions that can be addressed with optical methods in our previously published perspectives piece (O'Shea et al. 2016). The methodology we describe here can and will surely be improved considerably in macaques and marmosets going forward. And certainly by resolving these limitations, more expansive and definitive scientific conclusions about primate motor circuitry can be addressed. We also highlight the avenues for further scientific investigation opened by imaging superficial neural processes in the “Imaging neural projections” section of the discussion.

Minor:

I am curious as to whether the Supplementary Figures are not cited in order

We thank the reviewer for noticing that supplementary figures were referenced out of their presentation order and we have fixed this in the revised manuscript.

Line 6: “...from in...” is a mistake

Thank you, this typo is now fixed.

Line 29: “arm and hand movements”. Understanding hand motor control will be critical for FES- or spinal cord stimulation-based neuroprosthetics. With regards to neuroprosthetics, I thought the recent study by the Courtine group showing restoration of walking after SCI with BCI-controlled spinal stimulation was missing (Capogrosso et al 2016)

We've added this citation and enthusiastically agree with the reviewer that we should widen the scope of the framing to include locomotor prosthetics, which also benefit from basic science investigations of the arm and hand, spinal cord, among other areas.

Line 32: perhaps cite some tools to measure large neural populations?

We have added citations for several additional tools for measuring large populations in vivo, including the mesoscope [Sofreniew et al. 2016], and Neuropixels [Jun et al. 2018] as well as additional discussion regarding the tradeoffs associated with different measurement modalities.

Line 42: while I appreciate the potential advantage of recording from different brain areas by just moving the objective lens, doing so also requires having a larger imaging window, with the additional complications and risks this poses. Despite the impressive advances shown by the authors (e.g., 18 mm window), I think this should be discussed

We thank the reviewer for clarifying this particular point. There exists a tradeoff between imageable surface area (which is of course valuable to most experiments) and impact on the subject, which includes a risk of infection, which is difficult to quantify. The size of the durotomy and the design of the artificial dural window is similar to those reported by other groups previously [e.g.: Shtoyerman et al. 2000; Arieli, Grinvald, and Slovin, 2002; Seidemann et al. 2016]. We have added the additional text to the discussion to clarify this important tradeoff for readers, copied here:

This implant features an implantable titanium cylinder and silicone artificial dura and was designed to balance the competing goals of maximizing the area of imageable tissue while minimizing the craniotomy and durotomy diameter and subsequent impact on the subject and infection risk.

Line 146: “comment” should be “common”

Thank you, this typo is now fixed.

Fig 2 A,B,G: make the text larger. Same for Fig. 3H-K

Thank you, we have increased the text size.

Fig 4H: is the colored circle on the top left a legend that indicates the directional tuning of the ROI?

Yes, the color wheel conveys the perceptually uniform color scale used to convey directional tuning. We've revised these tuning map figure and updated the figure legend (now Fig. 5) to make this clearer, thank you. We have also used consistently colored directional arrows to reinforce this directional color scheme in other figures.

Fig 4M: Make larger; it's hard to appreciate in the current version.

We've separated the original Fig. 4 into separate figures 4 and 5 to provide more room for these key panels.

Fig 5G: aren't the “incorrect” and “correct labels of the top figure swapped?

The plots in 5G (Now Fig. 6G) represent the distance between test and training data used in the decoder, and lower values represent images that are closer to the training set. We have updated the figure caption to make our specific quantification clearer:

Offline decoder value (cross-condition mean subtracted mean-squared error) using rolling 6-frame average for single trials (lower values represent images closer to the training set).

Line 308: As shown in Fig 5 -> Fig 6?

Thank you, this is now fixed.

Fig 7B,D: Did the authors use yellow because the neural feature of interest was tuned to the yellow direction in A? I'm intrigued as to whether a part of the neural process is tuned to the yellow direction and the other to the nearby green direction?

We apologize for being unclear about our use of color throughout different figures. For the CLARITY results presented in Figs. 7 and 8, we used yellow to highlight morphological features of interest only and did not intend to associate this with the tuning results previously presented. The yellow highlight was applied to several vasculature features that we could reliably identify across imaging modalities and at different scales (Widefield fluorescence imaging, 2P in-vivo, and post-hoc CLARITY imaging).

Lines 339-340: this needs further quantification; my proposed new figure (Comment 5) would help.

Thank you, we've included this suggested figure, and have modified this sentence to read:

We leveraged wide-field imaging of vascular fiducial markers to return to the same neurons and observe consistent direction tuning in neuronal processes over seven sessions spanning 13 days (Supplementary Fig. 6).

Lines 459: the authors may find useful a recent preprint by Gallego, Perich et al, where they quantify the large changes in movement tuning parameters of Utah array recordings over up to two years, and show how they can stabilize a decoder using an idea similar to that in Trautmann et al 2019 despite these changes. The same group has a follow-up publication using deep learning methods to accomplish a similar goal (Farshchian et al 2018). Moreover, the Batista, Chase & Yu groups are working on a similar approach, but I do not believe it is available yet other than as SfN abstract. I also noticed a paper from the Shenoy group and another by Wu & Hatsopoulos that I think could be worth citing (Chestek et al 2011; Wu & Hatsopoulos 2008).

Thank you for this suggestion. We've added citations to Gallego, Perich et al. 2020, Sussillo et al 2016, and Chestek et al. 2011 to the discussion:

Multielectrode array recordings in M1 and PMd have been used to study population changes in motor cortex during visuomotor adaptation⁶³ and force field adaptation⁹¹ and have been used to study degradation in decode performance over time⁹². Recent work has demonstrated methods for making a decoder robust to changes in tuning and/or recording stability over timespans of months or years^{93,94}.

Citations relevant to the paragraph above:

63. Vyas, S. et al. *Neural Population Dynamics Underlying Motor Learning Transfer*. *Neuron* **97**, 1177–1186.e3 (2018).
91. Perich, M. G., Gallego, J. A. & Miller, L. E. *A Neural Population Mechanism for Rapid Learning*. *Neuron* **100**, 964-976.e7 (2018).
92. Chestek, C. A. et al. *Long-term stability of neural prosthetic control signals from silicon cortical arrays in rhesus macaque motor cortex*. *J. Neural Eng.* **8**, 045005 (2011).
93. Gallego, J. A., Perich, M. G., Chowdhury, R. H., Solla, S. A. & Miller, L. E. *Long-term stability of cortical population dynamics underlying consistent behavior*. *Nature Neuroscience* **23**, (2020).
94. Sussillo, D., Stavisky, S. D., Kao, J. C., Ryu, S. I. & Shenoy, K. V. *Making brain-machine interfaces robust to future neural variability*. *Nat. Comm.* **7**, 13749 (2016).

References:

-
- M Capogrosso et al. *A brain–spine interface alleviating gait deficits after spinal cord injury in primates*. *Nature* 2016
- C Chestek et al. *Long-term stability of neural prosthetic control signals from silicon cortical arrays in rhesus macaque motor cortex*. *J Neural Eng* 2011
- BM Dekleva, KP Kording, LE Miller. *Single reach plans in dorsal premotor cortex during a two-target task*. *Nature Communications* 2018
- A Farshchian, et al. *Adversarial Domain Adaptation for Stable Brain-Machine Interfaces*. arXiv:1810.00045
- JA Gallego, MG Perich, RH Chowdhury, SA Solla, LE Miller. *A stable, long-term cortical signature underlying consistent behavior*. bioRxiv 2018
- M Pachitariu, C Stringer, M Dipoppa, S Schröder, LF Rossi, H Dalgleish, M Carandini, KD Harris. *Suite2p: beyond 10,000 neurons with standard two-photon microscopy*. bioRxiv 2017
- EA Pnevmatikakis et al. *Simultaneous Denoising, Deconvolution, and Demixing of Calcium Imaging Data*. *Neuron* 2016
- W Wu, NG Hatsopoulos. *Real-time decoding of nonstationary neural activity in motor cortex*. *IEEE Trans. Neural Syst. Rehabil. Eng.* 2008

Reviewer #2 (Remarks to the Author):

Trautmann et al. used two-photon (2P) imaging of the rhesus macaque premotor and primary motor cortices to detect calcium signals from dendrites in the superficial layer that displayed stable tuning for different directions of arm movement across several weeks. In addition, they developed an optical brain-computer interface (oBCI) with these calcium signals to successfully decode the movement direction online. After imaging, the authors treated the imaged tissue with CLARITY, which showed that one of the imaged dendrites originated from a deep layer 5 neuron. This is the first demonstration of 2p calcium imaging in the macaque motor cortex and online decoding of movement direction. This method has some advantages over electrical recording because 2p imaging through a large cranial window can reveal the activity of large populations of densely localized neurons over several weeks and is not invasive. Also, it can be used to study neural population activity, which will be critical for the design of next-generation BCIs. The current study provides an important advance in the field of motor control. I recommend the paper be published in Nature Communications if the following concerns are properly addressed.

We thank the reviewer for their kind comments, and we believe that we have considerably improved the manuscript to address each of the helpful comments raised below.

1) It is crucial to show clearly the reliability of 2p calcium imaging.

We thank the reviewer for highlighting a number of crucial questions regarding consistency and reliability of 2P-based calcium signals in macaque motor cortex. To ensure that we've addressed each issue raised by the reviewer, we have broken up the reviewer's paragraph into individual questions and addressed each in turn.

How many ROIs morphologically corresponded to dendritic branches in each field of view?

As we noted in our response to the first reviewer, and we repeat here for convenience, we have replaced our original *ad hoc* ROI analysis with a standard pipeline in the field (Suite2P), which is capable of identifying dendritic ROIs as it does not employ prior assumptions about ROI shape. This enabled us to automatically identify ROIs in all 36 of our imaged volumes and to quantify their functional properties.

With respect to this specific question, we used the shape of the ROIs to identify a subset of putatively dendritic / axonal ROIs. We added the following text to the main results section addressing this question:

We identified ROIs within each of the 36 imaging datasets using Suite2P, an open-source tool which does not make assumptions about the shape of ROIs and can readily identify elongated structures belonging to neural processes (Pachitariu et al. 2017). Suite2P identified 596 ± 126 ROIs (mean \pm s.d.) across the 36 imaging datasets (total 21451 ROIs) Functional modulation of GCaMP-expressing neuronal processes was observed in gyral M1

(Fig. 5A) during motor behaviors and assessed using an instructed-delay reach task, which elicited rapid, straight reaches to each of four radially positioned targets (Fig. 5B). The majority of identified ROIs exhibited a response just prior to or during arm movement (65.9%, rank-sum test, $p < 0.01$). This modulation time-locked to movement was readily visible in single trial raster plots (Supplementary Fig. 8). In contrast, relatively few ROIs exhibited responses time-locked to the target cue (1.3%, rank-sum test, $p < 0.01$) or to the go cue (1.7%, rank-sum test, $p < 0.01$). During movement, 31.0% of ROIs exhibited direction-tuned responses (ANOVA, $p < 0.01$).

We identified a subset of ROIs corresponding to putative dendritic or axonal processes by using the shape of the ROIs (see Methods). This identified 20.3% of total ROI, or 121 ± 69 ROIs per imaging session (Supplementary Fig. 9) with an elongated shape. This subset of ROIs also exhibited significant responses to movement (81.3%, rank-sum test, $p < 0.01$); directionally-tuned movement responses were observed in 50.3% of putative dendritic/axonal ROIs (ANOVA, $p < 0.01$). Few responses to the target cue (2.8%, rank-sum test, $p < 0.01$) or the go cue (1.5%, rank-sum test, $p < 0.01$) were observed.

It should be noted that we were specifically attempting to image fields of view that possessed functionally modulated dendritic processes in superficial cortex and, in this way, there was some selection bias for such fields of view.

Did some ROIs originate from the same neuron and show highly correlated spontaneous activities? Although the authors indicated that they could access neurons in L2/3 and showed an L2/3 cell body image in Figure 4E, F, they did not show these activities.

We thank the reviewer for raising these interesting questions. Unfortunately, our 2P imaging datasets are acquired at a single depth plane, primarily because we sought to maximize the usefulness of the functional datasets for oBCI decoding. Consequently, we are not able to assess which ROIs belong to the same soma. Even for the two individually identified cells we were able to localize in the *ex vivo* CLARITY dataset, we could not confidently reconstruct the full dendritic arbor and localize all branches within the *in vivo* functional dataset. In future work, it will be interesting to pursue these questions, perhaps with a Z-scanning piezo.

The authors should show some examples of raw fluorescence traces of the dendrites and cell bodies over several minutes. What proportion of ROIs showed movement-related activity and direction tuning? Was there any neuron that responded to the go-cue presentation or the reward?

Thank you, we have added fluorescence traces to Fig. 5 and Supplementary Fig. 8 and 10. In Supplementary Fig. 8, we elected to show a window of time corresponding to 10s of seconds, where the trial structure is visible, as the same data on the timescale of minutes was too crowded to see this structure. However, each of the imaging sessions was many minutes in duration and all analyses used the full time course of ROI responses.

We have also added a systematic analysis of the ROI responses which addresses these questions. From the results:

The majority of identified ROIs exhibited a response just prior to or during arm movement (65.9%, rank-sum test, $p < 0.01$). This modulation time-locked to movement was readily visible in single trial raster plots (Supplementary Fig. 8). In contrast, relatively few ROIs exhibited responses time-locked to the target cue (1.3%, rank-sum test, $p < 0.01$) or to the go cue (1.7%, rank-sum test, $p < 0.01$). During movement, 31.0% of ROIs exhibited direction-tuned responses (ANOVA, $p < 0.01$).

We identified a subset of ROIs corresponding to putative dendritic or axonal processes by using the shape of the ROIs (see Methods). This identified 20.3% of total ROI, or 121 ± 69 ROIs per imaging session (Supplementary Fig. 9) with an elongated shape. This subset of ROIs also exhibited significant responses to movement (81.3%, rank-sum test, $p < 0.01$); directionally-tuned movement responses were observed in 50.3% of putative dendritic/axonal ROIs (ANOVA, $p < 0.01$). Few responses to the target cue (2.8%, rank-sum test, $p < 0.01$) or the go cue (1.5%, rank-sum test, $p < 0.01$) were observed.

Figure 4M and Supplementary Figure S4 elegantly showed that a subset of dendrites acquired tuning for the movement direction that remained stable over several days. However, it is not clear whether these dendrites showed stable tuning over all imaging days or over some imaging days. It is possible that the tuning of other dendrites was unstable, i.e., their tuning changed or even disappeared on some imaging days.

Please show changes in the proportions of the task-related neurons over several days.

Thank you, we've expanded this analysis and included additional data to better support the analysis. We note that these data were not explicitly collected with the goal of assessing neuronal stability, but that we can nonetheless address this question because we returned to the same field of view (or highly overlapping fields of view) multiple times. Rather than focus on a small number of individual dendrites, we have elected to show the entire field of view and the tuning map in from each session. The revised Supplementary Fig. 6 is copied below.

Supplementary Fig. 6: Consistent tuning across experimental sessions. Each pair of images corresponds to an individual imaging session, all of which were collected at highly overlapping fields of view. Left panels show enhanced mean images; right panels show ROI tuning maps. Individual dendritic / axonal processes can be identified using a combination of their idiosyncratic morphology and their location relative to the consistent shadow created by superficial vasculature. Two examples are identified with yellow and green stars in each image where they are visible, where the tuning maps show preferred directions of up (yellow) and left (green) in each session.

2) As the authors repeatedly pointed out, the major advantage of 2p imaging is the ability to identify a population of recorded neurons. However, the fluorescent changes in the whole field of view were used to drive the oBCI decoder. Thus, it is critical to show how many ROIs and different neurons were included in each oBMI decoding and the dependency of decoder accuracy on the number of neurons. It is also important to show that the field pixels, except for those of dendrites, could not decode the direction. These analyses could be done offline.

The authors very much agree with the spirit of this question - that it is important to ensure that the decoder is being driven by fluorescence signal changes originating from neural sources. We included additional data (which we had collected but not yet analyzed or included in the manuscript) to help make this point conclusively. We have added the following text to the results section to describe control experiments that we performed to make this case more conclusively:

To rule out the possibility that artifacts could be contributing to improve online decode success, we performed a control experiment by running online decode sessions in fields of view without GCaMP expression, but which included auto-fluorescent puncta with similar geometries to neurons and dendrites. Possible artifacts include movement induced motion of the imaging plane or minor differences in background illumination due to stimulus position on screen. We performed three decode sessions in areas with bright endogenous autofluorescence. As anticipated, imaging in these areas provided fluorescent signals that do not modulate. When running the online decoder in these sessions, all three performed decode at chance levels (session 1: 2 conditions, 38 success, 32 fail, 54.3%, binomial test $p=0.55$; session 2: 2 conditions, 65 success, 46 fail, 58.5%, binomial test $p=0.09$; session 3: 14 success, 14 fail, 50%, binomial test $p=1.0$), suggesting that the decode success was not driven by artifacts.

Regarding the first point of this question, we view the contribution of many dimly fluorescing (but potentially signal-containing) pixels as a possibly useful source of information, analogous to multiunit threshold crossings commonly used as a signal source for BMIs (e.g., Pandarinath et al. 2017) and becoming increasingly common for investigations of basic science (e.g., Trautmann et al. 2019). We have also added the following text to the results section to clarify this point:

We note that this online decoder uses raw pixel values and not detected ROIs. As such, it is likely that many pixels can contain modulated signal (even if weakly modulated) despite not obviously being associated with a neural process or soma, particularly in the cases where (1) a neuron may be only very weakly expressing GGaMP, or (2) a neuron with particularly shallow modulation depth, or (3) a pixel contains signals from multiple dendrites or other sources (e.g.: due to the anisotropic focal volume of the 2P laser due to the extended z-axis of the point spread function).

3) How many ROIs with calcium transients were reconstructed by CLARITY? Could any ROI that originated from the superficial layer neuron not be identified? How many cell bodies were

found in layers 2/3 and 5? Although 3D reconstruction of neuronal morphology is improving, the limitations of the current method and how these could be solved should be discussed.

We agree with the reviewer that this is an important question, also asked by reviewer #1. We have expanded the quantification and analysis of CLARITY data and have reported the results above. Please see the response to Reviewer #1, question 7, which also addresses all portions of this question and includes some additional text and figure panels we have added.

The authors should determine whether reaching direction preference was different between different types of neurons.

The reviewer raises an interesting question regarding the contribution of different cell types to the population activity governing motor control. We have no *a-priori* expectation that movements for different reach directions are driven by distinct populations of morphologically distinct cell classes. In order to observe this, different muscles would need to be driven by distinct cell types. Instead, we believe that it is more likely that all muscles are primarily driven by corticospinal tract (CST) cells and other projection neurons which serve as the output of motor cortex. As such, we would expect to see a mix of tuning for different reach directions in cells that project to different muscles, though this is outside of the scope of what we can answer with the current data. The imaging methods used in this study, however, do not have the ability to resolve which target muscle a particular cell innervates as this would require tracing cells from motor cortex through spinal cord, to the muscles. This is not presently possible in NHP unless another genetic access technique such as retrograde tracing is used in parallel with functional imaging. While this would constitute an interesting direction for future research, we believe our work represents an important foundation for this line of inquiry.

In line 217. The authors should show the image and fluorescent signals of neurons at a depth of around 550 μm from the cortical surface.

Thank you for this very interesting question. We originally included this image in response to feedback from colleagues who felt that the choice of silicone imaging window attenuated the fluorescence signal and prevented imaging neurons deeper than 200 μm from the surface. We found that in the animals where we saw dense somatic expression, we did not observe functional modulation of the GCaMP fluorescence signal, but were able to image neuron somas down to a depth of 550 μm (albeit with lower signal-to-noise ratio (SNR) and attenuated signal at this depth). We included this to make clear that the maximum imaging depth for this implant design is not merely limited to $\sim 200 \mu\text{m}$.

In the injection sites where we were able to perform successful functional imaging, we observed dense expression in superficial layers with many dendritic processes, but fewer somatic signals and did not observe somatic signals deeper than 250 μm . As such, we do not have data to assess the SNR during functional imaging 550 μm deep. This is not a fundamental issue or limitation of the imaging technique or preparation, but a function of the pattern of

GCaMP expression and variability that we observed. Regardless, we agree that this may be confusing to readers and changed the wording in the results section accordingly:

2P imaging fields of view included all fluorescent signals present in superficial cortical layers, typically 150-350 μm deep, including somatic signals in superficial cortical layers (Fig. 4E, F) as well as calcium signals from GCaMP-expressing dendrites (Fig. 4G).

The numbers and periods of imaging sessions were not clear. For example, “multiple times” over “several” experimental sessions in line 238 should be replaced with concrete figures.

Thank you, we have now replaced this line with the following:

Here, we demonstrate using vascular fiducial markers to return to the same neurons and neuronal processes over seven sessions spanning 13 days (Supplementary Fig. 6), and this approach should in principle work for considerably longer timescales.

We have also more clearly indicated the number of sessions both accompanying the ROI analysis:

We identified ROIs within each of the 36 imaging datasets using Suite2P...

And accompanying the oBCI decoding analysis:

Decoder accuracy for single sessions remained above chance for hundreds of trials (Fig. 6e), and aggregated decoder performance for all sessions was significantly above chance ($p = 1.68e-11$ for the two target task, 27 sessions spanning 16 days, rank-sum test[XS1] ; $p = 4.11e-5$ for the four target task, 9 sessions spanning 8 days, rank-sum test; Fig. 6f)

The authors mentioned that M1 neurons did not show preparatory activity in line 258. It would be better to show calcium transients in both PM and M1 neurons during the preparatory period.

The injection sites providing the functional modulation and majority of the data used in this study were in M1, and not far enough anterior to be located in PMd. In monkeys S, W, and X we injected constructs in PMd, but in the constructs injected in those areas, we did not observe physiologically healthy levels of GCaMP expression and instead observed static fluorescence in monkeys W, and X, and no expression in monkey S. The imaging implant does afford simultaneous access to PMd and M1, but we chose to focus on M1 for the recordings presented here. As such, we are unfortunately unable to show delay period activity in PMd from these recordings, though we agree that a direct comparison of neural dynamics between these brain regions is a well-motivated question. We added the following line to the results section to help clarify this point:

The decode results presented here were primarily imaged at injection sites for the AAV1-CaMKIIa-GCaMP6f construct, located in M1.

We also discuss some possibilities related to preparatory activity in our response to reviewer 1 for question 2.

The authors stated “Behavioral sessions lasted between 90-180 minutes and no bleaching of GCaMP6f was observed over time” (in line 573). Was imaging performed between 90-180 min and were all of the image data used?

Thank you, we've added the following to the methods section to clarify the time course of imaging during an experiment:

Typical sessions in which the subject was working in the imaging rig lasted between 90-180 minutes. Imaging was performed during decode blocks, typically lasting between 5-30 minutes, and different decoding sessions were performed in different fields of view during a single session in order to explore different injection sites and depths. We typically performed imaging (while the monkey was at rest and not performing the task) for several minutes in between decode blocks to localize distinct fields of view with neural features, or to localize an imaging field of view based on the surface vasculature. We did not observe photobleaching over the course of a decode block or across blocks within a session in the same region.

In Figure 4J, K, the range of the go cue onset should be added.

We thank the reviewer for pointing out where we could have been clearer. These data were aligned to go cue onset so there is no range present after alignment, despite the variable delay period. For clarity, we have added the delay period range to this Fig. 6B:

In Figure 4K, please show the amplitude of the Y axis,

Thank you for pointing this out. We have revised this figure substantially and the new panels in Fig. 5D show scale bars to interpret the magnitude of signal changes.

Show these ROI structures in the field of view.

In the revised Fig. 5, we found that it was more clear to view the ROI structure maps with the color scale independently from the contrast-enhanced mean intensity image, and found that it is difficult to visualize tuning when overlaid with the many unmodulated but autofluorescence objects within a field of view.

Figure 5G is a very important panel showing the time course of decoder accuracy. However, the explanation is unclear and requires further clarification. Is time 0 the movement onset or the onset of Tint? Was the Y-axis value at each time point calculated from each frame or from accumulated frames?

Thank you, we agree this deserves additional clarification, which we have now added to the results section.

Decoder confidence could also be assessed as a function of time within a trial by assessing the difference in decoder score for integrated frames over the course of single trials. These scores were calculated by averaging the six frames prior to each assessed timepoint and calculating the decode score for this frame relative to the training data for each condition. These signals diverged around the time of movement onset ($t=0$) and remained separated through the duration of movement (Fig. 6G, larger decoder value = more dissimilar from training data). We also compared this online MMSE decoder with an offline decoder using multi-class support vector machine classification with 5-fold cross validation to understand the timing of signal divergence from these signals and to benchmark our online performance against an optimal linear classifier. We observed offline decode performance diverge from chance levels ~250 ms after the go cue presentation (Fig. 6H) and ~100ms prior to movement onset (Fig. 6I), reaching a mean decode accuracy of 61.3% 200 ms after movement onset. As anticipated, this is somewhat better performance than the mean online decode performance for 4-condition decode (45.4%, 9 sessions).

Neutralizing antibody assay to seek the AAV that would not induce the immune response in each animal may be important. However, in the current study, the effectiveness of AAV selection based on this assay was not demonstrated. Was non-standard GCaMP expression related to specific combinations of AAV and marmoset? The authors should tone down the description regarding the neutralizing antibody assay.

Thank you, we agree, and we have toned down this description as suggested. We do not wish to in any way over-state what we have demonstrated, and instead merely wish to point out what we did, what may be an important consideration in future work, and why. We also note that this work was performed in rhesus macaques, where large scale viral screens may be prohibitively expensive or resource intensive. Consequently, any information that helps to optimize the probability of successful expression viral delivery for small n experiments is highly useful. We have moved the figure panels regarding this assay to supplemental materials and have toned down the results:

While these results are consistent with our expectations, further experiments and additional subjects are required to validate the efficacy of this approach. In addition, the impact of immunoreactivity on the longevity of expression at healthy levels and on the success of subsequent additional injections remains to be determined.

In line 468, the authors should also cite Masamizu et al. Nat Neurosci (2014).

Thank you, we have added this citation below this line where we explain the relative advantages of optical methods over electrophysiology for tracking neural populations during learning.

Precise tracking of large populations of neurons using imaging could be leveraged to study learning-related changes at single-cell resolution across longer timescales (e.g. Masamizu et al., 2014).

Recently, Shiming Tang's group in Peking University reported 2p calcium imaging of a large population of neurons in the macaque visual cortex. A discussion of the differences between the imaging techniques used in the current study and those used in Tang's study should be added because many readers might find this information useful.

We agree that it is important to highlight the differences between the technical capabilities and results presented by Shiming Tang and this work. In particular, we believe that the differences between the two approaches derive from the different requirements imposed by motor behavioral tasks vs. visual tasks, and different constraints on the implant design due to the brain region (Motor cortex vs. V1). Recording in the motor cortex requires a different and more complex implant design, which we have clarified this in the main text:

This level of mechanical rigidity in the implant and head stabilization (and associated complexity of the preparation) was not required for stable imaging while the monkey was sitting and viewing images on a screen, as is done when studying visual-processing in V1 for example (Li et al. 2017, Ju et al. 2018, Tang et al. 2020), but was essential when introducing motor behaviors.

...

In principle, fixed-depth rigid windows may be used to restrict tissue motion, as is common in rodent experiments. In our experience, for chronic implants on the dorsal aspect of the skull, the brain may often recede from the window over time rendering the stabilizing pressure of the window ineffective, in contrast with a recent report of recordings from V1.

We think it is important to acknowledge, however, that it is difficult to directly compare the imaging quality during behavior between the two implant designs due to differences in GCaMP expression patterns observed between the two sets of results. We have added additional mention and citations Shiming Tang's groups work throughout the text.

The laser power should be provided.

Thank you for catching this oversight. We have added the following line to the 2P methods section:

Laser power was adjusted as necessary to optimize SNR prior to each recording series and typical values were between 50-150 mW.

How was light shielding performed around the objective?

We have added the following to the methods:

We minimized the impact of background light via several methods:

1. *The monkey and apparatus were in the dark throughout experimental sessions, with the only source of light being the illumination of targets displayed on the screen. Targets were either red or blue (not green) to minimize the likelihood of interfering with GCaMP signals, and LEDs or light sources were taped over.*
2. *The geometry of the tissue stabilizer naturally blocks most residual background light due to the small space between the lens and the wall of the tissue stabilizer.*
3. *If background light was ever found to be an issue, we placed blacked foil around the implant and objective.*

In practice, we found that background light did not present a major challenge during the course of these experiments.

How was the motion artifact corrected? When did the artifact occur during the movement? The width of a dendrite is less than 2 um, so even a few um shift in the dendrite shown in Supplementary Figure S3 would distort the calculation of neural activity. What about the shift along the Z axis?

Our primary focus throughout the setup was in restricting tissue motion and minimizing the need for motion correction. Considerable iteration and engineering went into the final design for the head fixation and tissue stabilization system. We have added Supplementary Fig. 1, included below for convenience, to clarify the design and components of this system and approach. In practice, we found that tissue motion was restricted to the range of ~1-3 pixels, as shown in Supplementary Fig. 2b, c. We achieved robustness to translation of the tissue on this small spatial scale by pre-processing the raw pixels with a mild, approximate Gaussian blur, which is described in the methods. In addition, we added the following text to the methods section to clarify our efforts to ensure that motion artifacts did not affect online decode.

For online decode experiments, we sought to ensure that motion artifacts were not responsible for driving tuning and decode via several approaches:

- 1) *In post-hoc analysis of the data obtained during online decode experiments, we did not observe characteristic patterns of symmetric signal increase / decrease for opposite movements that we might expect if fluorescence modulation were primarily resulting from*

motion. In virtually every field of view imaged, only a sparse subset of neural features exhibited tuning, while the other fluorescent objects with a similar morphology did not. It would be extremely unlikely, if not impossible, to obtain this result if the observed tuning were the result of motion artifacts.

- 2) We performed online control experiments, imaging in areas with autofluorescence but no GCaMP expression and did not decode at above chance levels.*
- 3) For online decode experiments, we blurred the images with gaussian smoothing to minimize the impact of X/Y movement.*
- 4) While Z-axis motion is not directly observable, during initial experimental characterization, we imaged dendrites during behavior at several focal planes, offset in depth in sequences of 5-10 μm . In all cases, we observed consistent tuning regardless of imaging plane position, further indicating that the observed tuning was not the result of z-axis motion artifact (data not shown).*

Supplementary Fig. 2: Image stability during motor behavior. (A) X and Y pixel offsets after image registration. The majority of frames require only one or two pixels worth of shift, illustrating the high degree of image stability during behavior. (B) Histogram of the standard deviation of the 2D distribution of X/Y pixel offsets for all datasets. (C) Histogram of the magnitude of X/Y drift measured between the first 25% of frames and last 25% of frames within single image series.

Reviewer #3 (Remarks to the Author):

Trautmann, O'Shea, Sun and colleagues show a non-human primate calcium imaging system for optical brain-computer interfaces. The abstract emphasizes key, primarily technical, achievements: implant system for (a) chronic 2p imaging access and (b) motion-stabilized 2p access; (c) ability to image activity of deeper neurons in layer 5 via apical dendrites visible from the surface, (d) functional tuning of apical dendrites that can be used to decode arm movements, and (e) online decoding pipeline for images. Overall the work represents impressive technical contributions, though I feel the manuscript needs significant revision to more precisely quantify their results to more precisely support key claims/contributions, and to focus and clarify the presentation.

1. My primary concern is that, given the manuscript's focus on technical achievements/contributions, it is severely lacking in analyses to quantify claims and overall results.

Thank you, and we apologize for not including more analyses as you suggest. As discussed above, we have considerably revised the manuscript and added a great deal more quantification of the functional responses in imaged ROIs, of the oBCI decoder, and of the expression patterns observed in the *ex vivo* CLARITY imaging volume. We believe that this additional analysis and quantification should considerably strengthen the overall message and the scientific claims made here.

Some specific examples:

A: image stability claim (via implant/head-fixation design): The only quantification performed (supplemental fig S3) is for a single example video snippet of unknown length. To claim stability, please provide more documentation of alignment error over time (e.g. a distribution of alignment shifts for multiple days)

We agree that additional analysis and figures are warranted for conclusively demonstrating the reliability of stable imaging and to better quantify potential motion artifact and drift of the imaging field of view over the course of minutes or hours within a session.

We have repeated the analysis that we had previously applied to a single time series across all of the analyzed decode sessions and plotted this in a way to highlight drift over the course of single sessions for 20 example sessions. In addition, we calculated the standard deviation of pixel registration offsets for all acquired time series, and the difference in the mean registration offsets for the first 25% and last 25% of frames for each series. We note that there are some time series with non-zero drift over the course of the session (e.g.: Supplementary Fig. 2, panel A, row 4, column 1). In practice, this represented a minority of sessions and did not present a major challenge during experiments, due to the rigidity of the head fixation system that we developed. These three analyses are plotted in Supplementary Fig. 2, copied below:

Supplementary Fig. 2: Image stability during motor behavior. (A) X and Y pixel offsets after image registration. The majority of frames require only one or two pixels worth of shift, illustrating the high degree of image stability during behavior. (B) Histogram of the standard deviation of the 2D distribution of X/Y pixel offsets for all datasets. (C) Histogram of the magnitude of X/Y drift measured between the first 25% of frames and last 25% of frames within single image series.

B: Registration with anatomical landmarks and across datasets: The manuscript repeatedly makes statements about registration of data across days and different types (e.g. “These vascular features provide reliable landmarks allowing for imaging localization and reliable identification of individual neurons across multiple imaging sessions. These landmarks

facilitated registration with ex vivo CLARITY tissue clearing”) without any quantification of registration accuracy. Specific claims of “within a few tens of microns” for CLARITY registration is made without any evidence to support this. Most importantly, the manuscript includes no methodological detail on how these registrations were performed.

We thank the reviewer for highlighting an area where we were insufficiently detailed in our initial description. In general, it is important to note that there is no ground truth from which to calculate registration accuracy, without introducing exogenous fiducial markers. Instead, what we intend to convey is that by using unique and unambiguous vasculature features as fiducial markers, we are able to identify the same recording regions and imaging fields of view across sessions, and then to identify the same cellular features between 2P imaging and CLARITY imaging.

The Bruker 2P microscope we used cannot store absolute coordinates for its micrometer stages, requiring us to recalibrate the microscope position each session. We performed this position calibration using vascular fiducials which served as consistent landmarks within the imaging chamber. On a given session, we would identify consistent large vasculature features (millimeter scale), then follow these to a specific vascular branch point, where we zeroed the stage coordinates of the microscope. From there, it was possible to navigate using stage coordinates to injection sites based on coordinates alone, though we were also able to use small capillary features to ensure that we were targeting a specific field of view. We have attempted to make this clearer in the text and in the revised Fig. 4, both copied below:

Using vascular features identified via widefield imaging as fiducial landmarks, we located the sites of GCaMP expression with 2P imaging and observed neurons expressing GCaMP (Fig. 4E-G). These vascular features provided reliable landmarks across sessions, allowing for relative localization of the imaging field of view within the imaging chamber and subsequent identification of individual neurons across multiple imaging sessions. These landmarks also facilitated identification of neurons in ex vivo imaging using CLARITY (Fig. 4D and shown later in Figs. 7, 8).

Fig. 4 - Multiscale, multi-modal imaging. (A) Imaging chamber with stabilizer in place under ambient illumination. (B) Cortical surface imaged using widefield (1P) imaging. (C) zoomed in region highlighted in yellow box in B. Vascular landmark used to calibrate microscope stage positions indicated with blue arrow. (D) Further zoomed widefield image showing microvascular features used for localizing 2P FOVs and aligning 2P imaging with CLARITY marked with magenta arrows. (E) 2P image acquired from the region highlighted in (D) showing consistent landmarks. (F) CLARITY volume from the same FOV in D and E using anti-GCaMP antibody labelling (vasculature in white).

As discussed in our response to Reviewer 1, we feel that it is important to highlight a distinction between measurement precision and accuracy in this context. The brain is soft and can move from day to day, such that micron-scale positioning *accuracy* in the experimental setup cannot effectively ensure micron-scale *precision* when localizing neural features across sessions. While our system did afford reasonable levels of positional repeatability, we instead relied upon our ability to identify static constellations of vasculature (and the same neural features within) to localize imaging fields of view.

In practice, we found that this approach was straightforward, and navigating throughout the addressable volume of cortex to find a specific field of view was not a challenging task. This method enables alignment of imaging fields of view across sessions at micron scale, where the same neural features are unambiguously identifiable through those multiple sessions. This method, however, does not permit a direct measurement of the absolute

positioning of the halo system across sessions and we do not have a ground-truth measurement for day to day positioning error.

In brief, when navigating to a target imaging region, large and recognizable vascular features are identified first, and used to guide the translation of the microscope objective to specific vascular branch points, which serve as landmarks from which to identify specific injection sites. This procedure made it possible to image the same constellation of neurons and dendrites across multiple sessions, identifying the same features across sessions. The microscope translation stage has 1 μm precision, so our ability to align a given field of view to micron precision was not limited by mechanical precision. What we intended to convey by localizing an imaging field of view to “within a few tens of microns” was simply to indicate that the same vascular and neural features were identifiable, and we could place them with micron precision within a given field of view.

We used the same approach to register CLARITY imaging volumes with the 2P imaging. The same set of vascular features (spanning scales from multiple mm down to 10 microns) are visible in 2P and in CLARITY imaging. We registered the two forms of imaging based on these vascular features and identified corresponding neural features accordingly. This procedure is described in detail in Fig. 4 above. It is not clear, however, how one would calculate an absolute metric for localization accuracy without introducing exogenous fiducial markers to the live tissue and ensuring that these do not move prior to CLARITY imaging.

C: More through quantification of neural signals: The authors show examples of movement-tuned ROIs but do not include any information on the total # of ROIs measured, how many were tuned, etc. Given that they report (but don't show) many cells with filled nuclei that did not have behaviorally-modulated fluorescence, these details seem particularly important. The intro and discussion also repeatedly emphasize how useful Ca imaging is for recording large populations, but the data shown do not speak to this point at all.

Thank you for raising these important points regarding the efficacy, consistency, and issues with expression that we observed in this study. We believe that we have addressed these questions with additional analyses and clarification, which we summarize below:

First, we have completely redone our analysis of functional signals using Suite2P to identify ROIs, which enabled us to quantitatively assess functional responses across all 36 imaging datasets. to quantify the number of recorded ROIs and to assess the responsiveness and direction tuning of these responses. We have added the following text to the manuscript in the results section:

We identified ROIs within each of the 36 imaging datasets using Suite2P, an open-source tool which does not make assumptions about the shape of ROIs and can readily identify elongated structures belonging to neural processes (Pachitariu et al. 2017). Suite2P identified 596 ± 126 ROIs (mean \pm s.d.) across the 36 imaging datasets (total 21451 ROIs) Functional modulation of GCaMP-expressing neuronal processes was observed in gyral M1 (Fig. 5A) during motor behaviors and assessed using an instructed-delay reach task, which elicited rapid, straight reaches to each of four radially positioned targets (Fig. 5B). The

majority of identified ROIs exhibited a response just prior to or during arm movement (65.9%, rank-sum test, $p < 0.01$). This modulation time-locked to movement was readily visible in single trial raster plots (Supplementary Fig. 8). In contrast, relatively few ROIs exhibited responses time-locked to the target cue (1.3%, rank-sum test, $p < 0.01$) or to the go cue (1.7%, rank-sum test, $p < 0.01$). During movement, 31.0% of ROIs exhibited direction-tuned responses (ANOVA, $p < 0.01$).

We identified a subset of ROIs corresponding to putative dendritic or axonal processes by using the shape of the ROIs (see Methods). This identified 20.3% of total ROI, or 121 ± 69 ROIs per imaging session (Supplementary Fig. 9) with an elongated shape. This subset of ROIs also exhibited significant responses to movement (81.3%, rank-sum test, $p < 0.01$); directionally-tuned movement responses were observed in 50.3% of putative dendritic/axonal ROIs (ANOVA, $p < 0.01$). Few responses to the target cue (2.8%, rank-sum test, $p < 0.01$) or the go cue (1.5%, rank-sum test, $p < 0.01$) were observed.

Second, we have added additional figures and analysis to demonstrate the reliability of single trial responses across large neural populations. This is readily visible in Supplementary Fig. 8, copied below for convenience. We have also demonstrated that single trial neural trajectories constructed from optical responses exhibit dynamical structure similar to that observed with multielectrode arrays, which are commonly used to infer neural state from large populations. These are presented visually in Fig. 5 and Supplementary Fig. 10, and described in the results as:

Using demixing-PCA⁵⁷ and targeted dimensionality reduction (TDR)⁵⁸, we identified 3 dimensions of neural activity (weighted linear combinations of direction-tuned ROIs) exhibiting a condition-independent signal⁴⁸ and reach-direction dependent signals (see Methods). Single trial neural trajectories exhibited peri-movement modulation consistent with what we and others have previously using firing rates measured from M1 using electrode arrays^{49,50} (Fig. 5e, Supplementary Fig. 10e,j). Within the TDR subspace, single trial trajectories also separated according to reach direction, consistent with the direction tuning observed in individual ROIs. Using PCA on the raw imaging data, we find that 6-8 dimensions capture the majority of variance during this task (Supplementary Fig. 12).

Supplementary Fig. 8: Single trial rasters of all ROIs showing significant modulation during movement (see Methods). (A) Each row represents the activity of a single ROI over time as DF/F relative to baseline. The rows of the raster are sorted vertically using rastermap (<https://github.com/MouseLand/rastermap>), a multi-dimensional embedding algorithm that groups similar ROIs close together. Below the raster are the hand velocity in X and Y, including reaches from the center to the target and the reach back to the center position. Short breaks in the hand trace represent single unknown hand position samples; time runs continuously in this plot. (B) Same as (A) but for a second imaging session at a different field of view.

Third, we have analyzed all additional imaging datasets to provide a complete analysis of the functional responses. Supplementary Fig. 6 (copied below), in which a field of view with many ROIs is revisited over multiple days, helps to underscore the richness of the data that we report here:

Fourth, the issue with fluorescence but non-modulated cells is a particularly important one. Anecdotally, from our conversations, including at conferences, it appears that other labs have experienced similar challenges when expressing GCaMP in rhesus macaques, whereby large numbers of cells do not modulate and sometimes feature what appear to be filled nuclei. While we do not have an answer to why this challenge occurs, we agree that it is important to highlight as an issue facing the field moving forward. We do note, however, that we do not anticipate that all cells within a given recording volume in motor cortex will respond during an arm reaching task. As such, we anticipate that some fraction of neurons will be expressing GCaMP at healthy levels and could, in principle, exhibit dF/F responses, but aren't responsive during the experimental task. Disentangling these two types of non-modulated responses is beyond the scope of this paper but represents important future work.

Lastly, we have endeavored to present our results within the appropriate context. We observed somewhat sparser somatic expression relative to mouse imaging and to the results presented by Li and colleagues (Li et al. 2017) and acknowledge that there are further opportunities to improve upon the viruses, genetic access tools, and virus delivery protocol. While there are still opportunities to improve the overall imaging quality by obtaining denser somatic expression in superficial layers, we believe that the set of methodological advances reported in this manuscript serve to motivate and help enable these further experiments to refine the expression quality.

Supplementary Fig. 6: Consistent tuning across experimental sessions. Each pair of images corresponds to an individual imaging session, all of which were collected at highly overlapping fields of view. Left panels show enhanced mean images; right panels show ROI tuning maps. Individual dendritic / axonal processes can be identified using a combination of their idiosyncratic morphology and their location relative to the consistent shadow created by superficial vasculature. Two examples are identified with yellow and green stars in each image where they are visible, where the tuning maps show preferred directions of up (yellow) and left (green) in each session.

D: Chronic access claim: The claim of chronic imaging is supported by a single example neuron image at 3 time-points, without any quantification (e.g. statistical tests). This is quite minimal support for this claim. I would also recommend more precisely stating these claims to specify what is meant by "chronic" (# of days, ability to image vs. ability to image the same neurons, etc.)

We thank the reviewer for identifying a point which deserves additional clarification. We'd like to distinguish a difference between "chronic imaging", which here we intend to refer to the ability to record from an implant across multiple weeks or months, as opposed to the ability to track the same cells across multiple recording sessions. This definition is consistent with the terminology used to describe Utah array and other electrode implants. In the revised manuscript, we have made this clearer by distinguishing our chronic access demonstrations from claims regarding the tuning stability of single neurons across sessions.

Regarding this latter point raised by the reviewer regarding tuning stability, we had initially provided several examples of neurons which displayed similar tuning across multiple sessions, with a single example in the main figures and two additional examples in Supplementary Fig. 4 of the original submission. We included these examples more as a proof of concept of our ability to identify the same cells based on their unique morphology and unique vascular features, rather than as well supported scientific claim regarding the stability of tuning, for which we agree that additional data would be necessary.

In the revised figures, we instead show in Supplementary Fig. 6 (copied below) a collection of imaging sessions in which a field of view was revisited multiple times, resulting in a tuning map that displays consistent tuning in recognizable features over time. Certainly, this would be improved by a statistical quantification of tuning consistency across days. However, this would require each ROI to be manually identified across each session, which is made difficult by slightly different Z depths across sessions.

Despite this limitation, we note that our intended definition of chronic access does not depend on this claim. To further clarify this, we have added the following text to the method section to describe the duration of recording in each subject.

2P imaging sessions were collected over sessions spanning the following number of days in the three subjects included here: monkey S, 122 days; monkey W, 30 days; monkey X, 144 days. We note that for all three subjects, there was not a degradation of imaging quality, and imaging data were collected until we had sufficient data within the imageable injection regions. In particular, for monkey W, imaging quality remained excellent but as we did not observe functional tuning in the neurons we imaged, we did not continue to collect data beyond the first month of exploration.

Supplementary Fig. 6: Consistent tuning across experimental sessions. Each pair of images corresponds to an individual imaging session, all of which were collected at highly overlapping fields of view. Left panels show enhanced mean images; right panels show ROI tuning maps. Individual dendritic / axonal processes can be identified using a combination of their idiosyncratic morphology and their location relative to the consistent shadow created by superficial vasculature. Two examples are identified with yellow and green stars in each image where they are visible, where the tuning maps show preferred directions of up (yellow) and left (green) in each session.

2. Substantial portions of the manuscript are devoted to documenting varying viral injections and immunoreactivity characterization of subjects. These data are interesting and valuable to the community. But, as currently presented the authors do not provide sufficient data to prove particular points or make any claims. There are a wide variety of viral injections (AAVs, promoters) used and general statements about what seemed to work vs. not, but do not directly support strong claims or interpretation. I would suggest the authors shift these data into supplementary materials and discussion or revise the analyses to more quantitatively analyze the phenomena they qualitatively describe (e.g. differences in expression across constructs; progression of expression/nuclear filling).

I want to emphasize that I think these data are very valuable for the community, as they contribute to ongoing understanding of how to optimize viral delivery of opsins and fluorescent reporters in primates. But including these data as key focus of the manuscript when they cannot directly support rigorous conclusions has a diluting effect. In some places the authors also appear to make strong claims that are not well-supported by data about the safety and efficacy of their expression protocols (e.g. achieving “robust functional expression...aided by designing a macaque codon-optimized GCaMP which included a nuclear export signal”. This claim is particularly confusing, as the reaching data is stated to primarily come from an injection site that does not use the NES construct. The main data shown using the NES construct, as I understand it, is from supplemental wide-field imaging data in fig. S2).

We thank the reviewer for highlighting this point and for underscoring the difficulty of attaining reliable, quantitative information about the effect of viral injection parameters in macaques. We have revised the discussion of viral selection to reflect our desire to provide information to the community with an appropriate level of caution in interpretation. We believe that the neutralizing antibody assay we performed may be a useful tool in selecting serotypes when pre-existing immune responsivity is expected, as is the case with AAVs and macaques. However, we advise caution that our results are not conclusive. We have also moved the neutralizing antibody assay results to the supplemental figures to avoid diluting the manuscript.

While these results are consistent with our expectations, further experiments and additional subjects are required to validate the efficacy of this approach. In addition, the impact of immunoreactivity on the longevity of expression at healthy levels and on the success of subsequent additional injections remains to be determined.

We have also trimmed the portion of results to simply convey which constructs we injected and added a similar cautionary note regarding the codon optimization and the nuclear export signal modifications:

We note that because we have not performed a systematic comparison, we cannot be certain what effects if any including the primate codon-optimized transgene and nuclear export signal have on GCaMP expression.

3. The paper emphasizes the achievement of closed-loop BCIs with optical signals. However this is both not fundamentally novel (has been done in rodent models by several groups) and the presented experiment is not a truly compelling example of a closed-loop BCI. Given that other online imaging read-out systems have been implemented before, the authors should take effort to highlight any differences/strengths of their approach in order to be able to emphasize this as a key technical contribution. More generally, the latency and significant temporal binning used in the 'online' experiment overall serve to shift the experiment into a discrete read-out that runs in parallel to the animal performing the natural reaching behavioral task, more than an online, closed-loop BCI task. Thus, while certainly a technical feat, it is unclear to me what specifically this experiment demonstrates above and beyond the offline decoding analyses. The authors also state that their work provides "the ability to make closed-loop experiment adjustments based on real-time neural read-outs", but I do not see specific evidence of this in the existing experiment.

We thank the reviewer for highlighting a point that deserves additional clarification. Regarding the lack of novelty for oBCI, we note that the demonstration in rodents using an auditory 1-D BCI (Clancy et al. 2014) used ROI-masks and cell identification, binned data in 200ms bins, and do not report processing latency between bin end and cue update. Here, we present a processing framework capable of processing full frames (without applying ROI masks) within 10-15 ms. This code, which is now publicly available on Github, can be readily used for a range of experiments, including closed-loop continuous control of a multidimensional output.

We agree with the reviewer that the task employed here only loosely demonstrates the closed loop capabilities of this system. While we believe that the hardware and software collectively constitute an important new capability which enables new classes of experiments (including closed loop with low latency), we have tempered our discussion to emphasize low-latency computation on the input images, as opposed to closed-loop control in the task. Some examples of these modifications are copied below:

We demonstrated that signals imaged from dendrites located in superficial layers were modulated during movement, were directionally tuned, and exhibited sufficient signal-to-noise ratio to enable a real-time oBCI using low-latency image processing to decode a monkey's behavior from neural activity to provide low-latency visual feedback and reward.

*This work demonstrates new capabilities for combining emerging neural recording technologies with **low-latency** BCI.*

Real-time stimulus control was implemented by decoding frames acquired from the microscope directly from memory buffers on the acquisition hardware

We have only included the term "closed loop" to discuss potential future applications, which we believe are readily accessible using this system.

4. Overall, I think the paper could benefit from an effort to make it clear what exactly was done across all the animals and for specific data presented. Several things leave me a bit confused. For instance, methods state that immunological assays were performed on 5 monkeys, but then data is only shown for 3. It seems that reaching data and BCI experiments are performed primarily with monkey X? Are other data beyond immunological response included for monkeys S and W?

We agree that it is important to clarify which subjects were used for each of the experiments described in this manuscript and we apologize for not making this clear previously. In response to this suggestion, we worked to clarify the sets of experiments and which monkeys were included in the various components of the manuscript, primarily in the updated / revised Methods section where we include here the added passages included at the appropriate points in the Methods section:

Stable 2P imaging during the reaching task was demonstrated in Subjects S, W, and X. Decode experiments were performed with subject X. Subjects S and W did not exhibit functionally tuned responses due to a lack of GCaMP expression (monkey S) or static non-modulated expression (monkey W). Monkey W did exhibit healthy GCaMP expression at some injection sites, but neurons in those sites were not responsive during the task (data not shown).

Three male rhesus macaques (X, S and W) were used for 2P imaging experiments and a fourth rhesus macaque (L) was used for secondary validation of virus constructs in VI.

Blood samples were collected from four rhesus macaques. Blood draws were either performed at Stanford University (monkeys S, W, X) or at the University of Texas at Austin monkey (L), depending on the location of the monkey.

The reviewer astutely noticed that we erroneously mentioned a fifth subject in the neutralizing antibody assay results. The fifth animal subject, Monkey F at UT Austin, was intended to be used in these experiments but experienced health complications unrelated to the study and sufficient data could not be collected to be included in any analysis. Thus, we have removed mention of this animal.

Regarding the validation of GCaMP expression, in Monkey W, we observed widespread expression, across a range of sites from different viral constructs. Unfortunately, in this subject we tested a wide range of constructs across a broad area of cortex, and the constructs which exhibited healthy somatic expression, where we observed spontaneous neural activity, were outside of the arm-related regions of motor cortex. We observed healthy appearing “halo” shaped cells in L2/3 here as well as sporadic spontaneous modulation, but the neurons did not respond during the reaching task reported here.

In contrast, the constructs expression in the target regions of the shoulder and arm-related motor cortex exhibited static and bright expression with filled nuclei. Here, we did not observe any spontaneous activity and exclusively observed filled nuclei, suggesting unhealthy levels of expression or nuclear intrusion of GCaMP.

Minor/specific comments:

Error in abstract: "...imaging of calcium signals from in macaques..."

Thank you for catching this embarrassing typo.

I would suggest making the statement that rhesus macaques have been important for clinical BCIs (intro 1st paragraph) stronger. Motor BCIs are an example where techniques developed/demonstrated in NHPs directly translated to humans multiple times.

We enthusiastically agree with the reviewer and have update this sentence accordingly:

Investigations using rhesus macaques (Macaca mulatta) have served a vital role in developing clinically-viable BCIs by exploring decoding algorithms and system designs and by advancing our basic scientific understanding of the motor system³⁻¹³.

The logic of the argument in the introduction's second paragraph is not entirely clear to me. The authors argue that optical techniques, though not necessarily clinically translatable, will better inform our understanding of arm movement. This seems like a motivation for studies of warm movement representations via optical imaging in NHPs. But the authors then say they present an optical BCI.

We agree with the reviewer that this paragraph was slightly unclear and that this idea is slightly confusing. A number of recent studies have employed BCI paradigms for studying basic neurobiology (e.g., Sadtler et al 2014, Vyas et al, 2018). Here, we present a set of techniques and methods appropriate for either studying BCI or basic neurobiology and emphasize that advances in either domain may translate to the other. We have attempted to clarify this in the manuscript:

Towards this goal, the tremendous recent expansion of tools for measuring activity from large populations of neurons in non-human model organisms could provide a fertile experimental landscape for exploring the design of next-generation BCIs^{16,17}. Here, we demonstrate two photon (2P) imaging in macaque motor cortex and use this to implement an optical BCI (oBCI).

We have also fleshed out a discussion of many other use cases for the specific ability of imaging superficial apical dendrites arising from deep cortical neurons and imaging input signals along afferent projections in the discussion.

Please clarify the imaging area and implant size. Sizes of 1.2cm and 1.8cm are cited in different places in the manuscript.

Thank you, we have modified this language to make clear that the 18 mm figure represents the area of cortex visible, but a subset of that is accessible with the multiphoton lens:

The imaging chamber design strikes a balance between (1) enabling imaging access to a large volume of tissue (18mm visible, of which a 12 mm diameter region is accessible for 2P imaging down to ~ 1000 um deep using the Nikon 16x 0.8NA lens) using commercially available multiphoton objective lenses and (2) minimizing the implant size.

I am unclear on what specifically the role of supplemental fig. S2 is, particularly because these injections use a different construct (AAV1-CaMKIIa-GCaMP6f vs. AAV-CaMKIIa-NES-GCaMP6f) than the site selected for data from monkey X

As there is somewhat limited data on the range of constructs that have been used successfully in NHP, we included this additional data point as an observation of an additional construct with the targeting peptide that appears to express successfully and is worthy of additional testing and exploration.

Please explain what motivated the specific focus on a single injection site for data from monkey X

We observed the best imaging quality and most functional tuning from injection site 2 in monkey X. As such, since our work here was focusing on establishing a proof of concept for the full imaging pipeline and oBCI, as opposed to comparing the neural dynamics between different brain regions, we chose to focus our recordings where we observed the best signal quality. We note that we did observe that other viruses and injection locations did appear to have functional signals as well, but imaging in these areas was difficult due to idiosyncratic formation of thicker neomembrane tissue occluding those injection sites. However, we did image from multiple FOVs in the vicinity of this injection site so as to survey the nearby cortical surface.

Please specify the time-point (relative to implant date) for fig. 4a.

Fig. 4 - Multiscale, multi-modal imaging. (A) Imaging chamber with stabilizer in place under ambient illumination, approximately two weeks after implant.

I do not understand the value added by the analyses for fig. 4L.

We have updated this analysis somewhat, and the new results are presented in Fig. 5E. The intent behind this analysis was twofold. First, to show neural activity is separable between different reach conditions on individual trials, and second, to establish consistency with electrophysiology results from primate motor cortex. In this sense, it ties in with a growing body of work attempting to understand the computations of motor cortex by studying the dynamics of recorded neural populations recorded there. Here, the top panel, representing the condition-independent signal, is consistent with previously published observations (Kaufman et al. 2016). The bottom two panels use targeted dimensionality reduction (TDR) to visualize

direction-tuned modulation, which provides a single trial readout of movement direction. In addition, these plots also provide information regarding the time course of the neural response in this data on single trials, consistent with the decode results we present in Fig. 6 H, I.

Fig. 5: ... **(E)** Single-trial (thin lines) and trial-averaged (thick lines) population trajectories, projected along condition-independent signal (CIS) dimension and condition-dependent X and Y dimensions (see Methods), color coded by reach direction condition.

How many closed-loop BCI sessions are presented for data used in fig. 5?

We've clarified the number of datasets used for these analyses in the results section:

We processed each imaging session offline using Suite2P¹²⁰ (<https://github.com/MouseLand/suite2p>) and analyzed 36 imaging decode sessions collected on eight days.

While the registration with CLARITY is quite exciting and opens new experiments outlined in the discussion, it is important to acknowledge that this technique specifically requires sacrificing the animal and therefore both increases the number of animals needed and is limited to a single time-point. These limitations are not acknowledged.

Thank you, we changed all descriptions of CLARITY imaging as “post-hoc” to “ex vivo” to make this point clear. In addition, we have added the following to the discussion:

We note, however, that as this method requires sacrificing the animal, there are inherent limitations to the timescale and questions that can be addressed with this approach. For instance, it may be difficult to study the impact of learning on networks using CLARITY.

It is a bit strange to me that the low levels of expression achieved, compared to other animal models is not discussed or even alluded to until the penultimate paragraph of the paper.

We have now added the following sentence to the results section to discuss this important point earlier in the manuscript:

We note that the GCaMP expression we observed here is relatively sparser than typical rodent work or that observed by other groups imaging in NHP VI (e.g., ^{1,2}), though this may result from either differences in injection protocol, virus batch, immune status of the individual subject, or other unaccounted factors.

Behavioral task: please clarify how the target was jittered during the delay (i.e. hopping between the different possible targets? Vs. position jitter around the target location?)

Thank you, we've clarified this in the methods section:

Next, one of 4 radially arranged targets appeared 10 cm from the center and jittered around the target location (5-10 mm std. deviation) during the delay period (randomized period, ranging from 100-700 ms).

Some relevant citations appear to be missing:

- Artificial duras designed for NHP: Chen et al., j Neurosci methods 2002
- Other techniques that achieve highly stable 2p imaging in NHP: Choi et al., IEEE EMBS 2018

Thank you, we have added these citations and have double checked that other references are up to date.

References

1. Arieli, A., Grinvald, A. & Slovin, H. Dural substitute for long-term imaging of cortical activity in behaving monkeys and its clinical implications. *Journal of Neuroscience Methods* 114, 119–133 (2002).
2. Kaufman, M. T., Churchland, M. M., Ryu, S. I. & Shenoy, K. V. Cortical activity in the null space: permitting preparation without movement. *Nature neuroscience* 17, 440–448 (2014).
3. Kaufman, M. T., Seely, J. S., Sussillo, D., Ryu, S. I., Shenoy, K. V., & Churchland, M. M. (2016). The largest response component in the motor cortex reflects movement timing but not movement type. *ENeuro*, 3(4). <https://doi.org/10.1523/ENEURO.0085-16.2016>
4. Li, M., Liu, F., Jiang, H., Lee, T. S. & Tang, S. Long-Term Two-Photon Imaging in Awake Macaque Monkey. *Neuron* 93, 1049–1057.e3 (2017).
5. O'Shea DJ, Trautmann E, Chandrasekaran C, Stavisky S, Kao JC, Sahani M, et al. The need for calcium imaging in nonhuman primates: New motor neuroscience and brain-machine interfaces. *Exp Neurol*. 2016. doi:10.1016/j.expneurol.2016.08.003
6. Seidemann, E., Chen, Y., Bai, Y., Chen, S. C., Mehta, P., Kajs, B. L., ... Zemelman, B. V. (2016). Calcium imaging with genetically encoded indicators in behaving primates. *ELife*, 5(2016JULY), 1–19. <https://doi.org/10.7554/eLife.16178>
7. Shtoyerman, E., Arieli, A., Slovin, H., Vanzetta, I. & Grinvald, A. Long-term optical imaging and spectroscopy reveal mechanisms underlying the intrinsic signal and stability of cortical maps in V1 of behaving monkeys. *Journal of Neuroscience* 20, 8111–8121 (2000).

Reviewer #1 (Remarks to the Author):

I commend the authors for the thorough responses to all my previous comments. I think the manuscript has improved significantly after this revision —and, as I had said in my previous review, their results are very important for the field. I only have a couple minor suggestions:

- Perhaps I've missed this in the text, but what is the angular tolerance/error of the setup? I'm wondering if head/brain rotation with respect to the microscope may impact imaging stability across days: in Suppl Fig 6, there's quite a noticeable difference in tuned ROIs at +6 Days and the previous sessions, as well as between +13 days and the previous sessions. This caught my eye because the mean images seem quite similar, yet the processed ROI tuning maps look different. Do the authors think this could have a neural, behavioural or imaging origin (i.e. perfect alignment with single micron resolution is extremely hard)?

- The authors could consider colouring the markers in Suppl Fig 14 based on offline decoder performance, to have an idea how parameter choice influences decoder accuracy.

Reviewer #2 (Remarks to the Author):

The paper has been much improved; in particular, the presentation of much more raw data is persuasive. However, I still find that the authors have not appropriately answered two of my major concerns.

1) Many ROIs in the same field of view showed very similar activity patterns. For example, it appears that the activity patterns in approximately the upper half of Supplementary Figure 8A probably originated from the same neuron, while those in approximately the lower half of Supplementary Figure 8B probably originated from another neuron (strangely, the activity of these groups looks negatively correlated; could the authors explain this?). If this is the case, although over 100 active ROIs were extracted from each field of view, they originated from several neurons, and the authors cannot conclude that “many neurons” were imaged within a field of view. In the Abstract, the author claims, “we achieved optical access to large populations of deep and superficial cortical neurons”. The most valuable achievement of this paper is that the dendritic activity of large populations of cortical neurons was imaged, not the non-modulated fluorescence. The authors should perform clustering analysis to estimate the number of movement-responsive neurons imaged per field of view. This analysis does not take much labor and does not require reconstruction of the dendritic arbors from the CLARITY experiment.

2) The authors have still not demonstrated that oBCI can be driven by dendritic activity. Although the authors show that the decoded accuracy depended on the pixel number in Supplementary Figure 15, they do not show the decoded accuracy for extracted ROIs putatively corresponding to the dendritic arbors. The authors have added the text “it is likely that many pixels can contain modulated signal (even if weakly modulated) despite not obviously being associated with a neural process or soma” to the Results; however, this claim is contradicted by the authors' claims in the title, “Dendritic calcium signals in rhesus macaque motor cortex drive an optical brain-computer interface”, and the Abstract, “we developed an optical BCI (oBCI) driven by these dendritic signals and successfully decoded movement direction online”. Unless the authors demonstrate that the signal from the dendritic arbors that they extracted is critical for the oBCI, the authors cannot

justifiably conclude that dendritic calcium signals drove the oMBI. The authors' view is that many dimly fluorescing pixels are useful, with these being analogous to multiunit threshold crossings. However, if this is the case, the identification of dendritic signals of individual neurons by two-photon calcium imaging is not advantageous. Offline decoding analysis with the pixels without the ROIs that corresponded to modulated dendrites would not require much labor.

Minor:

What is the definition of movement onset? Is it the time to exceed some of the baseline activity SDs? If so, the preceding neural activity may reflect the initial movement. Please add the XY hand positions in Figure 5e and Supplementary Figure 10e, j.

Reviewer #3 (Remarks to the Author):

Trautmann, O'Shea, Sun and colleagues have made significant revisions to the manuscript that markedly improved the quality and presentation. While the revisions have largely addressed my concerns, there are still places where I feel the manuscript makes claims that are not well supported by their data, and where the presentation could be improved.

1. Viral expression strategy claims:

The results and discussion sections have been modified to make clear the results do not provide firm conclusions about optimal strategies for GCaMP viral expression. But the paper still maintains several statements in the introduction and discussion that are not supported by the paper's data. The authors continue to emphasize the possible importance of the NES construct while presenting functional data that is primarily from an injection site that does not use the NES construct and without any data analysis to compare NES injection sites to non-NES sites. ("we modified existing GCaMP6 constructs to promote cellular transport and promote expression of calcium indicators in the dendrites."; "We designed monkey codon-optimized GCaMP constructs... which may have increased functional expression in apical dendrites...") I do not feel that the addition of a statement that more data is needed is sufficient given that the paper as written does not present any data that can speak to the importance of NES. If the authors wish to retain these claims, they must provide some data to speak to these points. Otherwise the claims should be removed.

The paper also retains statements emphasizing the importance of AAV serotype work for their success: ("We co-optimized AAV serotype selection by immune-profiling individual macaques to improve viral transfection")

2. Quantification of responses and stability across time:

2a. The authors have added more quantification of response distributions, which is a significant improvement. They quantify the fraction of ROIs that show movement-related modulations, and then also further break down that population for ROIs that are identified as axonal/dendritic processes. While I appreciate that the author's ultimate focus in this paper is on the dendritic processes, I am a bit unclear on why this general analysis does not also present results for identified somas. Given the author's reported issues with filled nuclei and lack of functional responses in some cell bodies, this is an important detail to include.

2b. The authors also emphasize the stability of functional responses over time (supplementary

figures 6 and 13). The within-day analyses are nicely quantified, but the across-day results (fig. S6) are not quantified (the authors themselves at one point say they are “qualitative”) and are not presented in a format that easily facilitates cross-day comparisons. Because the fields of view across days are not spatially registered and identified ROIs vary across days, it is difficult to visually discern stability across days for a given ROI in this figure format. While I appreciate the author’s point (in the response to reviewers) that only focusing on a few example ROIs gives a limited data snapshot, this figure alone does not allow any quantitative or substantive comparison of ROIs across days. Given this limited analysis, I am therefore not clear how well the statement “We...return to the same neurons and observe consistent direction tuning in neuronal processes” is defended.

3. Methodological presentations:

I thank the authors for including more details on methods and what was done. The manuscript has significantly improved in this regard. A few smaller points, however, should still be addressed.

3a. Which monkey and injection site(s) are used should be presented consistently throughout the results/figures. This information is largely present, but not consistently.

3b. For multi-modal registration between CLARITY and functional imaging, I am still unclear on the exact way registration was performed. I follow the logic of figure 7, but how are the overlapping/consistent processes between images determined? For instance, how are the yellow vasculature in panel d and f, and the dendritic process in the white box for panel d, e, and f identified? The way the text is written seems to imply this is partly manual visual inspection. Is that the case? If not, some information on the algorithmic approaches should be provided. If it is being done manually, some discussion of what limits applying algorithmic approaches for registration are required.

Minor/specific:

“We developed a highly rigid three-point head restraint system... restricted head motion to just a few microns” – How did you confirm that head motion is the only source of motion error? Your test measures registration error across frames but does not identify the source of this motion. I would revise the sentence to make the distinction clear.

The Choi et al. paper presents imaging data from S1, not V1 as your citation implies. This paper is also relevant for several of the technical imaging details discussed in the paper, where it is not being referenced, such as imaging stability and expression levels.

“We demonstrated that signals imaged from dendrites... exhibited sufficiently high signal-to-noise ratio.” Sufficiently high SNR for what?

I am unclear on why the methods paragraph “For online decode experiments...” is included in the section on 2p imaging methods, rather than the section for oBCI decoding.

Point by point responses to reviewer comments

Key:

1. Reviewer comments are highlighted in blue,
2. Author responses are in black.
3. *Text included in the revised manuscript is italicized and indented*

REVIEWER COMMENTS

Reviewer #1 (Remarks to the Author):

I commend the authors for the thorough responses to all my previous comments. I think the manuscript has improved significantly after this revision —and, as I had said in my previous review, their results are very important for the field. I only have a couple minor suggestions:

Perhaps I've missed this in the text, but what is the angular tolerance/error of the setup? I'm wondering if head/brain rotation with respect to the microscope may impact imaging stability across days: in Suppl Fig 6, there's quite a noticeable difference in tuned ROIs at +6 Days and the previous sessions, as well as between +13 days and the previous sessions. This caught my eye because the mean images seem quite similar, yet the processed ROI tuning maps look different. Do the authors think this could have a neural, behavioural or imaging origin (i.e. perfect alignment with single micron resolution is extremely hard)?

We thank the reviewer for their generous comments. We note that these data were not collected with the intention of maximizing the similarity of fields of view across sessions, instead optimizing for maximizing SNR and signal for the oBCI decoder. We have further tempered any discussion and removed the qualitative claims of tuning consistency. While we are unable to ascertain the relative contributions of multiple sources of nonstationarity, we feel that a description of this challenge could be helpful for readers, and have added the following to the methods section:

The positioning of the imaging plane for each decode session was chosen to maximize the number of modulated processes observed in the field of view, and we did not attempt to optimize correspondence of the imaging field of view across sessions or days. Despite this, we note that we did record from the same fields of view showing some, but not complete correspondence between observed neural features. We believe that the session-to-session differences in mean image, and ROI tuning maps could arise from a combination of: 1) mechanical precision and orientation of the imaging plane, 2) changes in reactive tissue growth on the surface of the brain (which can alter imaging

performance in a subtle and inhomogenous manner across the imaging field), and 3) changes in viral expression over time.

With respect to the reviewer's more specific question regarding angular positioning accuracy, the angular precision of the orbital nosepiece was difficult to characterize for the same reason that we were unable to use the microscope stages to implement absolute positioning: After power cycling the system, the encoder positions are lost and need to be recalibrated from a reference location. As such, this made it difficult to characterize the angular repeatability for positioning the objective lens. As the reviewer noted, this could change the precise orientation of the imaging plane, leading to changes in both the mean images and processed ROI tuning maps.

The authors could consider colouring the markers in Suppl Fig 14 based on offline decoder performance, to have an idea how parameter choice influences decoder accuracy.

This is a great idea. We've replaced this figure with the following:

Reviewer #2 (Remarks to the Author)

The paper has been much improved; in particular, the presentation of much more raw data is persuasive. However, I still find that the authors have not appropriately answered two of my major concerns.

1) Many ROIs in the same field of view showed very similar activity patterns. For example, it appears that the activity patterns in approximately the upper half of Supplementary Figure 8A probably originated from the same neuron, while those in approximately the lower half of Supplementary Figure 8B probably originated from another neuron (strangely, the activity of these groups looks negatively correlated; could the authors explain this?). If this is the case, although over 100 active ROIs were extracted from each field of view, they originated from several neurons, and the authors cannot conclude that “many neurons” were imaged within a field of view. In the Abstract, the author claims, “we achieved optical access to large populations of deep and superficial cortical neurons”. The most valuable achievement of this paper is that the dendritic activity of large populations of cortical neurons was imaged, not the non-modulated fluorescence. The authors should perform clustering analysis to estimate the number of movement-responsive neurons imaged per field of view. This analysis does not take much labor and does not require reconstruction of the dendritic arbors from the CLARITY experiment.

We thank the reviewer for this keen observation, and for the suggested analyses. We have performed considerable additional analysis to address the question of how many neurons or distinct dendritic arbors are present within the imaging fields of view presented in this data. We first present the clustering analysis requested by the reviewer which is consistent with our view that the ROIs do not arise from a small number of arbors split into many ROIs, which has been incorporated into the manuscript. We then provide additional detail to explain the similar activity patterns the reviewer noticed, and why we think these patterns are to be expected given what is known about motor cortical neuronal responses.

We first perform clustering directly in the space of response profiles. We first cluster the responses using a noise-robust density-based clustering algorithm (DBSCAN). This algorithm automatically determines the appropriate number of clusters using distances in feature space. We can then visualize the results of clustering (in the full-d neural response space) by presenting each cluster as a point in a 2-dimensional embedding space provided by tSNE, as shown directly below.

X20170802_018 : 1 clusters

For this particular dataset, the responses form a smooth continuum comprising 1 cluster (blue) as well as several ROIs automatically marked as “noise” (red) due to their large distances from other ROIs in the full-d space.

We repeated this analysis for all 36 datasets. For every dataset, DBSCAN identified a single continuous cluster (see figure directly below), suggesting that the responses across ROIs form a smooth continuum rather than discrete clusters. Critically, we of course cannot verify directly that every ROI comes from a unique neuron. Certainly, some neural processes may be artificially separated into more than one ROI based on spatial proximity, minor differences in the estimated response profiles, or even due to compartmentally segregated response differences within the dendritic arbor. *However, it is clear from the data that our imaging sampled from many neurons rather than 2 or 3 specific neurons many times.*

We have chosen to include this particular analysis for the two datasets plotted in Supp Fig 8A below the raster plots, where we agree it would be helpful to assuage reader concerns that arise. We also have added this note to the clustering approach in the methods section of the paper, as follows:

We next performed a clustering analysis on individual functional ROI responses to check whether automatic ROI detection approach identified distinct neuronal signals rather than signals from a small set of individual neurons split into many separate ROIs. First, we assembled for each ROI a vector of trial-averaged responses aligned to movement onset for each reaching direction. These response vectors were then clustered using

DBSCAN, a non-parametric density-based clustering algorithm. DBSCAN which looks for clusters of points (ROIs) which are packed more closely together in feature space (have highly similar response profiles during reaching). DBSCAN can automatically determine the number of clusters present in the data. For all 36 datasets, DBSCAN identified only a single cluster including the vast majority of ROI responses, excluding a small fraction of “noise” ROIs that were highly dissimilar to all of the other ROIs and to each other. This indicates that in our imaging datasets, ROI responses form a continuum in the space of peri-movement responses, rather than distinct, separable clusters, that would be expected if the signals originated from the extended arbors of a small number of neurons. We also directly verified this non-clustered continuum in response space by using a t-SNE visualization (Supplementary Fig. 8d,e).

We would next like to highlight several established results from prior primate physiology work, which explain why we believe the similarity of the activity patterns are to be expected from prior work on motor cortical responses. First, Kaufman et al. (2016) showed that motor cortical responses during reaching in primates exhibit a condition invariant signal that dominates the total variance in the data. The first principle component of the variance captures considerable variance, while the condition-specific responses are smaller signals riding on top of this larger signal. Second, many groups have consistently found that neural activity in the motor cortex exhibits low-dimensional structure for reaching tasks similar to those used in this study (e.g. Yu *et al.*, 2009; Pandarinath *et al.*, 2018; Degenhart, Bishop and Oby, 2020; Gallego *et al.*, 2020). Based on these two known properties of neural responses, we naturally expect to observe a large component of variance in individual neurons (i.e., a global increase in firing rate) that does not vary across reach conditions, and strong correlations between distinct neurons or dendritic arbors. *These findings are consistent with the data that we report here.*

As presented in Supplemental Figure 8, the responses of many ROIs do look roughly similar. This is partially due to the dominant structure of neural firing in the motor cortex, and partly due to the presentation format. If we focus on the top half of rasters highlighted by the reviewer:

We can then plot the responses of these ROIs in a different format. Specifically, we compute the trial-averaged responses of these ROIs for reach of the four reach directions:

Indeed, nearly all of these ROIs exhibit a downward deflection just before movement begins. The presence of neurons with this response profile has been previously described in (Chandrasekaran *et al.*, 2017) as “decreased” neurons. These responses are clustered at the top of the raster plot due to the optimization within Rastermap (i.e., Rastermap rearranges rows, so the row number is not necessarily related to the signal’s origin in space.) These selected ROIs exhibit a diverse set of responses during movement. We also expect that some of the true heterogeneity in these responses is diminished due to the temporal smoothing of the calcium reporter. As an aside, it is possible that these decreasing responses may be overrepresented in these particular datasets relative to their proportion in cortex (~14% reported in Chandrasekaran *et al.*, 2017). However, we note that many of the datasets we collected exhibit a much lower proportion of decreasing response profiles. There may be interesting characteristics of the response properties exhibited by superficial cortical processes, which may provide interesting avenues for future research.

If we add in the remaining ROIs in the raster (so that all those with significant responses are included):

We observe the full diversity of responses present, which are generally time-locked to movement. Adding in the excluded ROIs with non-significant responses (shown below in gray) fills in the gap between these increasing and decreasing response profiles, forming a continuum with no obvious clustering:

Indeed, the correlations (both positive and negative) among these responses at coarse timescales (hundreds of ms) relative to behavior are primarily due to the engagement of this entire brain region during the production of arm movements. Indeed, neural activity in the macaque motor cortex during a reaching task is dominated by generation of the arm movement, especially in a well-behaved monkey that remains otherwise still during the recording session. *As mentioned earlier in this response, this is consistent with many prior reports describing low-dimensional structure in motor cortical responses during reaching.*

This effect is especially visible in Supp. Fig 8, because our goal was to provide the reviewers with a sense of the raw data, as requested in the previous round of review. We selected a wide time region that would demonstrate the single trial activity over many reaches. Our goal was to highlight how readily visible the movement-locked activity is in the ROIs, even in single trials. The ROIs in the Supp. Fig 8 rasters are selected based on having statistically significant movement responses and then sorted along the y axis using the tool Rastermap [<https://github.com/MouseLand/rastermap>] further highlight this structure by optimizing the order of the 1d embedding so as to minimize the differences between rows.

As an illustration, below we show data from thousands of individual neurons imaged in larval zebrafish are presented in the documentation for the rastermap package. While the time axis here is $\sim 15x$ wider, overall there is a clear banding structure present, in which hundreds to thousands of neurons exhibit very similar responses. This is presumably due to the shared drive from visual inputs, the shared representation of and contribution to motor output, and the very zoomed-out view of these neural responses presented in the heatmap:

(from <https://github.com/MouseLand/rastermap/blob/master/README.md>)

To bolster this point, we present (unpublished) spiking data recorded in macaque motor cortex during a similar reaching task. Here we recorded electrical spiking activity from well isolated single neurons using a Neuropixels probe inserted acutely in the cortex. Using Kilosort2 to sort the spikes detected across the probe channels into individual neurons, we can then use the spike triggered electrical multi-channel spiking waveform of each neuron to determine its location along the probe. Using these electrical waveforms and statistics of each neuron's spiking (e.g. the inter-spike interval distribution), we can be extremely confident that these spikes come from hundreds of individual neurons. The left panel in the image below shows the mean waveform for each sorted neuron plotted at its estimated centroid location along the probe shank.

We can then take the recorded spike trains from these cells and simulate the calcium-dependent fluorescence that would be emitted (above right). For this, we follow the same process laid out in (Sun *et al.*, 2017)). Briefly, we convolve the spike train with a double exponential filter tuned to emulate GCaMP6f Ca^{2+} binding in response to spiking, and then convert this to fluorescence using a weakly saturating transformation. Given these traces, we can repeat the steps of our original analysis, selecting those neurons with a statistically significant movement response, ordering the neurons using Rastermap, and then presenting the data in a similar format:

This raster comprising 310 individual neurons shows banded structure time-locked with movement. While this raster is certainly not identical to those in Supplementary Figure 8, *our intent here is simply to demonstrate that the presence of strong responses in motor cortex time-locked to movement can produce this kind of similarity in a raster plot, even when we are highly confident that many individual neurons were present.*

2) The authors have still not demonstrated that oBCI can be driven by dendritic activity. Although the authors show that the decoded accuracy depended on the pixel number in Supplementary Figure 15, they do not show the decoded accuracy for extracted ROIs putatively corresponding to the dendritic arbors. The authors have added the text “it is likely that many pixels can contain modulated signal (even if weakly modulated) despite not obviously being associated with a neural process or soma” to the Results; however, this claim is contradicted by the authors’ claims in the title, “Dendritic calcium signals in rhesus macaque motor cortex drive an optical brain-computer interface”, and the Abstract, “we developed an optical BCI (oBCI) driven by these dendritic signals and successfully decoded movement direction online”. Unless the authors demonstrate that the signal from the dendritic arbors that they extracted is critical for the oBCI, the authors cannot justifiably conclude that dendritic calcium signals drove the oBCI. The authors’ view is that many dimly fluorescing pixels are useful, with these being analogous to multiunit threshold crossings. However, if this is the case, the identification of dendritic signals of individual neurons by two-photon calcium imaging is not advantageous. Offline decoding analysis with the pixels without the ROIs that corresponded to modulated dendrites would not require much labor.

Thank you for this excellent suggestion. We agree with the reviewer that there are considerable additional opportunities to more deeply explore the data beyond what we had initially presented. We have performed additional analyses to explore the degree to which dendritic signals contributed to the online, and subsequent offline decoding performance. These analyses are now presented in Fig. 7 of the main manuscript and copied here (see below). We did not intend to convey that the online decoder was exclusively driven by dendritic signals, and instead intended to make the point that these signals contributed meaningfully to the online decoder.

We used the suite2P package (<https://github.com/MouseLand/suite2p>) to identify regions of interest (ROIs) corresponding to putative dendritic signals. We compared the range of each pixel's value across reach conditions (i.e., directional tuning) in the training data set used for online decode and found that dendritic pixels contained greater cross-condition range than non-dendritic pixels in the training data (Fig. 7d). We compared the variance of each pixel across all timepoints and across reach directions between dendritic and non-dendritic pixels and similarly found greater variance within dendritic ROIs than for non-dendritic pixels (Fig. 7e). Only a small fraction of the total pixels in each time series were associated with a dendrite (Fig. 7f). Despite this, decode performance remains high when only using dendritic pixels vs. performing decode with all pixels (Fig. 7g). Decoding using a randomized subset of non-dendritic pixels, with the number of pixels matched to the number of dendritic pixels, yields much poorer decode performance than using dendritic pixels (Fig. 7h). Lastly, sessions with a higher fraction of dendritic pixels generally tended to have higher offline decode performance. Taken together, this suggests that dendritic signals contributed important signals to the online decoder and are sufficient to explain offline decode performance.

Fig. 7 - Dendritic signals drive online decode. **(a)** Mean image for example imaging session, **(b)** Dendrite ROI pixel mask for example in **(a)**. **(c)** Peak-to-peak pixel signal range across four reach directions for the online imaging decoder training data for example in **a)**. Large values indicate large modulation and higher variability across different reach directions. **(d)** Comparison of distributions of pixel peak-to-peak range between dendritic pixels and non-dendritic pixels. **(e)** Comparison of distributions of standard deviation of pixel df/f value across all timepoints and across reach directions between dendritic pixels and non-dendritic pixels. **(f)** Histogram of percentage of pixels inside a dendritic ROI for all sessions, monkey X. **(g)** Comparison of offline decode performance using dendritic pixels only (ordinate) vs. all pixels (abscissa). **(h)** Comparison of offline decode performance using dendritic pixels only (ordinate) vs. a random selection of the same number of pixels as those within dendritic ROIs. **(i)** Decode performance as a function of the percentage of pixels associated with a dendritic ROI for each session.

Minor:

What is the definition of movement onset? Is it the time to exceed some of the baseline activity SDs? If so, the preceding neural activity may reflect the initial movement. Please add the XY hand positions in Figure 5e and Supplementary Figure 10e, j.

Here, we define movement onset as a purely behavioral measure, which we find by determining the time at which hand speed exceeds a threshold (set as 5% of the peak speed in that trial, although the relative timing of movement onset across trials is largely insensitive to the thresholding technique used). For the offline analysis of neural data, where appropriate, we have re-aligned neural data across trials to this behaviorally-defined movement onset time so as to reveal movement-aligned responses in the single trial or trial-averaged neural activity. This removes the trial-to-trial reaction time variability that would otherwise mask the structure in the neural signals by smearing and averaging across poorly-aligned data.

Given that the hand is entirely stationary before this behaviorally-defined movement onset, we ensure that the early neural activity indeed precedes the movement, consistent with electrophysiological measurements of motor cortical activity as well. We also note that the optimal lag analysis presented in Supplementary Figure 11 argues that the correlation between neural signals and velocity is neural leads behavior by ~50 ms, as expected.

We have clarified this definition in the methods accordingly:

Movement onset was defined as the time when in-plane hand speed exceeded 5% of peak speed for each trial. We note that movement onset is defined entirely behaviorally, independently of neural responses. Reaction time (RT) was defined as the time between the 'go cue' and movement onset.

We note that the XY hand positions and hand velocity, aligned to the same movement onset time, are already present in Figure 5 as panel b and Supplementary Figure 10 as panels b, g. We have clarified in the figure caption that indeed these signals are co-aligned.

Reviewer #3 (Remarks to the Author)

Trautmann, O'Shea, Sun and colleagues have made significant revisions to the manuscript that markedly improved the quality and presentation. While the revisions have largely addressed my concerns, there are still places where I feel the manuscript makes claims that are not well supported by their data, and where the presentation could be improved.

We thank the reviewer for the kind words and patience with the remaining points of concern. We appreciate these points and have worked hard to address each of these to add appropriate language to ensure we do not make claims that are insufficiently supported. We continue to include the details of our methodology in the methods section, as we anticipate (and know from ongoing conversations) that our serotype selection profiling and particular constructs are of interest. But we now add appropriate caveats that ensure that our approach does not constitute a systematic comparison across constructs.

1. Viral expression strategy claims:

The results and discussion sections have been modified to make clear the results do not provide firm conclusions about optimal strategies for GCaMP viral expression. But the paper still maintains several statements in the introduction and discussion that are not supported by the paper's data.

The authors continue to emphasize the possible importance of the NES construct while presenting functional data that is primarily from an injection site that does not use the NES construct and without any data analysis to compare NES injection sites to non-NES sites. (“we modified existing GCaMP6 constructs to promote cellular transport and promote expression of calcium indicators in the dendrites.”; “We designed monkey codon-optimized GCaMP constructs... which may have increased functional expression in apical dendrites...”) I do not feel that the addition of a statement that more data is needed is sufficient given that the paper as written does not present any data that can speak to the importance of NES. If the authors wish to retain these claims, they must provide some data to speak to these points. Otherwise the claims should be removed.

In the introduction, we have removed the statement about NES constructs. In the discussion, we have modified the NES-related statements as follows:

We designed monkey codon-optimized GCaMP constructs, some of which included nuclear export signal (NES) target peptide⁶⁹. While we selected these NES-constructs for the purpose of achieving functional expression in dendrites, we cannot draw any conclusions regarding its efficacy in primate cortex without a systematic comparison between different constructs.

We also adjusted the language in the results section as well:

The AAV1-CaMKII α -GCaMP6f and AAV1-CaMKII α -NES-GCaMP6f as well as primate codon-optimized AAV1-CaMKII α -mGCaMP6f and AAV1-CaMKII α -NES-mGCaMP6f viruses were also tested in the visual cortex of monkey L, and functional signals were observed via widefield imaging in response to visual stimuli for all four viruses (AAV1-CaMKII α -NES-GCaMP6f, Supplementary Fig. 5, recorded using a separate imaging setup, see Methods and ²¹ for more details). We note that because we have not performed a systematic comparison, we cannot be certain whether the primate codon-optimized transgene and/or nuclear export signal (NES) influence GCaMP expression in the primate cortex.

The paper also retains statements emphasizing the importance of AAV serotype work for their success: (“We co-optimized AAV serotype selection by immune-profiling individual macaques to improve viral transfection”)

In the introduction, we have removed the assertion by rephrasing the statement as follows:

We performed immune-profiling for individual macaques to measure pre-existing antibodies before and after viral injection.

Similarly, in the results, the language now reads:

Although it remains unclear whether pre-injection immunological status affects expression of virally-delivered constructs in the CNS⁵⁵, preexisting antibodies may neutralize the virus before transfection and result in low expression. We performed immune-profiling for individual macaques to measure preexisting antibodies before viral injection, with the goal of avoiding viral serotypes that might trigger an immune response (see Methods). While our results did not directly show a causal relationship between preexisting antibodies and viral transfection efficiency, we found that one monkey subject with significant pre-injection anti-AAVs (monkey W) developed significant antibody responses to AAV viruses injected into cortex(Supplementary Fig. 3).

In the discussion, we also added cautionary language:

The probability of achieving successful expression may also have been improved by performing a neutralizing antibody assay to pre-screen viral serotypes against subject-specific immune status, though systematic comparison is needed to validate this approach.

Although our results did not directly show a causal relationship between preexisting antibodies and viral transfection efficiency, we found that monkey subjects with significant pre-

injection anti-AAVs could develop immunoreactivity to small volumes of AAV viruses injected into cortex and thus we selected the AAV1 serotype in the hopes of avoiding an immune response in monkey X. Because this is, as the reviewer points out, essentially a description of our methodology rather than a report of systematically validated scientific findings, we have moved the entire AAV section to the Methods (to accompany the description of the immunoprofiling assay), with a brief description written in the main text of performing immune profiling before selecting AAV1 for monkey X. We have added the following explicit note at the end of this methods section as well:

We caution that we do not have direct evidence that our particular serotype selection ensured better expression in our subject. While these results are consistent with our expectations, further experiments and additional subjects are required to validate the efficacy of this approach. In addition, the impact of immunoreactivity on the longevity of expression at healthy levels and on the success of subsequent additional injections remains to be determined.

2. Quantification of responses and stability across time:

2a. The authors have added more quantification of response distributions, which is a significant improvement. They quantify the fraction of ROIs that show movement-related modulations, and then also further break down that population for ROIs that are identified as axonal/dendritic processes. While I appreciate that the author's ultimate focus in this paper is on the dendritic processes, I am a bit unclear on why this general analysis does not also present results for identified somas. Given the author's reported issues with filled nuclei and lack of functional responses in some cell bodies, this is an important detail to include.

We thank the reviewer for raising this point. In the revised manuscript, we do note the fraction of ROIs that are identified as axonal/dendritic processes, as the reviewer mentioned. However, we cannot conclude that the remainder of the ROIs should not be considered clearly identifiable somas. Many of these not-clearly-dendritic/axonal ROIs are simply portions of dendrites/axons that traverse the imaging plane at a slight angle (in depth), and therefore only occupy a small region of pixels due to the 2P optical-depth-sectioning. These ROIs cannot be reliably identified as neural processes, simply because we do not have 3d volumes taken above and below for all ROIs that could be used to automatically identify cross-sections of neural processes in the imaging plane. Other ROIs not marked as dendrites/axons also appear to be large, swatches in which comparatively weaker, movement-modulated neuropil signal is visible, but in which no individual dendrites/axons can be resolved. Given that we performed imaging in superficial layers I and II of cortex (0-300 μm deep), it is highly likely that the source of this fluorescence is an amalgam of many neuronal fibers, however, we refrain from labeling them as dendritic/axonal in our analysis simply to reflect our ambiguity.

Ultimately, as the reviewer notes, the paper is focused on the dynamics of superficial dendritic and axonal processes. Our offline analysis of neural responses is focused on 36 imaging datasets collected entirely in superficial cortex, where we expect nearly all modulated ROIs to arise from neuronal processes and expect few somatic calcium sources to be present.

While we do note that some clearly identifiable somatic sources were present in the brain, some of which were healthy and modulated, and some which showed signs of unhealthy nuclear aggregation, we did not systematically image these deeper somatic sources.

We have added this note (to the results section) as follows:

ROIs that did not meet this selection criterion comprised mainly processes that cut through the imaging plane at an angle (appearing as small puncta) and background neuropil without clearly distinguishable processes. Relatively few somatic signals were identified at the superficial planes at which we imaged.

And described the selection criterion in the methods more clearly:

We identified putative dendritic/axonal ROIs in the datasets as those ROIs with a computed aspect ratio greater than 2 (ratio of long axis to short axis of ROI shape) and manually verified that this selection criterion identified only ROIs that appeared to be neuronal processes.

2b. The authors also emphasize the stability of functional responses over time (supplementary figures 6 and 13). The within-day analyses are nicely quantified, but the across-day results (fig. S6) are not quantified (the authors themselves at one point say they are “qualitative”) and are not presented in a format that easily facilitates cross-day comparisons. Because the fields of view across days are not spatially registered and identified ROIs vary across days, it is difficult to visually discern stability across days for a given ROI in this figure format. While I appreciate the author’s point (in the response to reviewers) that only focusing on a few example ROIs gives a limited data snapshot, this figure alone does not allow any quantitative or substantive comparison of ROIs across days. Given this limited analysis, I am therefore not clear how well the statement “We...return to the same neurons and observe consistent direction tuning in neuronal processes” is defended.

We agree with the reviewer that a large-scale and quantified assessment of tuning stability, at least across the neurons recorded in these datasets, would be required to make broad claims regarding the stability of neuronal tuning across datasets. As these datasets were not collected with the explicit intent of interrogating this particular question, the overlap of standard features is insufficient to enable automated association across days. As such, we have removed any claims regarding the consistency of tuning, and instead merely note that it is possible to record from the same neurons across multiple sessions. We believe that this is an important point that could inform future studies building on this capability, in which data collection is intended to maximize the alignment of imaging planes across sessions to address specific hypotheses.

We have updated the language in the main text and supplement accordingly:

We leveraged wide-field imaging of vascular fiducial markers to return to the same neurons in one region over seven sessions spanning 13 days (Supplementary Fig. 6).

And

We returned to certain fields of view and observed the same neural processes across multiple experimental sessions (Supplementary Fig. 6).

3. Methodological presentations:

I thank the authors for including more details on methods and what was done. The manuscript has significantly improved in this regard. A few smaller points, however, should still be addressed.

3a. Which monkey and injection site(s) are used should be presented consistently throughout the results/figures. This information is largely present, but not consistently.

We have now added which monkey the data was acquired from in each figure and panel as appropriate, and have noted the injection sites in the figure captions.

3b. For multi-modal registration between CLARITY and functional imaging, I am still unclear on the exact way registration was performed. I follow the logic of figure 7, but how are the overlapping/consistent processes between images determined? For instance, how are the yellow vasculature in panel d and f, and the dendritic process in the white box for panel d, e, and I identified? The way the text is written seems to imply this is partly manual visual inspection. Is that the case? If not, some information on the algorithmic approaches should be provided. If it is being done manually, some discussion of what limits applying algorithmic approaches for registration are required.

Yes, the reviewer is correct that we performed the alignment of fields of view using visual inspection to identify and align large features of the vasculature in images taken with different modalities (or across experimental sessions within a modality). The hierarchical nature of this visual search process is illustrated in Fig. 4 of the main manuscript: Large vascular features that are visible without magnification are used to find smaller reference points.

While it is possible that automated alignment algorithms might be employed to scale up this approach in the future, we feel that this approach could be most helpful for the course-scale registration. In practice this was not challenging, and we were able to localize vascular features precisely (within several pixels), without ambiguity. In addition, non-uniformities in the contrast of vasculature and neurons between modalities greatly increased the challenge of performing alignment with automated processes. To be specific, differences in contrast between *in-vivo* 2P calcium imaging and immunostaining in CLARITY pose great challenges to an automated algorithm. As the automation of this aspect of the work was not the focus and would not improve the alignment results, we did not pursue a fully automated approach.

Minor/specific:

“We developed a highly rigid three-point head restraint system... restricted head motion to just a few microns” – How did you confirm that head motion is the only source of motion error? Your test measures registration error across frames but does not identify the source of this motion. I would revise the sentence to make the distinction clear.

The reviewer is correct that there are multiple potential additive sources of motion in the imaging plane, which are not dissociable from the existing data. We have updated this sentence to reflect this point:

We developed a highly rigid three-point head restraint system (Fig. 3c,d), which is conceptually similar to existing “halo” style head restraint systems⁴⁹⁻⁵¹ but, in conjunction with the tissue stabilizer, restricted the motion of tissue within the imaging plane to 1-2 pixels for the majority of imaging sessions (Supplementary Figs. 1-2).

The Choi et al. paper presents imaging data from S1, not V1 as your citation implies. This paper is also relevant for several of the technical imaging details discussed in the paper, where it is not being referenced, such as imaging stability and expression levels.

Thank you for bringing this to our attention. We have corrected this mistake / omission, as the Choi et al. paper reports injections of GCaMP in, and implantation of, an imaging chamber over both S1 and V1, but we agree that the imaging data presented in Figure 5 appears to be from S1. We have referenced this highly relevant work in our description of the technical advances needed for stable 2P imaging in the discussion:

As previously reported in primary somatosensory cortex[1] , direct stabilization of the brain was required for dorsally-located frontal cortex than in the occipital lobe (e.g.,^{20,21}) due to the effect of gravity.

“We demonstrated that signals imaged from dendrites... exhibited sufficiently high signal-to-noise ratio.” Sufficiently high SNR for what?

We thank the reviewer for highlighting this ambiguous language. Originally, we intended to note that the signal-to-noise ratio was sufficient to be used for decoding of movement direction. (Signal meaning movement related responses, noise meaning any modulation not related to movement and noise in the imaging measurements due to tissue motion, Poisson noise, etc.) However this is redundant given the description of the real time decoding that follows so we have removed this sentence.

I am unclear on why the methods paragraph “For online decode experiments...” is included in the section on 2p imaging methods, rather than the section for oBCI decoding.

These “online decode experiments” paragraphs are now moved to the oBCI decoding section.

References

Chandrasekaran, C. *et al.* (2017) ‘Laminar differences in decision-related neural activity in dorsal premotor cortex’, *Nature communications*, 8(1), p. 614. doi: 10.1038/s41467-017-00715-0.

Degenhart, A. D., Bishop, W. E. and Oby, E. R. (2020) ‘Stabilization of a brain-computer

interface via the alignment of low-dimensional spaces of neural activity', *Nature Biomedical*. nature.com. Available at: <https://www.nature.com/articles/s41551-020-0542-9>.

Gallego, J. A. *et al.* (2020) 'Long-term stability of cortical population dynamics underlying consistent behavior', *Nature neuroscience*, 23(2), pp. 260–270. doi: 10.1038/s41593-019-0555-4.

Kaufman, M. T. *et al.* (2016) 'The Largest Response Component in the Motor Cortex Reflects Movement Timing but Not Movement Type', *eNeuro*, 3(4). doi: 10.1523/ENEURO.0085-16.2016.

Pandarinath, C. *et al.* (2018) 'Inferring single-trial neural population dynamics using sequential auto-encoders', *Nature methods*, 15(10), pp. 805–815. doi: 10.1038/s41592-018-0109-9.

Sun, X. *et al.* (2017) 'Feasibility analysis of genetically-encoded calcium indicators as a neural signal source for all-optical brain-machine interfaces', in *2017 8th International IEEE/EMBS Conference on Neural Engineering (NER)*, pp. 174–180. doi: 10.1109/NER.2017.8008320.

Yu, B. M. *et al.* (2009) 'Gaussian-Process Factor Analysis for Low-Dimensional Single-Trial Analysis of Neural Population Activity', *Journal of neurophysiology*, pp. 614–635. doi: 10.1152/jn.90941.2008.

Reviewer #1 (Remarks to the Author):

I have no more comments on the manuscript. Congratulations on the authors for this important and thorough manuscript!

Reviewer #2 (Remarks to the Author):

The authors have added a lot of analyses to support their findings. I greatly respect the authors' efforts. These improvements mean that this manuscript provides information of value to the field. However, I still have a serious technical concern about the reliability of the dendritic responses. In my opinion, the DBSCAN result provides no information about the cell numbers. The authors should show the distribution of pairwise correlations between dendrites, as shown by Peters et al. (Figure 2C; Nat Neurosci 20, 1133, 2017). This is the simplest method of estimating the proportion of the dendrites from the same neuron and the independence between pairs of dendrites. The correlation matrix would also be useful for visualizing the grouping. The trial-averaged responses of ROIs aligned to movement onset (Supplementary Figure 8C) appear very short and symmetrical, irrespective of whether responses were increasing or decreasing. Although the kinetics of GCaMP6f are faster than those of other GECIs, I doubt that the responses were heavily contaminated by the motion artifact, and only a small subset of dendrites actually showed a physiological response. How do the authors interpret the fast decay (~100 ms!) of the responses? Did these short fluorescent changes really reflect action potentials? When the fluorescent changes are aligned to their onset, does the expected decay appear? Please show that the decrease in activity did not result from z-axis motion artifact. I really need to be convinced about the percentage of dendrites that showed physiological responses to movement.

Reviewer #3 (Remarks to the Author):

The authors have satisfactorily addressed my concerns. I now feel the article is suitable for publication. Congratulations on the well done work.

Reviewer #4 (Remarks to the Author):

I was asked to comment on Reviewer 2's concerns regarding reliability and stability of the recorded dendritic Ca²⁺ signals and if the data shown by the authors demonstrate the capabilities/advantages of the optical BCI.

The authors show their dendritic images could keep stable for about thirteen days (SI Fig. 6). But for BCI application, this period is too short, almost equal to saying it is unworkable. Without long-term reliable signals, a conceptual demonstration of BCI is not convincing. Additionally, the demonstrated period of stable recordings is also shorter than routine neuronal two-photon calcium imaging with months of stability.

I also fully agree with the point 2 from reviewer 2. The author should demonstrate the signal can really drive the oBCI, if they want to make such a statement. To suggest other people to build BCI using apical dendric signal rather than somatic signal, the author should demonstrate advantages in

accuracy and long-term stability of the apical dendritic strategy over routine somatic strategy. Looking at the authors' data and the figures, I am afraid it is very hard to demonstrate either of those two points. The best signals seem from a few highly expressing neurons as the reviewer 2 pointed out. Thus, the recorded signals may not be reliable nor enable high decoding accuracy. I also note that the quality of the somatic images could be improved. In my opinion, the authors have not demonstrated that strategy relying on Ca²⁺ signals from apical dendrites works better than the strategy relying on somatic Ca²⁺ signals.

Dendritic calcium signals in rhesus macaque motor cortex drive an optical brain-computer interface

Eric M. Trautmann*, Daniel J. O'Shea*, Xulu Sun*, James H. Marshel, Ailey Crow, Brian Hsueh, Sam Vesuna, Lucas Cofer, Gergő Bohner, Will Allen, Isaac Kauvar, Sean Quirin, Matthew MacDougall, Yuzhi Chen, Matthew P. Whitmire, Charu Ramakrishnan, Maneesh Sahani, Eyal Seidemann, Stephen I. Ryu, Karl Deisseroth**, Krishna V. Shenoy**

Response to reviewers

11/24/2020

Introduction

We thank the reviewers for their patience and diligence throughout multiple rounds of review. We have performed considerable additional analysis to support the points raised by Reviewer 2, and feel that the resulting manuscript makes a much stronger case as a result. In our estimation, we were able to address all concerns raised by the reviewer in the updated manuscript. Most importantly, we are happy to work to adjust any language in the manuscript that overreaches or misstates the findings of the manuscript in accordance with keeping the focus on the key advance being reported. In addition, we thank reviewers 1 and 3 who have now signed off on the manuscript. Throughout this document, reviewer comments are indicated in blue, responses to reviewers are in normal black text, and *text added to the manuscript is italicized*.

Reply to Reviewer #2

The authors have added a lot of analyses to support their findings. I greatly respect the authors' efforts. These improvements mean that this manuscript provides information of value to the field.

We thank the reviewer for these kind comments and continued patience in reviewing this manuscript.

However, I still have a serious technical concern about the reliability of the dendritic responses. In my opinion, the DBSCAN result provides no information about the cell numbers. The authors should show the distribution of pairwise correlations between dendrites, as shown by Peters et al. (Figure 2C; Nat Neurosci 20, 1133, 2017). This is the simplest method of estimating the proportion of the dendrites from the same neuron and the independence between pairs of dendrites. The correlation matrix would also be useful for visualizing the grouping.

We apologize for any misunderstanding regarding the DBSCAN analysis. In response to the reviewer's helpful suggestions, we have performed new analysis to assess the correlations between dendrites, as performed by Peters et al. (2017). The reviewer expressed concern that many of the dendrites could originate from very few source neurons, and the reviewer suggested that a clustering analysis could reveal the existence of these parent neurons as discrete clusters. Our DBSCAN analysis rejected the existence of multiple discrete clusters, but indeed, as the reviewer notes, it does not indicate the precise cell numbers as well.

We have now performed the analysis suggested by the reviewer by following [1]. We also note (and apologize) that we had intended to cite this paper alongside [2], as it provides additional evidence for the similarity of signals in dendritic and somatic neural compartments in the mouse motor cortex. We have now corrected this accidental omission.

Figure excerpted from Peters et al. (2017) highlighting referenced analysis.

We computed the correlation between all pairs of dendritic ROIs for each imaging FOV. We present the results below for three representative datasets highlighted in the manuscript, and summarize the results.

Distances between each pair of ROIs were computed as $d = (1 - \text{correlation})$. **Top row:** Dendrograms were computed using hierarchical clustering, and the order of ROIs was optimized to maximize the sum of the similarities between adjacent ROIs. **Middle:** correlation matrix between ROI signals, using the optimized ROI ordering. **Bottom row:** histogram of pairwise correlation coefficients between pairs of ROIs.

In general, the correlation matrix captures the dominant structure of dendritic ROIs exhibiting both increasing and decreasing responses during movement. Some additional structure appears to also reflect directional tuning preference among the ROIs for each of four reach directions. However, the pairwise correlation histograms below reveal that similarity between dendritic ROIs is low on average. These highlighted datasets are representative. Below is the histogram of pairs of simultaneously imaged dendrites, aggregated across the 36 imaging datasets:

Histogram of correlation coefficients between all pairs of dendritic ROIs from the same imaging dataset, aggregated across all 36 imaging datasets.

If we use the threshold of $\rho \geq 0.8$ from Peters et al. for considering two ROIs to originate from the same cell, we would retain 4178 of the original 4365 imaged dendritic ROIs. If we use a more conservative threshold of $\rho \geq 0.6$, we would retain 3537 / 4365. This overall dissimilarity among dendritic ROIs is unlikely to arise due to noise. If we filter down to the dendritic ROIs that exhibit statistically significant tuning (i.e. whose movement related signals are significantly larger than the noise), we retain 2134 / 2196 and 1896 / 2196 of the tuned dendritic ROIs after merging above these two thresholds, respectively. We note, however, that it is possible that there could be compartmental differences within branches of the same dendritic arbor that create pairs of ROIs from the same neuron with dissimilar responses.

For comparison, we also repeated this analysis using (unpublished) electrophysiology data recorded in macaque motor cortex during a similar reaching task. Here we recorded electrical spiking activity from well isolated single neurons using a NeuroPixel probe inserted acutely in the cortex. Using Kilosort2 to sort the spikes detected across the probe channels into individual neurons, we can then use the spike triggered electrical multi-channel spiking waveform of each neuron to determine its location along the probe. Using these electrical waveforms and each neuron's response statistics (e.g. the interspike interval distribution), we can be extremely confident that these spikes come from hundreds of individual neurons. We have repeated the pairwise correlation analysis below by filtering each neuron's spike trains using the same slow exponential filter to simulate intracellular calcium. The analysis

successfully reports that nearly all (416/418) of the neurons should be considered from separate cells. Also, the presence of blocky structure in this correlation heatmap below (where we have high confidence that every neuron recorded on the probe is distinct) indicates that this structure should be considered as evidence of functional similarity during the task, rather than as evidence that two cells originate from the same neuron.

Pairwise correlation analysis repeated on an example NeuroPixel electrophysiology dataset for comparison. **Left:** Mean waveform for each sorted neuron plotted at its estimated centroid location along the probe shank. **Right:** Spike trains were filtered using an exponential filter to simulate intracellular calcium vs. time, which were then used to compute pairwise correlations between neurons. **Top row:** Dendrograms were computed using hierarchical clustering, and the order of neurons was optimized to maximize the sum of the similarities between adjacent ROIs. **Middle:** correlation matrix between ROI signals, using the optimized ROI ordering. **Bottom row:** histogram of pairwise correlation coefficients between pairs of ROIs.

To be clear, our intent is not to argue that the impact of our manuscript primarily revolves around high cell counts. Indeed, there are many ways that the number of simultaneously imaged neurons could be improved in future work, e.g. by improving expression, longevity / ongoing maintenance of the optical window, larger FOV imaging methods. The primary contribution of our manuscript centers on our demonstration for the first time that superficial dendritic calcium signals in the macaque motor cortex provide access to movement-tuned signals throughout the cortical lamina. We demonstrate that these directionally-tuned signals can then drive a discrete online decoding system in

real-time. We agree with the reviewer that imaging from more dendrites and from more unique neurons, would certainly improve the accuracy, reliability, and stability of the optical BCI system.

We have incorporated these figures into Supplementary. Figure 8.

The trial-averaged responses of ROIs aligned to movement onset (Supplementary Figure 8C) appear very short and symmetrical, irrespective of whether responses were increasing or decreasing. Although the kinetics of GCaMP6f are faster than those of other GECIs, I doubt that the responses were heavily contaminated by the motion artifact, and only a small subset of dendrites actually showed a physiological response. How do the authors interpret the fast decay (~100 ms!) of the responses? Did these short fluorescent changes really reflect action potentials? When the fluorescent changes are aligned to their onset, does the expected decay appear?

We thank the reviewer for raising this concern. The offline image processing pipeline that we used to create these original figures employed a high-pass filtering step on the ROI fluorescence signals in order to eliminate the effects of any baseline drift. This filtering did not affect the presence or tuning of the responses but did unfortunately accelerate the slow decay of the tail. We note first that none of the other analysis relied on these high-passed filtered traces; only these trial-averaged raster plots employed this processing pipeline. The image processing code we used for the other analysis, and for the updated versions of these figures uses an iterative baseline identification approach described in [3]. This approach has the advantage of identifying the F_0 baseline during inactive regions of time without removing the slower frequency content of the responses as with high-pass filtering.

We include the updated version of two of the rasters below, and have updated figure 5 and supp. figure 10 accordingly.

Updated versions of Figure 5D, Supp Fig 8d: Trial-averaged responses of dendritic ROIs, normalized to baseline fluorescence, for the four reaching directions indicated by the colored arrow in the bottom left of each raster. ROIs are sorted by preferred

direction beginning with rightwards and proceeding counterclockwise; triangular ticks at left edge indicate locations of preferred directions of up-right, up-left, down-left, down-right.

We note however, that these responses still exhibit faster decay kinetics than typically expected for GCaMP6f. We have performed the suggested analysis, by aligning the transients at each dendritic ROI to their onset. The responses that we recorded appear asymmetric with a much faster rise than decay, as anticipated. This rise-aligned fluorescence time course exhibits a decay time of $\tau_{\text{off}} = 212$ ms, which is consistent with reported tau off values for GCaMP signals from single action potentials, verified using in vivo cell attached recordings ($\tau_{\text{off}} = 204.8$ ms) [4, Supp. Table 3, reproduced below].

Collected dendritic ROI response transients, aligned to their onset. Individual responses from dendritic ROIs are collected for the reach direction which drives the maximal positive response, and then aligned using the first derivative of the fluorescence signal to localize the onset. Average across ROIs is shown in white, and an exponential fit to the decay is shown in red.

Supplementary Table 3 | GCaMP comparison in mouse V1 *in vivo* (cell attached recording).

Sensor	$\Delta F/F_0$, 1 AP	Decay $\tau_{1/2}$ 1 AP (ms)	Rise τ_{peak} 1 AP (ms)	Detection efficiency (at 1 % false positive)
GCaMP5K	3.6±1.9 %	268±20	60±20	19±12%
GCaMP6s	23±3.2 %	550±52	179±23	99±0.2%
GCaMP6m	13±0.9 %	270±23	80±7	94±0.5%
GCaMP6f	19±2.8 %	142±11	45±4	84±6%

Table reproduced from [4, Supp. Table 3]

Of course, the reported decay kinetics for calcium events arising from 10 action potentials in dissociated culture is certainly slower ($\tau_{\text{off}} = 577$ ms) [4, Supp. Table 1]. However, dendritic calcium signals have been previously reported to exhibit faster kinetics than somatic signals in mice [2]. In [2], Beaulieu-Laroche et al. record somatic and dendritic GCaMP6f calcium signals in V1 cortical slices. Following somatic current injection to drive 10 action potentials at 200 Hz, which produced backpropagating dendritic spikes, they recorded the somatic and dendritic calcium transients reproduced below. We have annotated their figure 3H and estimated the half-life decay time $\tau_{1/2}$, from which we compute $\tau_{\text{off}} = \tau_{1/2} / \ln(2)$. The dendritic responses we record in superficial macaque motor cortex exhibit slightly slower decay kinetics than these mouse V1 *ex vivo* signals; consequently, we believe that the asymmetric dendritic calcium signals could very well arise from backpropagating action potentials as well.

Reproduced from Beaulieu-Laroche et al. [2]. Blue annotation lines were used to estimate the half-life decay time, from which we computed the decay time constant $\tau_{off} = \tau_{1/2} / \ln(2)$.

We note that in the macaque motor cortex, baseline firing rates are quite high relative to mice (20-50 Hz is common even at rest), and that the modulation of firing rate during movement is very large (modulation depths of 30-50 Hz are typical). Consequently, it is very unlikely that we would see calcium transients arising from individual backpropagating action potentials. Rather, our signals may reflect a temporally low-pass filtered version of the underlying modulation of firing rate, and the exact kinetics would depend on a convolution of the underlying response modulation, the kinetics of the reporter, and the dynamics of calcium buffering in the dendrites. Future work exploring the relationship between optical physiology of the dendrites and spiking activity recorded electrically may provide a better understanding of the forward model that links spikes to dendritic calcium [e.g., 5]. Future experiments in macaques may also identify improved constructs, viral vectors, or titers to optimize expression levels in cortex for superficial dendritic imaging in motor cortex. Regardless, we believe that multiple lines of evidence (including those discussed below) indicate that the calcium transients we recorded reflect modulations of neural firing rates accompanying movement execution.

Please show that the decrease in activity did not result from z-axis motion artifact. I really need to be convinced about the percentage of dendrites that showed physiological responses to movement.

With regard to the motion artifact, we have described in detail our engineered mechanisms to minimize brain motion in the manuscript. However, the question of z-motion is a very salient concern, and we thank the reviewer for highlighting it. Z-motion of the brain uncorrelated with behavior would of course not produce a decodable signal. However if the z-motion were caused by the arm movement itself resulting in mechanical displacement of the brain, then we would risk decoding movement related signals from artifacts produced by movement itself. This was of course a primary concern during our decoding experiments. Here, we present two new lines of evidence that we believe argue that our calcium signals are physiological in origin and do not arise from the movement artifact. In addition, we note that in three control sessions for online oBCI decode in brain areas not expression GCaMP (but containing autofluorescent puncta), we were not able to decode reach direction at levels above chance. Naturally, future work could indeed leverage volumetric imaging or real-time tracking of the brain surface to better quantify and minimize the effect of z-motion [e.g. 6].

1) Time course of fluorescence modulation: First, for a majority of the dendritic ROIs that we imaged, the fluorescence signal modulation began before the movement. If these signals originated due to the result of motion artifacts, we would expect to observe fluorescence modulation closely following the start of a movement. We have performed an analysis to quantify the time at which each tuned dendritic ROI exhibits directionally tuned responses to movement (see legend). As reported in the manuscript 81.3% of dendritic ROIs exhibit a statistically significant response to movement (rank sum test, $p < 0.01$), and 50.3% of dendritic ROIs exhibited statistically significant directional tuning in their responses (ANOVA, $p < 0.01$). Of these directionally tuned ROIs, the majority begin responding well in advance of any detectable hand motion, when the monkey is holding still at the central hold location. During the instructed delay period, we penalize any movement of the hand that arises before the visual go cue, and eliminate any trials with delay period movement from further analysis. Any motion of the body (adjusting posture in the primate chair) typically causes the marker attached to the hand to move, as detected by a polaris IR imaging sensor with 10 micron resolution. As such, via behavioral training over many months, the monkey learns to remain very still in advance of the go cue. The amount of z-axis motion in the brain just before movement begins is comparable to the other portions of a trial where the dendritic ROI calcium signals are not strongly modulated.

Supplementary Figure 11: Top panel: cumulative proportion of direction-tuned dendritic ROIs that exhibit directionally tuned responses at or before a given time point relative to behaviorally defined movement onset ($t=0$). The time at which a given ROI exhibits a directionally tuned response is determined by performing a Kruskal Wallis nonparametric one-way analysis of variance on the single-trial fluorescence measurements at each time in a sliding window. The response time is taken when significance at $p < 0.01$ is reached for 5 consecutive 10 ms windows. **Bottom panel:** hand speed (mean \pm std.) aligned to behaviorally defined movement onset for the same trials.

Fluorescence changes which *precede* movement cannot result from motion artifact, and are much more likely to result from real physiological changes in calcium concentrations within the imaged neurons, which in turn reflects underlying firing rate changes which generate the movement. Very similar patterns of modulation preceding movement (but during the peri-movement epoch) have been observed in extracellular physiology in the same instructed-delay reaching task. This is also consistent with the optimal lag we identified for decoding hand kinematics from calcium signals, with calcium signals preceding hand kinematics by 50 ms maximizing the correlation. This matches the value typically reported in electrophysiological datasets as well [7]. If the hand movement itself drove

motion artifacts (in z or in plane) that in turn modulated the fluorescence value, we would expect that this correlation should appear at zero lag or slightly positive lags, depending on mechanical delays created by the viscoelastic properties of brain tissue within the skull.

2) Decode of reaches to return to center: We performed a new analysis on data from previously unanalyzed timepoints during a trial. The instructed-delay reaching task we use is a center-out task, in that we focus on reaching movements beginning at the central hold region towards peripherally located targets. By training and the constraints of the computer task, these outwards reaches are required to be brisk and accurate in order to receive a juice reward. However, interleaved between trials is a second reach back to the central target, which is typically slower. We reasoned that these slower reaches would cause lower forces due to lower arm acceleration, and would be less likely to cause motion artifacts. We found that dendritic ROIs exhibited movement related responses which precede these back to center reaches as well. The exact response pattern is not identical as a function of reach direction (as we would expect) because motor cortical neural responses reflect the precise velocity and position of the movement being performed. While this analysis does not rule out z-axis motion on it's own, it helps to build even more confidence that these signals are primarily physiological rather than mechanical in origin.

Left and right columns include behavior and neural responses for center-out and back-to-center reaches, respectively. **Top panel:** hand positions on the 2D plane during reaching colored by the direction towards the reach endpoint. Black dot: movement onset; gray dot: target acquisition. **Middle panel:** Hand speed (mean \pm s.e.m.), grouped by reach direction. **Bottom panel:** trial-averaged fluorescence responses for one representative dendritic ROI during reaching (same ROI in left and right plot).

Summary: In summary, the data presented here provides extensive evidence to support the conclusion that the signals we are observing are neural in origin, and do not exclusively result from a very small number of neurons per FOV. This includes the judgement of a number of co-authors who are experts in calcium imaging in mice, despite the fact that the dendritic signals imaged in primate cortex bear some features which are distinct from calcium imaging in mice. We agree that imaging quality can be improved through future work to improve the density of healthy viral expression throughout cortical lamina. In addition, we are happy to adjust any language so as to not overstate any claims or features of the data in this manuscript.

References:

1. Peters AJ, Lee J, Hedrick NG, O'Neil K, Komiyama T. Reorganization of corticospinal output during motor learning. *Nat Neurosci*. 2017;20: 1133–1141. doi:10.1038/nn.4596
2. Beaulieu-Laroche L, Toloza EHS, Brown NJ, Harnett MT. Widespread and Highly Correlated Somato-dendritic Activity in Cortical Layer 5 Neurons. *Neuron*. 2019;103: 235–241.e4. doi:10.1016/j.neuron.2019.05.014
3. Peters AJ, Chen SX, Komiyama T. Emergence of reproducible spatiotemporal activity during motor learning. *Nature*. 2014;510: 263–267. doi:10.1038/nature13235
4. Chen, Tsai-Wen, Trevor J. Wardill, Yi Sun, Stefan R. Pulver, Sabine L. Renninger, Amy Baohan, Eric R. Schreiter, et al. 2013. “Ultrasensitive Fluorescent Proteins for Imaging Neuronal Activity.” *Nature* 499 (7458): 295–300.
5. Siegle JH, Ledochowitsch P, Jia X, Millman D, Ocker GK, Caldejon S, et al. Reconciling functional differences in populations of neurons recorded with two-photon imaging and electrophysiology. *Cold Spring Harbor Laboratory*. 2020. p. 2020.08.10.244723. doi:10.1101/2020.08.10.244723
6. Choi J, Goncharov V, Kleinbart J, Orsborn A, Pesaran B. Monkey-MIMMS: Towards Automated Cellular Resolution Large- Scale Two-Photon Microscopy In The Awake Macaque Monkey. 2018 40th Annual International Conference of the IEEE Engineering in Medicine and Biology Society (EMBC). 2018. pp. 3013–3016. doi:10.1109/EMBC.2018.8512994
7. Kaufman MT, Churchland MM, Ryu SI, Shenoy KV. Cortical activity in the null space: permitting preparation without movement. *Nat Neurosci*. 2014;17: 440–448. doi:10.1038/nn.3643

Reply to Supplementary Reviewer #4

I was asked to comment on Reviewer 2's concerns regarding reliability and stability of the recorded dendritic Ca²⁺ signals and if the data shown by the authors demonstrate the capabilities/advantages of the optical BCI.

The authors show their dendritic images could keep stable for about thirteen days (SI Fig. 6). But for BCI application, this period is too short, almost equal to saying it is unworkable. Without long-term reliable signals, a conceptual demonstration of BCI is not convincing. Additionally, the demonstrated period of stable recordings is also shorter than routine neuronal two-photon calcium imaging with months of stability.

We respectfully note that several disparate concepts are being combined in this critique. We imaged for a period of eight months in Monkey W, and over five months in Monkey X, demonstrating that this primate imaging model can serve as a basis for investigations in basic science or BCI development. We make no claims regarding the stability of the decoder across multiple sessions, however. Instead, as reported in the paper, we re-train the imaging decoder at the start of each session. This is customary practice for electrophysiology-based BCI in both primates and humans, though some exciting experiments have begun to investigate the possibility of using a decoder across longer periods of time [1-3]. SI Figure 6 is included simply to illustrate that the identification of some cells across multiple sessions is possible, but we make no claims about the tuning stability, image decoder stability, or stability of GCaMP expression across those multiple sessions. These data were not collected with the intent of performing such across-session analysis. As such, slight differences in the orientation and depth of the imaging plane result in differences in the resulting image, making such quantitative comparison difficult or impossible. We anticipate that future work, intending to pursue interesting questions regarding the stability of expression or of an imaging decoder, could build on the work reported here to conduct a study of stability across sessions, but this is not within the scope of the current experiments or novelty of this manuscript.

The concern that a BCI should be demonstrated over the course of many weeks, however, does not reflect the current state of the art BCI literature. The question of whether a decoder can remain stable across multiple sessions is an active area of research, but we note that many state-of-the-art decoders in 2020 would not function without training at the start of each session. As such, the utility of the approach presented in this manuscript hinges on whether we can observe and decode neural signals within a single session. We have demonstrated successful imaging over the period of many months, but in each session, perform decoder training at the beginning of the day. This is customary practice for the majority of BCI experiments.

Developing decoder algorithms that are stable across multiple days represents interesting and important future work, but is also outside of the scope of the results presented in this manuscript. In addition, multi-week decoder stability is not a standard to which current electrophysiology-based BCI experiments are held. Even chronically-implanted multielectrode arrays such as the Utah array routinely yield nonstationary neural signals over time courses of days [4,5].

As such, we do not feel that this concern is appropriate for dismissing the novelty of the work presented in this paper. The key demonstrations in our work are: 1) mechanically stable 2P imaging during a motor behavior in rhesus macaque monkeys, 2) proof of concept online decode of reaching behavior, 3) a suite of engineering, surgical, and software advances, all of which we are making available to the broader community, to build on these results to perform real-time imaging experiments.

We understand that the term “stability,” used throughout the paper, could be interpreted to mean either the stability of viral expression, imaging, or decoder across many sessions or weeks. Here, we exclusively intend for “stability” to refer to the reduction or elimination of frame-to-frame jitter which could be introduced via mechanical forces imparted on the implant during a motor behavior. We have added the following sentence to clarify this point in the manuscript:

We have also explicitly noted that we were not able to assess tuning stability over time, due to the fact that these data were not collected with exactly the same imaging depth or orientation, as previously described:

We note, however, that the data reported here were not collected with the exact same field of view or imaging depth, leading to the across-session differences visible in supplementary Fig. 6, which make it difficult to perform quantitative assessment of the tuning stability, stability of viral expression, or assessment of image decoder stability across multiple sessions.

...

We returned to certain fields of view across multiple experimental sessions and observed similar neural structures, although we were not able to quantitatively assess tuning stability over time (Supplementary Fig. 6).

I also fully agree with the point 2 from reviewer 2. The author should demonstrate the signal can really drive the oBCI, if they want to make such a statement. To suggest other people to build BCI using apical dendritic signal rather than somatic signal, the author should demonstrate advantages in accuracy and long-term stability of the apical dendritic strategy over routine somatic strategy.

We interpreted the comments from reviewer 2 as primarily concerned with making sure that the observed signals were neural in origin, rather than the result of motion artifacts, and not a comparison of dendritic vs. somatic signals. We have performed extensive additional analysis to demonstrate that these signals indeed reflect neural activity, which multiple lines of evidence now support. These data, however, were not collected with the purpose of evaluating the relative information content of signals originating from dendritic vs. somatic sources.

Our manuscript does not argue that recording from dendritic signals is more desirable than recording somatic signals from superficial layers. Instead, we assert that it is *possible* to record dendritic signals from neurons, which are otherwise completely inaccessible to 2P imaging (without resorting to much more disruptive methods like implanting large prisms or GRIN lenses within the cortex). We further demonstrate the technical advances necessary to run an online decoder, which amounts to a considerable software and hardware engineering challenge. We of course acknowledge the potential for improvements in online decoder performance with improvements to the viral expression or with ultra wide field of view imaging to record from more neurons simultaneously.

We also note that this is the first demonstration of 2P calcium imaging in macaque motor cortex, and that these datasets should not be considered routine. The macaque motor cortex is significantly thicker (> 2.5 mm), presents additional challenges in terms of achieving healthy levels of reporter expression and mechanically stable imaging. We are of course aware and highly impressed by two photon imaging experiments which are routinely performed in mice

(including optical BCI demonstrations), relatively recent work in marmosets, and imaging in visual regions of the macaque. However, the macaque remains a model organism of tantamount translational importance to the development of clinical BCIs to treat paralysis. Our work represents a significant first step towards enabling these kinds of all-optical experiments in macaques, where two-photon calcium imaging remains a recent and actively developed neurotechnology.

We are not suggesting, however, that oBCI should replace electrophysiology derived BCI systems, nor are we arguing that our oBCI system would be used in humans. This work provides a foundation on which to perform additional investigation and scientific discovery, and we acknowledge the multiple ways in which imaging performance may be improved in future work.

Looking at the authors' data and the figures, I am afraid it is very hard to demonstrate either of those two points. The best signals seem from a few highly expressing neurons as the reviewer 2 pointed out. Thus, the recorded signals may not be reliable nor enable high decoding accuracy. I also note that the quality of the somatic images could be improved. In my opinion, the authors have not demonstrated that strategy relying on Ca²⁺ signals from apical dendrites works better than the strategy relying on somatic Ca²⁺ signals.

As addressed above, we do not feel that this critique accurately assesses the purpose of this manuscript or the data presented. We agree that the data here is not sufficient to compare the relative contributions of somatic vs. dendritic signals. While we feel that these could form the basis of fascinating scientific investigations, this is outside the scope of what we claim or address in this paper.

Dendritic vs. somatic signals: It is important to note that we are not suggesting that researchers should build BCIs using apical dendritic signals rather than somatic signals. In this manuscript, we demonstrate that apical dendritic signals provide a means of imaging responses from neurons spanning all layers of cortex, including those >1.5 mm deep that remain optically inaccessible. This manuscript does not argue, however, that dendritic signals provide a superior readout of intended movement than somatic signals. Rather, we argue that dendritic signals exhibit sufficient movement-related modulation that we can reconstruct single trial neural trajectories (and thereby investigate neural dynamics), and also drive a real-time decoder (and thereby perform closed loop experiments where the readout is known, as has been leveraged in many electrophysiology BCI experiments). Our study focuses on dendritic signals because they are readily accessible relative to somatic signals, but we do not intend to assert any claim of functional superiority over somatic signals. Naturally, future work could investigate this relationship, and improve the accuracy and stability of imaging in macaque motor cortex overall.

Stability and reliability: We also feel that it is important to note that our goal was *not* to demonstrate reliability and stability of the recorded dendritic signals across multiple sessions, but rather to demonstrate that superficial dendrites may be used to optically interrogate neurons spanning all layers of macaque motor cortex. A key component of this is to demonstrate stability within an imaging session, which we have shown conclusively through both a characterization of pixel registration offsets in offline analysis and via the continued decoder performance for many hundreds of trials.

Deep neurons are not optically accessible without destructive (large) optical implants. Reliability and stability across sessions remain avenues for future improvements, but here we demonstrate within-session recording stability. This is sufficient to assess the recorded dendrites' movement related responses and direction tuning to train and implement a real time decoding system. We note that this is analogous to acute electrophysiology. If we record extracellular

potentials from one or many neurons using a linear probe, it is virtually impossible to record from the same set of neurons with a second penetration on a second session. Despite this limitation, linear electrode arrays are routinely used for neuroscientific investigations and for BCI demonstrations.

Taken together, we believe that we have conclusively demonstrated the ability to image populations of neurons, including dendritic signals originating from layer 5 cells, for lengths of time lasting many months in rhesus macaque monkeys. We use this preparation to implement an online imaging decoder, which is trained and run within a single session, and this preparation is capable of maintaining stable imaging for many hundreds of trials. This collectively represents a large quantity of individual innovations in surgery, hardware, software, and computation. Collectively, these provide a platform for future investigations aimed at basic science and BCI development.

References

- [1] Sussillo, David, Sergey D. Stavisky, Jonathan C. Kao, Stephen I. Ryu, and Krishna V. Shenoy. 2016. "Making Brain-machine Interfaces Robust to Future Neural Variability." *Nature Communications* 7 (1): 13749.
- [2] Nuyujukian, Paul, Jonathan C. Kao, Joline M. Fan, Sergey D. Stavisky, Stephen I. Ryu, and Krishna V. Shenoy. 2014. "Performance Sustaining Intracortical Neural Prostheses." *Journal of Neural Engineering* 11 (6): 066003.
- [3] Gallego, J. A., Perich, M. G., Chowdhury, R. H., Solla, S. A. & Miller, L. E. Long-term stability of cortical population dynamics underlying consistent behavior. *Nature Neuroscience* **23**, (2020).
- [4] Chestek, Cynthia A., Vikash Gilja, Paul Nuyujukian, Justin D. Foster, Joline M. Fan, Matthew T. Kaufman, Mark M. Churchland, et al. 2011. "Long-Term Stability of Neural Prosthetic Control Signals from Silicon Cortical Arrays in Rhesus Macaque Motor Cortex." *Journal of Neural Engineering* 8 (4): 045005.
- [5] Suner, Selim, Matthew R. Fellows, Carlos Vargas-Irwin, Gordon Kenji Nakata, and John P. Donoghue. 2005. "Reliability of Signals from a Chronically Implanted, Silicon-Based Electrode Array in Non-Human Primate Primary Motor Cortex." *IEEE Transactions on Neural Systems and Rehabilitation Engineering: A Publication of the IEEE Engineering in Medicine and Biology Society* 13 (4): 524–41. Sussillo, David, Sergey D. Stavisky, Jonathan C. Kao, Stephen I. Ryu, and Krishna V. Shenoy. 2016. "Making Brain-machine Interfaces Robust to Future Neural Variability." *Nature Communications* 7 (1): 13749.

Reviewer #2 (Remarks to the Author):

The authors have appropriately responded to my concern by performing additional analyses. I am convinced by their arguments.

I recommend the paper for publication.